# Integrin-alpha-6+ Candidate stem cells are responsible for whole body regeneration in the invertebrate chordate *Botrylloides diegensis*

Susannah H. Kassmer [1 ✉], Adam D. Langenbacher[2] & Anthony W. De Tomaso[1]

Colonial ascidians are the only chordates able to undergo whole body regeneration (WBR), during which entire new bodies can be regenerated from small fragments of blood vessels. Here, we show that during the early stages of WBR in *Botrylloides diegensis*, proliferation occurs only in small, blood-borne cells that express *integrin-alpha-6 (IA6), pou3* and *vasa*. WBR cannot proceed when proliferating IA6+ cells are ablated with Mitomycin C, and injection of a single IA6+ Candidate stem cell can rescue WBR after ablation. Lineage tracing using EdU-labeling demonstrates that donor-derived IA6+ Candidate stem cells directly give rise to regenerating tissues. Inhibitors of either Notch or canonical Wnt signaling block WBR and reduce proliferation of IA6+ Candidate stem cells, indicating that these two pathways regulate their activation. In conclusion, we show that IA6+ Candidate stem cells are responsible for whole body regeneration and give rise to regenerating tissues.

[1] Molecular, Cellular and Developmental Biology, University of California, UCEN Rd, 93106 Santa Barbara, CA, USA. [2] Molecular, Cell, and Developmental Biology, University of California, 610 Charles E Young Dr S, Los Angeles, CA, 90095 Los Angeles, CA, USA. ✉email: Susannah.kassmer@lifesci.ucsb.edu

The ability to regenerate missing structures is diverse and widespread among the metazoan phyla. Some invertebrate species within the Platyhelminthes, Cnidaria, and Echinodermata, can regenerate whole bodies from small fragments of tissue, while some vertebrates such as amphibians and fish can regenerate ablated limbs and distal structures of some organs. In contrast, groups such as insects, birds, and mammals have nearly lost the ability to regenerate[1–3]. Just like their general regeneration abilities, the cellular sources for regeneration are highly variable among regenerating species. In the Cnidarian Hydractinia and the Planarian Schmidtea, pluripotent stem cells are present in the adult and are responsible for regeneration, while in amphibians and zebrafish, lineage-committed progenitors arise from de-differentiation[1,4].

In the majority of chordates, the ability to regenerate following a major injury is severely limited, usually resulting in scar formation. In contrast, a group of invertebrate chordate species; colonial ascidians of the genus Botrylloides, have been shown to regenerate whole bodies, including all tissues and organs, from small fragments of the vasculature. This process is called Whole Body Regeneration (WBR)[5–10].

We are studying WBR in Botrylloides diegensis. Ascidians are the sister group of vertebrates, and begin life as a tadpole larvae with a typical chordate body plan. Following a free-swimming stage, the larvae settle and metamorphose into a sessile invertebrate adult, called a zooid. Zooids have a complex anatomy: they are filter feeders with siphons and a branchial basket, gastrointestinal tract, central and peripheral nervous system, endocrine glands, a heart, and a circulatory system consisting of 8–12 blood cell types. In colonial ascidians that belong to the Botryllinae, such as B. diegensis, zooids reproduce asexually, and the adult body plan consists of a colony of clonally derived individuals[5,10–14] embedded within a gelatinous structure called a tunic (Fig. 1a). Asexual regeneration results in an ever-expanding number of zooids, embedded in the tunic, and linked by a common, extracorporeal vasculature (Fig. 1a). This vascular bed extends beyond and encircles the zooids, and at the periphery of the colony terminates in structures called ampullae (Fig. 1a).

Several species of Botrylloides can also use WBR to produce a zooid following injury. WBR occurs in the vasculature, and is initiated by surgical removal of all zooids, or separation of small vascular fragments from the rest of the colony (Fig. 1b). WBR progresses through distinct visual stages[9]. For the first 24 h following surgical ablation, the blood vessels undergo regression and remodeling (stage 1, Fig. 1b). During this time, blood flow, usually powered by the hearts of each zooid, continues and is driven by contractions of the remaining ampullae[8]. During the next 48 h, blood vessel remodeling forms a dense, contracted network (stage 2, Fig. 1b). An opaque mass of nonpigmented cells becomes apparent; creating a clear area that is the presumptive site of bud development (stage 3, arrow in Fig. 1b). The mass of cells next forms into a hollow, blastula-like epithelial sphere. The vascular epithelium then wraps itself around this sphere, leading to the formation of a distinct visible double vesicle. Vessel fragments usually reach the double vesicle stage (stage 3) within 48–120 h post injury. The inner vesicle increases in size, while undergoing a series of invaginations and evaginations that lead to organogenesis (stage 4, 96–168 h, Fig. 1b), and the eventual regeneration of a zooid (stage 5, 120–240 h post injury). WBR is defined as complete when the new zooid is actively filter feeding, and occurs within a range of 7–10 days (stage 5; Fig. 1b). The zooid immediately commences normal palleal budding, and the colony regrows. WBR can be induced in fragments as small as 5 ampullae, and is not dependent on the stage of asexual reproduction of the colony[5,7].

In the present study, we aim to assess the cellular origins of WBR in B. diegensis, specifically the role of circulatory cells in this process. WBR has been studied in different species of Botrylloides[5,6,10]. In all cases a population of cells with an undifferentiated appearance, termed hemoblasts, have been suggested to initiate this regenerative process[5–7]. In B. violaceus, 15–20 small hemoblasts that express Piwi protein have been shown to aggregate under the epidermis of a blood vessel during early WBR[6]. These cells are present during the early vesicle stage and occasionally within the epithelium of a vesicle[6]. In B. leachii, Blanchoud et al. showed an increase in the population of hemoblasts very early after injury during WBR[8]. Knockdown of piwi mRNA in B. leachii resulted in inhibition of WBR[15]. These results had suggested that blood-borne stem cells might play a role in WBR in Botrylloides, but to date, it has never been directly tested whether such cells give rise to regenerating tissues.

In Botryllus schlosseri, the blood contains self-renewing, lineage restricted germline stem cells (GSCs) that migrate to newly developing buds during repeated cycles of asexual reproduction, where they give rise to eggs and testes[16–18]. Germline and gonad development in B. schlosseri has been reviewed previously[19]. In a previous study, we have shown that these GSCs can be enriched by flow cytometry using Integrin-alpha-6 (IA6) as a marker[20] and express piwi as well as other genes associated with germ cells, such as vasa, and pumilio[18,21,22]. Since vasa and piwi are part of the germline multipotency (GMP) program[23], and IA6 is a biomarker for various kinds of mammalian stem cells, including embryonic stem cells and primordial stem cells[24], we hypothesized that IA6 and vasa might likewise be expressed in blood-borne stem cells that are involved in WBR in Botrylloides.

Here we use a rescue assay and a prospective isolation strategy to identify the cells that give rise to regenerating tissues. We show that during the very early stages of WBR in B. diegensis, cell proliferation occurs only in blood-borne cells that express integrin-alpha-6, pou3, vasa, and piwi. IA6+ cells are required for WBR and lineage tracing using EdU labeling reveals that they directly give rise to regenerating tissues. Finally, we show that proliferation of IA6+ cells during WBR is regulated by Notch and Wnt signaling.

## Results

**Cell proliferating during early WBR express Integrin-alpha-6 and pou3.** To assess whether Integrin-alpha-6 enriches for cells expressing stem cell-associated genes in B. diegensis, we used an antibody against the extracellular part of human Integrin-alpha-6 (IA6). An alignment (shown in the Supplementary Methods section) of human and B. diegensis integrin alpha 6 protein sequences shows that both proteins share significant overlap (=50% positive amino acid alignment). We used this antibody to isolate IA6+ cells from the blood of healthy colonies by flow cytometry (sorting strategy shown in Supplementary Fig. 5) and quantified the expression of integrin-alpha-6 mRNA and germline pluripotency genes such as vasa, piwi1, piwi2, and pou3. For the latter, we hypothesize that the octamer binding transcription factor Pou3 plays a role in stem cell pluripotency in ascidians, similar to Oct4 in mammalian pluripotent cells. Oct4 is a member of the Pou class 5 gene family; a vertebrate specific family of pou genes[4,25,26]. It is likely that Pou5 inherited this function from an ancestral Pou paralog[26], and in the cnidarian Hydractinia echinata, Pou3 plays a role in stem cell pluripotency[27]. We cloned pou3 from B. diegensis and B. schlosseri and constructed a phylogenetic tree that confirms the close relationship of ascidian pou3 to pou5 (Supplementary Fig. 1A, sequences in Supplementary Data 1 and 6). Pou3 is expressed in the developing germline of palleal buds in B. diegensis colonies (Supplementary Fig. 1B). IA6+ cells are highly enriched for expression of ia6, pou3, vasa, piwi1, and piw2, (Supplementary Fig. 1D) when compared to IA6− cells. To assess the expression

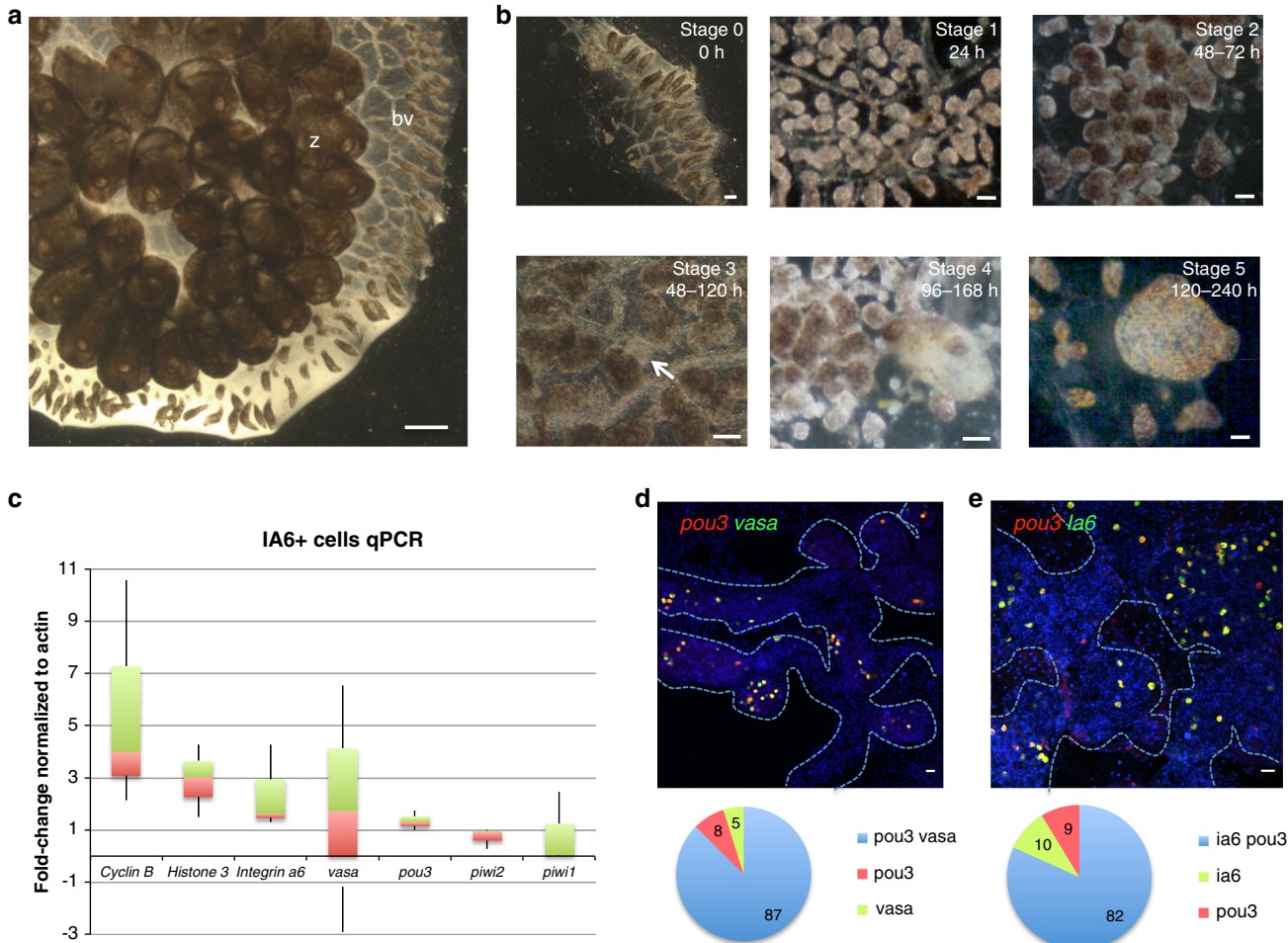

**Fig. 1 Stages of whole body regeneration, gene expression in IA6+ cells. a** *Botrylloides diegensis*, whole colony brightfield image, dorsal. Zooids (z) are embedded in a common, transparent tunic. Blood vessels (bv) extend throughout the tunic and are shared by the zooids of the colony. At the periphery of the colony, blood vessels form terminal sacs termed ampullae. Images are representatives of 20 independent experiments. Scale bar 1 mm. **b** Stages of whole body regeneration, brightfield images. Blood vessels are surgically separated from the colony (stage 0). During the first 24 h, the blood vessel begins to remodel (stage 1). After 48 h, blood vessels become condensed and highly pigmented (stage 2). Regeneration begins when a double vesicle is formed, consisting of two layers of epithelium (stage 3, white arrow). This double vesicle undergoes organogenesis (stage 4) and gives rise to a new, filter feeding body (stage 5). Images are representatives of 20 independent experiments. Scale bars 200 μm. **c** qPCR analysis showing expression of *cyclin B, integrin-alpha-6, vasa, pou3, piwi2,* and *piwi1* in IA6+ cells isolated by flow cytometry. Data are expressed as averages of fold changes normalized to *actin*, n = 4. Box plot shows interquartile range (the distance between the upper and lower quartiles) and whiskers indicate the minimum and maximum of the data. **d** FISH showing co-expression of *pou3* (red) and *vasa* (green) in stage 0. DNA was stained with Hoechst (blue). Blue dashed lines outline blood vessel boundaries. Scale bar 20 μm. Single-positive and double-positive cells were counted using the cell counter feature in FIJI, and for each stage, four images from four independent samples were counted. Pie graph shows averages of percentages of *pou3/vasa*-double-positive cells as well as *pou3* and *vasa* single-positive cells. **e** FISH showing co-expression of *pou3* (red) and *ia6* (green) in stage 0. DNA was stained with Hoechst (blue). Blue dashed lines outline blood vessel boundaries. Scale bar 20 μm. Single-positive and double-positive cells were counted using the cell counter feature in FIJI, and for each stage, four images from four independent samples were counted. Pie graph shows averages of percentages of *pou3/ia6*-double-positive cells as well as *pou3* and *ia6* single-positive cells. Source data are provided as a Source Data file.

levels of these genes within the IA6+ cells, we normalized their expression levels to *actin*. *Cyclin B, histone 3, integrin a6, vasa,* and *pou3* are expressed at higher levels than *actin* (Fig. 1c). Previously, only one *piwi* gene had been reported in *B. leachii*[28], but we found that, like most animals, *B. diegensis* has two *piwi* genes (sequences in Supplementary Data 3 and 4)[29,30]. In IA6+ cells, *piwi2*, and *piwi1* expression is lower than *actin* (Fig. 1c). Using double-labeled FISH, we confirmed that 81% of either *ia6*+ or *pou3*+ cells co-express both genes, while 9 and 10% express only *pou3* or *ia6*, respectively (Fig. 1e). The overlap between *vasa* and *pou3* is 87% (Fig. 1d).

To assess whether *ia6*+ *pou3*+ cells are involved in responding to injury, we analyzed the expression of these two genes in

proliferating cells during the early time points after surgical separation of blood vessels from the colony. We used *histone 3* mRNA expression as a marker of proliferating cells, as it is upregulated in S-phase of the cell cycle in plants and animals, and has been used as a marker for cell proliferation in in situ hybridization previously[31–35]. Analyzing expression of *histone 3* (*h3*) together with *integrin-alpha-6* by double fluorescent in situ hybridization (FISH), we found that 90% of all *ia6*+ cells proliferate in the blood of a healthy colony (stage 0, Fig. 2a, b). After separation of blood vessels form the colony, the percentage of *ia6* that proliferate stays high during stages 1 and 2 (94 and 94%, respectively, (Fig. 2a, b), and the total number of proliferating *ia6*+ *h3*+ cells increases dramatically (Supplementary Fig. 2B, compare

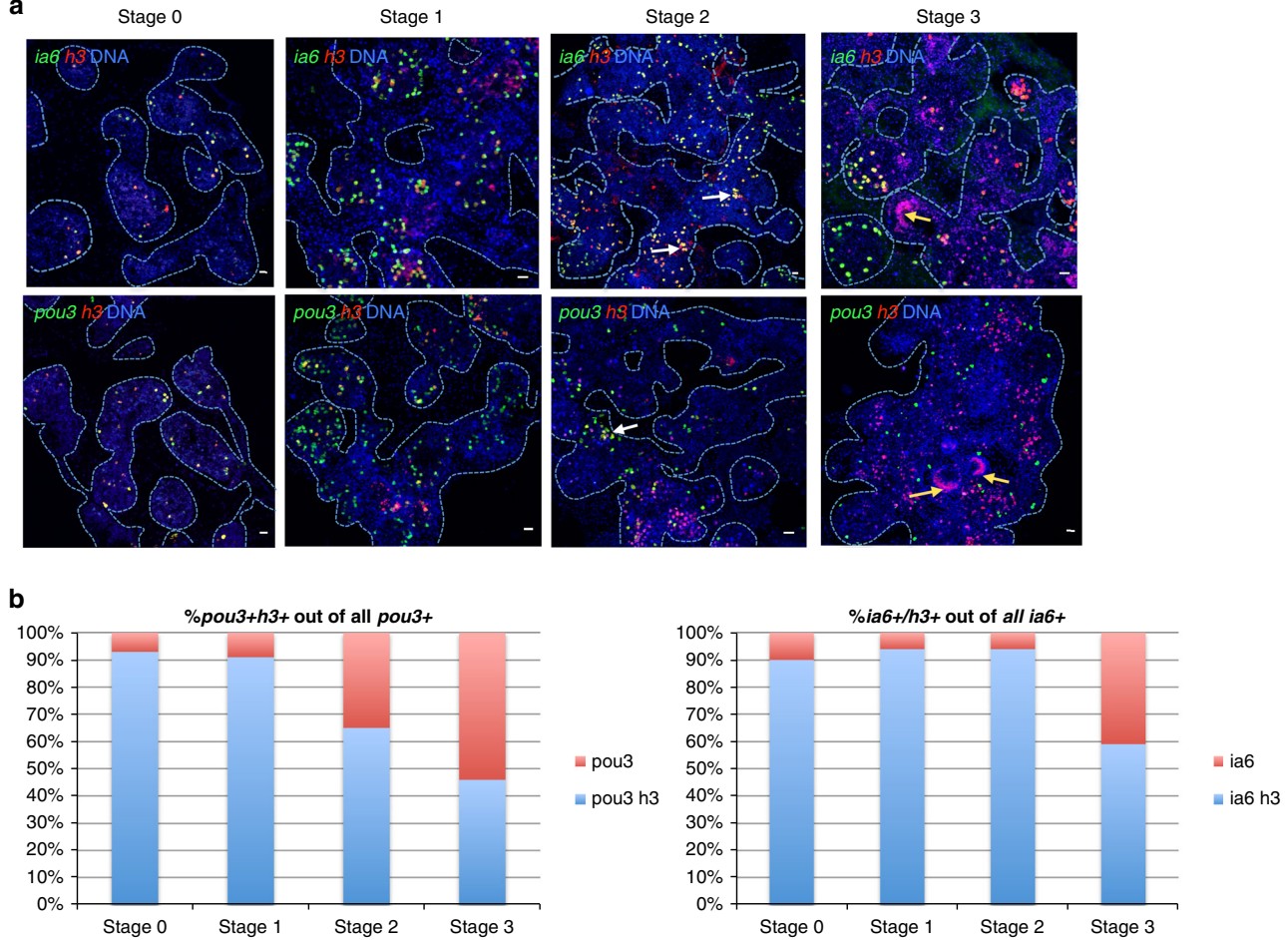

**Fig. 2 Whole body regeneration is associated with proliferation of blood-borne *integrin-alpha-6*+ cells. a** Fluorescent in situ hybridization (FISH) showing expression of *integrin-alpha-6* (*ia6*, green) and *histone 3* (*h3*, red) during stages 0, 1, 2, and 3 of WBR. White arrows indicate *ia6*+ cells beginning to cluster during stage 2. As clusters increase in size, they begin to lose *ia6* expression. As cell clusters differentiate to form the blastula-like structure, *ia6* expression is not detected (yellow arrows). DNA was stained with Hoechst (blue) in all panels. Blue dashed lines outline blood vessel boundaries. Images are representatives of four independent experiments. Scale bars 20 μm. **b** Single-positive (*ia6*, *pou3*, or *h3*) and double-positive (*ia6/h3* or *pou3/h3*) cells were counted using the cell counter feature in FIJI, and for each stage, four images from four independent samples were counted. Graph shows percentages of *ia6/h3* double-positive or *pou3/h3* double-positive cells among all *ia6*+ or *pou3*+ cells. Source data are provided as a Source Data file.

stage 0 to stage 1 and 2). Furthermore, *ia6*+ cells make up the majority of proliferating cells during the early stages of WBR, as the number of *h3* single-positive cells is consistently much lower than that of *ia6*+ *h3*+ double-positives in stages 0, 1, and 2 (Supplementary Fig. 2B). During stages 1 and 2, some proliferating *ia6*+ cells aggregate (white arrows in Fig. 2, stage 2), a structure that we refer to as regeneration foci. Cells in the aggregates continue to proliferate, and the structure increases in size (Fig. 2, stage 3, yellow arrows). As these foci increase in size, many proliferating cells lose *ia6* expression (Fig. 2, stage 3, yellow arrows), indicating the generation of differentiating progeny. The aggregate next begins to form the epithelial sphere (Fig. 2, stage 3, red arrows), which will eventually close into a double vesicle and continue developing into the new body. *Ia6*+ cells continue to be present in the blood vessels surrounding the developing epithelial spheres and continue to proliferate (Fig. 2a, b).

Very similar results are obtained when analyzing the proliferation of *pou3*+ cells by *h3* and *pou3* double FISH. 93% of all *pou3*+ cells proliferate in a healthy colony (stage 0) and this percentage stays high during stage 1 (Fig. 2a, b, Supplementary Fig. 2B). By stage 3, 54% of *pou3*+ cells become quiescent, and cell proliferation is predominant in *pou3*- regenerating double

vesicles (yellow arrows in Fig. 2, stage 3, graph in Supplementary Fig. 2B). *Pou3*+ cells continue to be present in the blood vessels surrounding the developing epithelial spheres and developing bodies and continue to proliferate (Fig. 2a, b, Supplementary Fig. 2A). Together, these results show that *ia6*+/*pou3*+ cells proliferate in steady state and make up the majority of proliferating cells during the early stages of WBR. Based on these findings, we hypothesize that *ia6*+ *pou3*+ proliferating cells are stem cells responsible for WBR.

**IA6+ cells are required for regeneration.** To functionally assess whether proliferating IA6+ cells are involved in WBR, we ablated proliferating cells using the drug Mitomycin C (MMC, 60 μM). Vascular tissue was soaked in MMC immediately after surgery and treated for 24 h until stage 1, when overlap between *ia6* and the proliferation marker *histone 3* is 94% (Fig. 2). In MMC-treated vessels, proliferating cells and ia6+ cells are completely eliminated at 24 h, and at 5 days post treatment, *pou3*+ or *histone3*+ cells remain absent (Supplementary Fig. 3A). Elimination of proliferating cells with Mitomycin C results in subsequent loss of regenerative capacity (Fig. 3a). Normally, regeneration is

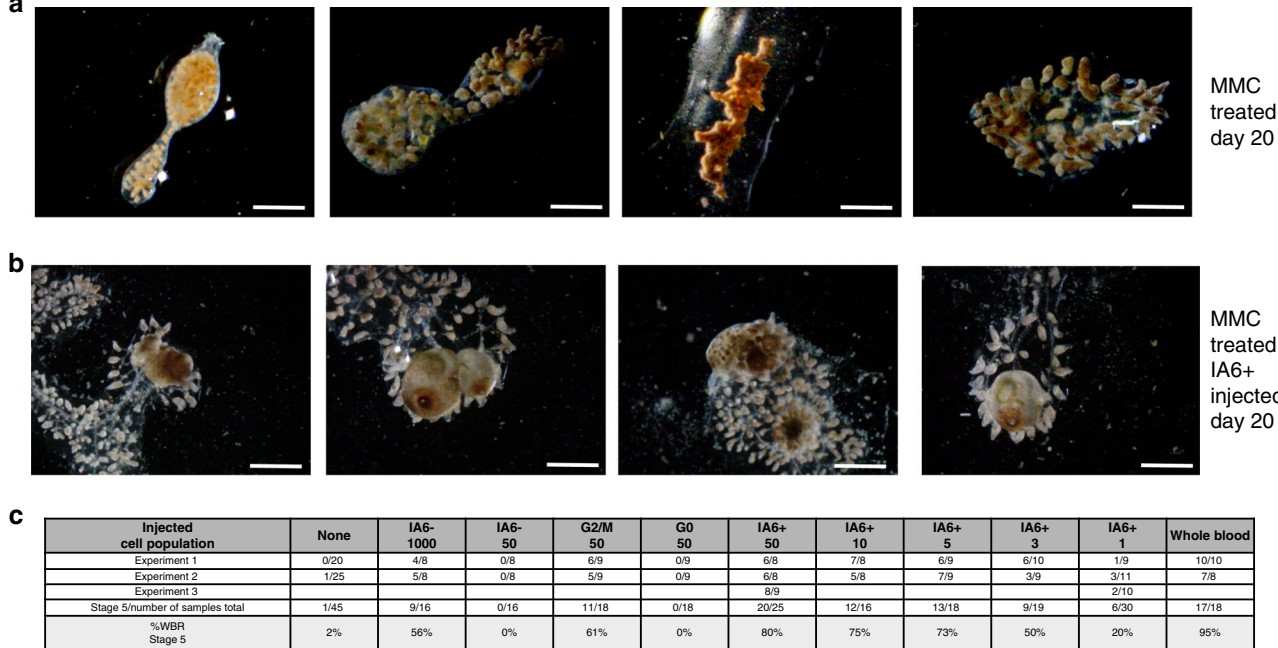

**Fig. 3 IA6+ cells are required for whole body regeneration. a** Mitomyin C (MMC) treatment prevents regeneration. Brightfield images of vessel fragments 20 days post MMC treatment. Vessel fragments were treated with MMC for 16 h immediately after surgery. After 16 h, MMC was removed, and the samples followed for 20 days. Scale bar 1 mm. **b** Injection of IA6+ cells 24 h after MMC treatment rescues WBR. Brightfield images of MMC-treated vessel fragments 20 days post injection of IA6+ cells. Scale bar 1 mm. **c** Table showing rates (percentages) of WBR (reaching stage 5) 20 days after injection of different cell populations. For each condition, 16–20 samples were injected in two independent experiments.

| Injected cell population | None | IA6-1000 | IA6-50 | G2/M 50 | G0 50 | IA6+ 50 | IA6+ 10 | IA6+ 5 | IA6+ 3 | IA6+ 1 | Whole blood |
|---|---|---|---|---|---|---|---|---|---|---|---|
| Experiment 1 | 0/20 | 4/8 | 0/8 | 6/9 | 0/9 | 6/8 | 7/8 | 6/9 | 6/10 | 1/9 | 10/10 |
| Experiment 2 | 1/25 | 5/8 | 0/8 | 5/9 | 0/9 | 6/8 | 5/8 | 7/9 | 3/9 | 3/11 | 7/8 |
| Experiment 3 | | | | | | 8/9 | | | | 2/10 | |
| Stage 5/number of samples total | 1/45 | 9/16 | 0/16 | 11/18 | 0/18 | 20/25 | 12/16 | 13/18 | 9/19 | 6/30 | 17/18 |
| %WBR Stage 5 | 2% | 56% | 0% | 61% | 0% | 80% | 75% | 73% | 50% | 20% | 95% |

complete after 11–14 days (Fig. 1), but MMC-treated samples do not reach stage 4 or stage 5 even after 20 days (Fig. 3a), and most MMC-treated blood vessels appear to be permanently arrested in stage 2 (Fig. 3a). However, regeneration can be rescued in MMC-treated vasculature by injection of 2000 total blood cells isolated from an untreated, healthy individual, demonstrating that WBR depends on blood-borne cells (Fig. 3c). To identify the population of cells responsible for this rescue, we injected different populations of cells isolated by flow cytometry and assessed rescue of WBR (Fig. 3c). We initially compared cells in the G2/M phase of the cell cycle versus those in G0, and found that 50 cycling cells could rescue regeneration, while 50 G0 cells could not. We next compared IA6+ and IA6− populations. While 50 IA6− cells are not able to rescue WBR, injection of 1000 IA6− cells achieved a 57% rescue efficiency (Fig. 3c). This could be due to cell-sorting impurities or it could suggest that a rare population of IA6− stem cells exists (about 1/2000), which is supported by the finding that some *pou3+* cells are *ia6-* (Fig. 1e). In contrast, 80% rescue is achieved by injecting only 50 IA6+ cells (Fig. 3b, c). We next carried out limiting dilution analyses of the IA6+ population, and found that injection of a single IA6+ cell isolated from the blood of a healthy animal can rescue WBR in 20% of MMC-treated samples (Fig. 3c). Using ELDA analysis software (http://bioinf.wehi.edu.au/software/elda/[36]), we calculated the estimated frequency of IA6+ cells capable of rescuing WBR to be about 1 in 11.3 (upper estimate 7.84, lower estimate 16.4). These results show that IA6+ Candidate stem cells can independently rescue WBR.

**IA6+ Candidate stem cells give rise to regenerating tissues in developing bodies.** Since IA6+ Candidate stem cells appear to be functionally required for WBR, we wanted to assess whether progeny from IA6+ cells is incorporated into regenerating tissues. We used EdU to label IA6+ cells in steady state in an intact animal and subsequently track the presence of their progeny in regenerating tissues by transplanting them into MMC-treated vessel fragments. The modified thymidine analog EdU (5-ethy-nyl-2′-deoxyuridine) is incorporated into newly synthesized DNA and detected in fixed tissues using a fluorescent dye. We ensured that EdU could be detected at stage 3 of WBR in vessels injected with EdU at stage 0 (Supplementary Fig. 3B), while uninjected samples showed no signal (Supplementary Fig. 3B). IA6+ cells were isolated from donor animals that had been injected with EdU the day before to allow EdU incorporation into the DNA of cycling cells. 1000 EdU-labeled IA6+ cells were injected into MMC-treated recipients as in the rescue experiments described above. When recipients reached stages 2–4 of WBR, the regenerating tissues were fixed and stained for EdU (Fig. 4). By stage 2, EdU+ cells were present in large and small cell aggregates of early regeneration foci (Fig. 4, red outlines, white arrows) as well as in circulation (Fig. 4). After transplantation of Edu-labeled IA6+ cells, many bright Edu-positive cells in the blood of the rescues sample continue to express *pou3*, even after WBR has progressed to the double vesicle stage (Supplementary Fig. 3C, pink arrows). This suggests that *pou3+ ia6+* cells self-renew in the recipient and give rise to *pou3+* cells. At stage 4, EdU+ cells could be identified in the tissues of the regenerating body that are undergoing organogenesis (Fig. 4, red outlines and white arrows). In one case, we also saw EdU+ cells either part of or directly underneath the outer epithelium of the regenerating body (Fig. 4, green arrow). These results show that IA6+ cells are the source of regenerating tissues in newly developing bodies.

**Proliferation of IA6+ Candidate stem cells and WBR require Notch and Wnt signaling.** We next aimed to analyze the expression of components of the Notch- and Wnt-signaling pathways in proliferating cells from the blood during the early time points of WBR. These pathways are known to play roles in regulating stem cell activity, proliferation of blastema cells and cellular differentiation in other organisms[37–44] and are upregulated during

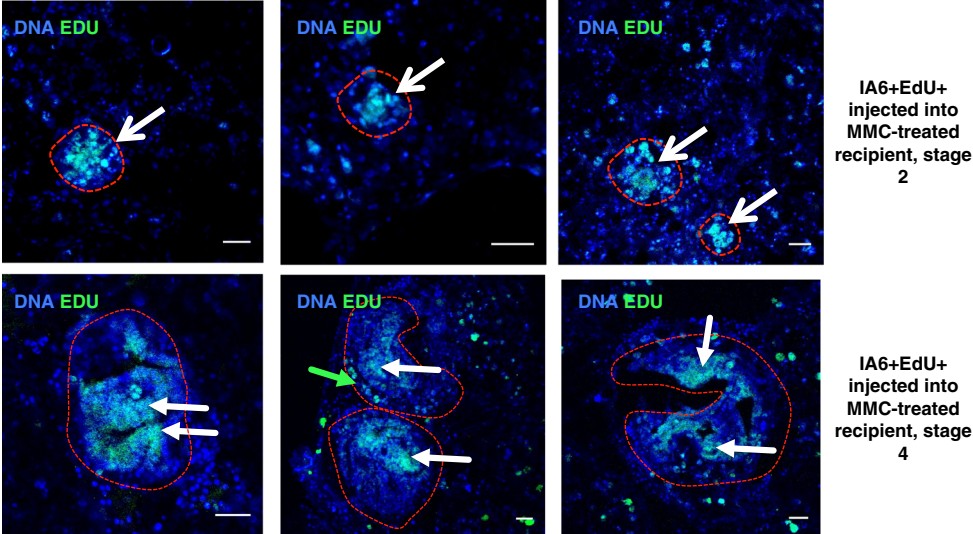

**Fig. 4 IA6+ cells give rise to regenerating tissues.** Overlay of nuclear staining (Hoechst, blue) and Edu (green). IA6+ cells were isolated from Edu-treated animals and injected into MMC-treated vessel fragments. Edu-positive cells give rise to regeneration foci at late stage 2 or regenerating bodies at stage 4 (white arrows). Scale bars 20 μm. Images are representative of ten samples from two independent experiments.

WBR in *Botrylloides*[9]. We isolated cycling G2/M cells by flow cytometry (Fig. 5a, complete sorting strategy shown in Supplementary Fig. 5) from stage 0 as well as at 24 and 48 h after injury of vessel fragments. We specifically isolated G2/M cells instead of IA6+ cells in order to be able to analyze gene expression in all proliferating blood cells, even those that may have begun to downregulate IA6 as they differentiate (Fig. 2). As expected, G2/M cells isolated from the blood of healthy, unmanipulated animals (0 h), express high levels of *ia6*, mitosis-specific *cyclin b*, and *pou3* (Fig. 5a). *Notch 1, notch 2* and the downstream gene *hes1*, as well as the Wnt pathway components *frizzled5/8, disheveled*, and *beta catenin*, are also expressed, indicating active Notch and Wnt signaling in proliferating cells (Fig. 5b). Twenty-four hours after injury, *notch2, frizzled 5/8* are upregulated in G2/M cycling cells compared to 0 h (Fig. 5c). Forty-eight hours after injury, *notch2, pou3* are even more highly upregulated. *Notch1* is downregulated at 24 and 48 h (Fig. 5c). *Piwi2* is slightly upregulated in cycling G2/M cells at 48 h together with *vasa* (Supplementary Fig. 4). These results show that the pluripotency/stem cell related genes *pou3, ia6, piwi2*, and *vasa* are highly expressed in cells that proliferate during the early stages of WBR. Double FISH for *notch1* and *h3* shows that *notch1+* cells proliferate during stage 0 and stage 1, but fewer of these cells are present in stage 2 (Fig. 6a). However, the number of *notch2+ h3+* cells increases by stage 2 (Fig. 6a). These results suggest that *notch1* is primarily responsible for regulating proliferation of IA6+ Candidate stem cells during steady state and at early time points post injury. In stage 2, *notch1*-positive cells become more quiescent and fewer in number. At the same time, *notch2* is expressed in a higher proportion of proliferating cells. This could be related to a role of *notch2* in regulating proliferation of differentiating progenitors, while *notch1* might be associated with stem cell maintenance.

To assess whether Notch or Wnt signaling are required for WBR, vessel fragments were allowed to regenerate in the presence of inhibitors of either Notch (DAPT) or Wnt signaling (Endo-IWR). Inhibition of either Notch or canonical Wnt signaling blocked regeneration in a dose dependent manner (Fig. 6b). In either 2 μM of the Notch-signaling inhibitor DAPT or 1 μM of the canonical Wnt-signaling inhibitor Endo-IWR, the vessel tissue underwent remodeling up to stage 2 in both treatments, but never progressed beyond stage 2 while kept in drug (for up to

4 weeks), and appeared otherwise healthy and alive. Upon removal of the inhibitors (after 96 h of treatment), regeneration progressed normally (14/15 samples reached stage 5 for IWR and 15/16 for DAPT, Fig. 6b), indicating that regeneration is only halted, but that all cell types required for WBR are still viable and fully functional. As controls, we used Exo-IWR which is a 25-fold less active against the Wnt/β-catenin pathway, and the gamma-secretase modulator E2012, which does not affect the Notch specific gamma-secretase. Both control drugs Exo-IWR (1 μM) and E2012 (1 μM) did not affect regeneration rates (Fig. 6b).

We next assessed for the presence of proliferating cells in drug-treated vessel fragments using double FISH for *h3* and *pou3*. Following 72 h of treatment, fewer *pou3*-positive cells are present in drug-treated vessels compared to controls (Fig. 6c), and regeneration foci do not form. Of the few *pou3+* cells that remain, very few (Notch inhibitor) to none (Wnt inhibitor) are proliferating. We conclude that inhibition of signaling downstream of Wnt and Notch receptors blocks IA6+ Candidate stem cells from entering the cell cycle. Because IA6+ Candidate stem cells remain quiescent, they are unable to form regeneration foci. Inhibition of Notch or Wnt signaling likely also affects differentiation of IA6+ Candidate stem cells, as there are much fewer proliferating *pou3-* cells compared to controls. Upon removal of the inhibitors, IA6+ Candidate stem cells are able to resume proliferation and differentiation normally. In addition, the cells that are producing the Wnt and Notch ligands are still capable of doing so when the inhibitors are removed. Together, these results show that WBR depends on both Wnt and Notch signaling, and that both signaling pathways control the response of IA6+ Candidate stem cells to injury and affect their proliferation and differentiation.

## Discussion

Here, we show that Integrin-alpha 6-positive blood-borne cells are responsible for WBR and give rise to regenerating tissues in an invertebrate chordate.

At the level of a single cell, we have found that IA6+ Candidate stem cells can give rise to regenerating tissues during WBR, and that WBR cannot proceed when IA6+ cells are ablated. We show that IA6+ cells are constantly dividing in healthy colonies and

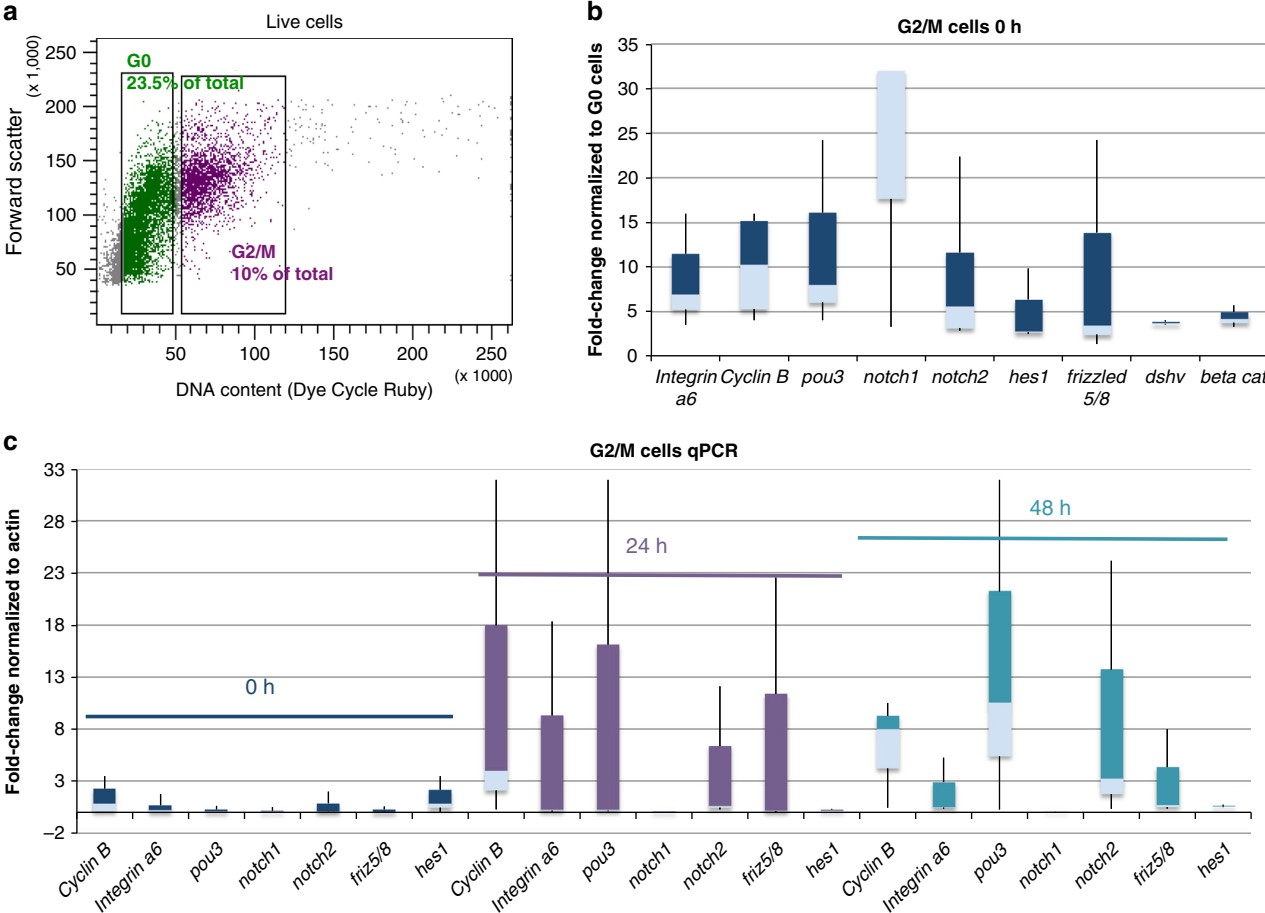

**Fig. 5 Gene expression in cycling blood-borne cells in steady state and during WBR. a** Isolation of cycling G2/M cells by flow cytometry. Cells were stained with the live DNA stain Dye Cycle Ruby. Linear analysis of fluorescence intensity plotted against forward scatter (cell size) shows a clear separation of G2/M cells with 4n DNA content and G0/G1 cells with 2n DNA content. G2/M cells comprise 10% of live cells. **b** qPCR analysis showing expression of, *integrin-alpha-6, cyclin b, pou3, notch1, notch2, hes1, frizzled 5/8, disheveled (dshv)*, and *beta catenin* in G2/M cells in steady state (0 h). Data are expressed as fold changes normalized to G0 cells from three independent experiments. Box plot shows interquartile range (the distance between the upper and lower quartiles) and whiskers indicate the minimum and maximum of the data. **c** qPCR analysis showing expression of *cyclin b, integrin-alpha-6, pou3, notch1, notch2,* and *frizzled 5/8* in G2/M cells at 24 and 48 h post injury. Data are expressed as fold changes normalized to actin, from three independent experiments. Box plot shows interquartile range (the distance between the upper and lower quartiles) and whiskers indicate the minimum and maximum of the data. Source data are provided as a Source Data file.

express genes associated with pluripotency, such as *pou3*. IA6+ Candidate stem cells also express germ plasm components such as *vasa* and *piwi*, yet we have shown that they directly give rise to somatic tissues during WBR. So far, we have developed two main hypotheses about the nature of these cells.

The first hypothesis is that *Botrylloides* maintains a pluripotent stem cell used for WBR. In several other invertebrate species, germ plasm components such as *vasa, nanos,* and *piwi* are expressed in cells which have somatic potential and have therefore been defined as GMP genes[23]. In the primordial stem cell hypothesis, these GMP expressing cells are called Primordial stem cells and are defined as evolutionarily conserved stem cells that give rise to germ cells and can contribute to somatic tissues[45]. Examples of germ plasm containing cells that give rise to somatic tissues include small micromeres from sea urchins, neoblasts in planarians, i-cells in hydrozoan cnidarians and archeocytes in sponges. Besides its known role as a translational regulator in the maintenance of germline cells, Vasa is essential for mitotic progression in stem cells with somatic potential, such as sea urchin blastomeres or planarian neoblasts[46,47]. The common stem cell gene repertoire of ancestral metazoan stem cells comprises *cyclin B1* and *vasa*[48]. Both of these genes are expressed in IA6+

Candidate stem cells (Fig. 1c). *Vasa*-positive GSCs are present in the blood of *B. schlosseri* and persist throughout adult life and give rise to gonads during repeated rounds of asexual reproduction[16,19]. We hypothesize that in some species of colonial ascidians, the *vasa*+ stem cell population retains somatic potential that is utilized during WBR, and that a subset of IA6+ cells might be related to primordial stem cells. The estimated frequency of IA6+ cells capable of rescuing WBR is about 1 in 11.3, suggesting some underlying heterogeneity in this cell population.

An alternative hypothesis is that a normally lineage restricted IA6+ germline progenitor is triggered to proliferate and differentiate into somatic tissues following the loss of all zooids. This is intriguing, as it has been shown that in a range of vertebrate species (from fish to mammals), primordial germ cells are not fully committed to germline fate and maintain somatic potential until they reach the genital ridge[49–51]. Thus, the processes underlying WBR in *Botrylloides* might be somewhat similar to teratoma formation in mammals. Teratomas are germ cell tumors that are derived from primordial germ cells that migrate to ectopic sites during embryogenesis and differentiate into somatic cell types, forming tissue structures from all three germ layers:

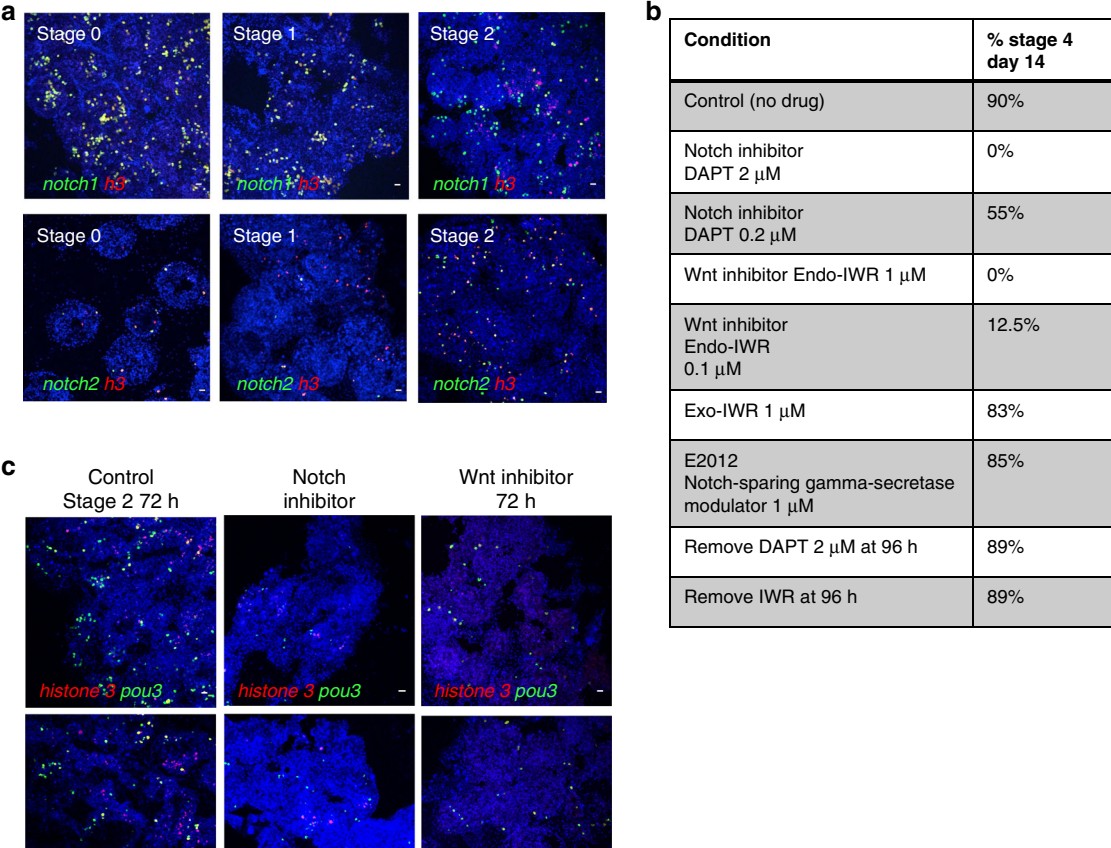

| Condition | % stage 4 day 14 |
|---|---|
| Control (no drug) | 90% |
| Notch inhibitor DAPT 2 μM | 0% |
| Notch inhibitor DAPT 0.2 μM | 55% |
| Wnt inhibitor Endo-IWR 1 μM | 0% |
| Wnt inhibitor Endo-IWR 0.1 μM | 12.5% |
| Exo-IWR 1 μM | 83% |
| E2012 Notch-sparing gamma-secretase modulator 1 μM | 85% |
| Remove DAPT 2 μM at 96 h | 89% |
| Remove IWR at 96 h | 89% |

**Fig. 6 Inhibition of Notch or Wnt signaling prevents WBR and blocks proliferation of pou3+ cells. a** Top panel: FISH for *notch1* (green) and *h3* (red) during steady state (stage 0) and during early stages of WBR. Bottom panel: FISH for *notch2* (green) and *h3* (red) during steady state (stage 0) and during early stages of WBR. Images are representatives of four independent experiments. Scale bar 20 μm. **b** Table showing the percentages of vessel fragments reaching stage 4 at day 14 post injury during treatment with different doses of inhibitors of canonical wnt signaling (Endo-IWR), Notch-signaling (DAPT) or control drugs Exo-IWR (1 μM) and the Notch-sparing gamma-secretase modulator E2012 (1 μM). The number of colonies that reached stage 4 was counted at 14 days post injury. Percentages represent the number of fragments reaching stage 4, with n = 14–18 fragments for each condition. Controls (subclones of the same genotype) received vehicle only. **c** FISH for *pou3* (green) and *h3* (red) showing a reduction in proliferating *pou3+* cells 72 h after surgery when treated with an inhibitor of either Notch or Wnt signaling. Images are representatives of four independent experiments. Scale bars 20 μm.

endodermal, mesodermal, and ectodermal. In *Botrylloides*, the mechanisms that normally restrict circulatory germline progenitors to germ cell fate might be temporarily lifted following injury and subsequent WBR. In *B. schlosseri*, we have shown that *vasa*-positive GSCs are lineage restricted and give rise to new gonads during repeated rounds of asexual reproduction[16,21,52], and can be isolated based on expression of IA6[53]. While we have not functionally demonstrated that IA6+ cells in *B. diegensis* are lineage restricted (*B. diegensis* is not constantly fertile in lab-reared conditions), they express equivalent pluripotency genes. Thus, upon separation of vessels from bodies in *Botrylloides*, the pluripotency of IA6+ cells may be utilized for somatic regeneration. In contrast, all somatic tissues that are formed during asexual reproduction (palleal budding) in healthy animals are derived from the peribranchial epithelium of the parental zooids and not from blood-borne cells[5,12,17,54].

IA6+ cells are constantly cycling in a healthy colony—as shown by expression of *cyclin b* and *histone 3* (Fig. 1c, Fig. 2), and the fact that G2/M cells express high levels of *ia6* and *pou3* (Fig. 5). Since *ia6+ pou3+* cells are constantly cycling and never exhaust for the lifetime of the colony demonstrates that they are constantly self-renewing. Following transplantation, Edu-labeled *ia6+/pou3+* cells give rise to Edu+*pou3+* cells in the regenerating recipient (Supplementary Fig. 3C, pink arrows) that are derived from self-renewing IA6+ cells that were transplanted.

In a large-scale gene expression study, Zondag et al. showed that Wnt and Notch-signaling components are upregulated during early stages of regeneration in *B. leachii*[9]. Here, we show that these pathways are required for proliferation and possibly maintenance of IA6+ Candidate stem cells. Both of these pathways have been shown to play roles in regulating regeneration and stem cell proliferation in several other regenerating species[2], and Wnt signaling is required for blastema formation[42] and stem cell maintenance[41]. Notch regulates blastema cell proliferation during zebrafish fin regeneration, and it mediates cell fate decisions such as proliferation and differentiation in many stem cell types[39]. Notch also regulates blastema formation during distal regeneration in the solitary ascidian *Ciona*[38]. Future studies will investigate which Notch and Wnt ligands are expressed by the regeneration niche in *Botrylloides*, and how and when these signaling pathways act on IA6+ Candidate stem cells to regulate proliferation and differentiation.

In summary, we have identified a candidate stem cell population responsible for WBR in a chordate species, *B. diegensis*, and developed the toolsets necessary for future detailed molecular analysis of the stem cells involved in WBR and the pathways that regulate their function.

## Methods

**Animals.** *B. diegensis* colonies used in this study were collected in Santa Barbara, CA and allowed to attach to glass slides. Colonies were maintained in continuously flowing seawater at 19–23 °C in the dark and cleaned with soft brushes every 14 days. Animals were fed daily with live algae. Growing colonies were subcloned onto independent slides, until a large number of genetically identical individual colonies was established. These genotypes propagate in mariculture for many months. Only healthy, well growing colonies with healthy, expanded vasculature and ampullae and transparent tunic were used for WBR experiments.

**Whole body regeneration.** Only healthy, well growing colonies with healthy vasculature and transparent tunic were used for WBR experiments. Under a stereo dissection microscope, blood vessel fragments were surgically separated from the rest of the colony using razor blades. For time point experiments, all time points were collected from subclones of the same genotype, and at least three different genotypes were used to generate averages. Regenerating fragments were kept attached to slides in flowing filtered seawater at 19–23 °C.

**Integrin-alpha 6 and cell cycle flow cytometry.** Genetically identical, stage matched samples were pooled, and blood was isolated by cutting blood vessels with a razor blade and gentle squeezing with the smooth side of a syringe plunger. Blood was isolated with filtered seawater and passed through 70 and 40 μm cell strainers. Anti-Human/Mouse-CD49f–eFluor450 (Ebioscience, San Diego,CA, USA, clone-GoH3) was added at a dilution of 1/50 and incubated on ice for 30 min and washed with filtered seawater. *B. diegensis integrin-alpha-6* mRNA sequence is listed in Supplementary Data 2, and alignment of human Integrin-alpha-6 protein with translated *B. diegensis* integrin alpha 6 mRNA is shown in Supplementary Methods). For cell cycle sorting, Vybrant Dye Cycle Ruby Stain (Thermo Fisher Scientific, Waltham, MA, USA) was added at a dilution of 1/100 and incubated for 30 min at room temperature. Fluorescence activated cell sorting (FACS) was performed using a FACSAria (BD Biosciences, San Jose, CA, USA) cell sorter. Live cells (P1) were gated based on forward scatter (FSC) and side scatter (SSC) properties (plots shown in Supplementary Fig. 5). Doublets were excluded by plotting FSC-signal height (FSC-H) against FSC-signal width (FSC-W) (plots shown in Supplementary Fig. 5). The violet channel (Excitation 405, filter set 450/50) was chosen for staining of Integrin-alpha-6 because of very low autofluorescence (natural fluorescent background), as shown in the unstained samples (natural fluorescent background) in Supplementary Fig. 5. IA6-negative cells were gated based on isotype-control staining (RatIgG2A-isotype-control eFluor450, Ebioscience, San Diego,CA, USA). IA6+ cells comprised 9.7% of total cells in the sample stained with Anti-Integrin-alpha-6-violet. DNA content was analyzed on a linear scale using the Dye Cycle Ruby signal (excitation 633, filter set 660/20). G2/M cells have double the fluorescence intensity (DNA content) than G0 cells on a linear scale. Analysis was performed using BD FACSDiva 8.0.1 (BD Biosciences, San Jose, CA, USA). Cells were sorted using high-pressure settings and a 70 μm nozzle and collected into filtered seawater. The full sorting strategy is shown in Supplementary Fig. 5.

**Quantitative RT PCR.** Sorted cells were pelleted at 700 g for 10 min, and RNA was extracted using the Nucleospin RNA XS kit (Macherey Nagel, Bethlehem, PA, USA), which included a DNAse treatment step. RNA was reverse transcribed into cDNA using random primers (Life Technologies, Carlsbad, CA, USA) and Superscript IV Reverse Transcriptase (Life Technologies, Carlsbad, CA, USA). Quantitative RT-PCR (Q-PCR) was performed using a LightCycler 480 II (Roche Life Science, Penzberg, Germany) and LightCycler DNA Master SYBR Green I detection (Roche, Penzberg, Germany) according to the manufacturers instructions. The thermocycling profile was 5 min at 95, followed by 40 cycles of 95 °C for 10 s, 60 °C for 10 s. The specificity of each primer pair was determined by BLAST analysis, by melting curve analysis (LightCycler® 480 Software, Version 1.5) and gel electrophoresis of the PCR product. Primer sequences are listed in Supplementary Data 5. Relative gene expression analysis was performed using the $2^{-\Delta\Delta CT}$ Method. The CT (threshold cycle) of the target gene was calculated using LightCycler® 480 Software, Version 1.5 and normalized to the CT of the reference gene *actin*: $\Delta CT =$ CT (target) − CT (actin). The expression ratio was calculated: $2^{-\Delta CT} =$ Expression ratio normalized to actin (Fold change). To calculate the expression ratio between cell populations, the $\Delta CT$ of the test sample (e.g. IA6-positive cells) was first normalized to the $\Delta CT$ of the calibrator sample (e.g.IA6-negative cells): $\Delta\Delta CT = \Delta CT$(IA6-positive)- $\Delta CT$(IA6-negative). Second, the expression ratio was calculated: $2^{-\Delta\Delta CT} =$ Normalized expression ratio. The result obtained is the change of the target gene in the test samples relative to the calibrator sample. Each qPCR was performed at least three times on cells from independent sorting experiments performed on three different genotypes. Each gene was analyzed in duplicate in

each run. The expression ratio (fold change) was first calculated for each replicate and then averaged across replicates. For time point experiments, all time points were collected from subclones of the same genotype, and at least three different genotypes were used to calculate averages.

**Mitomycin C treatment and rescue.** Vessel fragments were cut and soaked in 60 μM Mitomycin C (Tocris, Bristol, UK) in filtered seawater for 24 h. The same dose of MMC was used in a previous study on *Botrylloides*[28]. We tested this dose on our vessel fragments. The vessels continued to have blood flow and were arrested in stage 2, and remained that way for many weeks after removal of the drug. MMC was removed and fragments were returned to flowing seawater for 24 h before being micro-injected with cells (different numbers of IA6+, IA6−, G0 or G2M, as indicated) isolated from the blood of normal, healthy colonies. Sorted cells were collected into 3 ml of filtered seawater, pelleted by centrifugation (700 g, 10 min) and resuspended in 100 μl filtered seawater. Cells were counted using a hemocytometer. Only round, bright, intact cells were counted. Cells were labeled with Hoechst 33342 and diluted in filtered seawater to the appropriate concentration. 0.1 μl volume containing either 1000, 50, 10, 5, or 3 cells was injected per sample. For injections of 10 cells or less, 1 μl cell suspension stained with Hoechst 33342 was pipetted onto a microscope slide and cells were counted under brightfield and fluorescence to verify the correct cell concentration. For single cell injections, cells were diluted so that 1 μl contained 5 cells, and 0.1 μl was injected per sample = 0.5 cells per sample. Cells were injected into ampullae of MMC-treated vessel fragments with pulled glass capillary needles and an Eppendorf FemtoJet microinjector under a stereomicroscope. Rescue efficiency (number of fragments reaching stage 5) was scored after 20 days. All recipients in each experiment were subclones from the same colony, transplanted with cells derived from a healthy subclone not treated with MMC. Each independent experiment was performed on a different day with freshly sorted cells, and subclones from a different genotype than in the previous experiment were used. The frequency of IA6+ cells capable of rescuing was calculated using ELDA analysis software (http://bioinf.wehi.edu.au/software/elda/[36]).

**FISH on cryosections.** Vessel fragments at different stages of regeneration were fixed overnight in 4% paraformaldehyde in PBS at room temperature. Samples were washed in PBS and soaked in 15 and 30% sucrose for 30 min each before embedding in OCT medium. 20 μm sections were cut using a Leica cryostat. FISH was adapted from[34] for cryosections. Briefly, *B. diegensis* homologs of genes of interest were identified by tblastn searches of our own B. diegensis transcriptome using human or *Ciona* (when available) protein sequences. Primer pairs were designed to amplify a 500–900 bp fragment of each transcript (Primer and probe sequences are listed in Supplementary Data 5. *B. diegensis integrin-alpha-6* mRNA sequence is listed in Supplementary Data 2. *B. diegensis pou3* mRNA sequence is listed in Supplementary Data 1.). PCR was performed with Hotstar DNA Polymerase (Qiagen Germantown, MD 20874) and products were cloned into the pGEM-T Easy vector (Promega, Madison, WI, A1360). Sense probes were used as negative controls. All gene fragments cloned for FISH probes were confirmed by sequencing. In vitro transcription was performed with SP6 or T7 RNA polymerase (Roche, Penzberg, Germany 10810274001, 10881767001) using either digoxigenin, fluorescein, or dinitrophenol labeling. Cryosections were air-dried and fixed with 4% PFA for 10 min and washed with PBS/1%Triton-X-100. Probes were diluted in hybridization buffer and hybridized at 65 C for 30 min. Probes were removed and slides washed with 2xSSC/1%Triton and 0.2xSSC/1%Triton for 15 min each at 65 C. HRP-conjugated anti-digoxigenin antibody (Sigma-Aldrich 11207733910 Roche) 1/500, HRP-conjugated anti-fluorescein antibody (Sigma-Aldrich 11426346910 Roche) 1/500 or unconjugated anti-DNP antibody (Vector labs SP-0603-1) 1/100 followed by anti-rabbit HRP (Goat Anti-Rabbit IgG H&L (HRP) (ab6721) Abcam) 1/500 were used to detect labeled probes. Fluorophore deposition was performed by incubating slides for 20 min at RT in Tyramides (Alexa488-Tyramide, Alexa-555 Tyramide or Alexa-594-Tyramide, all from Thermo Fisher) diluted 1/100 in PBS with 0.001% hydrogen peroxide. Slides were washed twice with PBS and nuclei were stained with Hoechst 33342 (Life Technologies). Imaging of labeled samples was performed using an Olympus FLV1000S Spectral Laser Scanning Confocal. Image processing and analysis was performed using FIJI (Fiji.sc). Quantification: images from four independent samples (three different genotypes) per time point were taken with a ×20 objective and *ia6/h3* double-positive cells, *ia6* single-positive cells and *h3*-positive cells were counted using the cell counter feature in FIJI. Counts were normalized to the number of nuclei for each image. Graphs represent cell counts in percent of nuclei, averaged over four independent samples per time point. Between 2000 and 3000 cells (Hoechst-positive nuclei) were counted for each time point.

**Immunofluorescence on cryosections.** Cryosections from regenerating and healthy vessels were prepared as for FISH (see above). Cryosections were air-dried and fixed with 4% PFA for 10 min and washed with PBS/1%Triton-X-100. Anti-Integrin-alpha 6 (avian, P2C62C4, DSHB) 1/50, AntiPou3F2 (PCRP-POU3F2-1A3, DSHB) 1/50 or anti-histone h3 phospho S10 (Abcam ab47297) 1/100 were diluted in PBS/1%Triton-X-100 + 2%BSA and incubated overnight at room temperature. After washing three times with PBS, secondary antibodies anti-mouse HRP

(Abcam Goat Anti-Mouse IgG H&L HRP ab6789) or anti-rabbit HRP (Goat Anti-Rabbit IgG H&L (HRP) (ab6721) Abcam) were diluted 1/500 in PBS/1%Triton-X-100 + 2%BSA and incubated for 1 h at room temperature. Fluorophore deposition was performed by incubating slides for 20 min at RT in Tyramides (Alexa488-Tyramide, Alexa-555 Tyramide or Alexa-594-Tyramide, all from Thermo Fisher) diluted 1/100 in PBS with 0.001% hydrogen peroxide.

**Tracking of EdU-labeled cells**. For every ten zooids, 2 µl of 1 mM EdU (Thermo Fisher) dissolved in filtered seawater was injected into the blood stream of healthy colonies. After 24 h, Integrin-alpha-6-positive cells were isolated from EdU-injected colonies by flow cytometry and 1000 IA6+ cells were injected into recipient vessel fragments 24 h after Mitomycin treatment (see above). The experiment was performed two times. At stage 3 ($n = 10$) and stage 4 ($n = 10$) of regeneration, injected samples were fixed and cryosections were prepared as described above. Sections were fixed with 4% PFA for 30 min and treated with proteinase K for 7 min. Sections were post-fixed with 4% PFA for 20 min and blocked with PBS/1% Triton and 3% BSA for 1 h. The clickit-reaction cocktail was prepared according to the manufacturer's instructions and the reaction was stopped after 1 h at room temperature. Nuclei were stained with Hoechst 33342. Regenerating samples injected with EdU were used as positive control, and uninjected regenerating samples were used as negative controls. Imaging of labeled samples was performed using an Olympus FLV1000S Spectral Laser Scanning Confocal. Image analysis was performed using FIJI software (Fiji.sc).

**Small molecule inhibitor treatment**. Vessel fragments were cut and placed in the bottom of 24 well plates. 1 ml of filtered seawater containing the Notch-signaling inhibitor DAPT (Tocris, 0.2–2 µM), the Wnt-signaling inhibitor endo-IWR1 (Tocris, 0.1–1 µM) or control drugs Exo-IWR (1uM) and the Notch-sparing gamma-secretase modulator E2012 (1 µM) were added to filtered seawater and replaced every other day. The number of colonies that reached stage 4 was counted at 14 days post injury ($n = 16$ for each condition). Vehicle-treated controls were subclones from the same colonies as the experimental samples.

**Pou-phylogenetic analysis**. Protein sequences used for phylogenetic analysis were downloaded from the NCBI database or determined in this study (sequences included in Supplementary Data 6). A sequence of 1376 base pairs of a *B. diegensis* pou3 mRNA was sequenced by Sanger sequencing (Sequence listed in Supplementary Data 1). This translates to an open reading frame of 457 amino acids. POU protein sequences were aligned using the ClustalW algorithm in the MEGA7 application[55]. Phylogenetic analysis of POU family members was performed with the software RAxML version 8.0.0 using a maximum likelihood method, the JTT substitution matrix, and empirical frequencies[56]. RAxML software was accessed using the CIPRES Science Gateway (Creating the CIPRES science gateway for inference of large phylogenetic trees. In: Proceedings of the gateway computing environments workshop (GCE), New Orleans, LA, USA; 2010. p. 1–8.) and trees were visualized using the Interactive Tree of Life website (Letunic I and Bork P (2006) Bioinformatics 23(1):127-8 Interactive Tree Of Life (iTOL): an online tool for phylogenetic tree display and annotation).

**Reporting summary**. Further information on research design is available in the Nature Research Reporting Summary linked to this article.

## Data availability
The authors declare that all data supporting the findings of this study are available within the article and its Supplementary Information/data files or from the corresponding author upon reasonable request. Source data are provided with this paper.

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

## Acknowledgements

The authors would like to thank Bill Smith for providing facilities to grow *Botrylloides* colonies. We also thank Ben Lopez and the NRI-MCDB microscopy facility at UCSB. Shane Nourizadeh, Jessa Alcaide, and Phillip Ahn are acknowledged for help with cloning. Delany Rodriguez and William Jeffery are acknowledged for helpful discussions and critical input. *Eunice Kennedy Shriver* National Institute of Child Health and Human Development (NICHD) HD092833 to AWD and SHK.

## Author contributions

S.H.K.: study design, data acquisition and interpretation, funding acquisition, manuscript writing, final approval of manuscript. A.D.L.: data acquisition and interpretation, final approval of manuscript. A.W.D.: study design, data interpretation, funding acquisition, manuscript writing, final approval of manuscript.

## Competing interests

The authors declare no competing interests.
