## [Peer Review File · Nature Communications]

Reviewers' comments:

Reviewer #1 (Remarks to the Author):

This manuscript identifies a population of blood-borne cells responsible for whole body regeneration (WBR) in the colonial ascidian *Botrylloides diegensis*. These cells were found to co-express: (i) integrin- α -6 (IA6, also known as CD49f), a transmembrane receptor expressed in many types of cancer stem cells, (ii) *pou3*, a transcription factor of the Pou 5 gene family closely related to mammalian reprogramming factor Oct4, and (iii) *vasa*, an ATP dependent RNA helicase expressed in the germline of many animals. The authors show that one-single IA6 cell can rescue WBR in vascular fragments treated with the anti-tumor Mitomycin C (MMC), in which stem cells and other cells in proliferation have been presumably depleted. Injected IA6 cells proliferate, migrate, and home into regeneration foci contributing to early precursor cells of WBR. These results demonstrate the stem cell potential (i.e. the capacity to self renew and to differentiate into multiple cell types) of IA6 cells. Last, the authors report the up-regulation of expression of two important signaling pathways known to be involved in cell proliferation and regeneration during early stages of WBR. Finally, the authors conclude that the IA6 blood borne cells represent the pluripotent hemoblast population of colonial botryllid ascidians, functionally in close resemblance to neoblasts of planarians, i-cells of hydra, or archeocytes of sponges. Although not entirely unexpected, the degree of conservation in gene expression of colonial ascidian stem cells and mammalian stem cells contributes significantly to our understanding of the molecular and cellular mechanisms of blood-borne stem cells required to activate WBR, and highlights important implications about the evolution of stem cells in humans.

This study provides answers to long-standing questions about the nature of hemoblasts that have remained unanswered and extensively debated in the tunicate community for decades. Thanks to the progress in cancer and stem cell research, the authors have selectively isolated stem cells in the blood of colonial ascidians using CD49f, a well-known marker for tumor cells, and have used MMC, a chemotherapy agent, to deplete the stem cells in vascular fragments of the colonies. Because these approaches are novel and interesting, the experiments need to be rigorously tested, and experimental optimizations provided and well-presented. I found several instances in the manuscript, where additional evidence is, in my opinion, required. I was also disappointed with the last section of the manuscript, which explores the involvement of Wnt and Notch during WBR, because the qPCR expression patterns shown only leave speculative conclusions and raise questions about where they are expressed, and the extent of their involvement in WBR. Besides, the expression of these two pathways during early WBR were already published for another *Botrylloides* species, as noted by the authors themselves. However, this data only represents secondary results, and at this point are irrelevant to the main findings of the study i.e. the molecular characterization of IA6 cells with the functional rescue of these cells to the MMC treated colony fragments.

Major concerns:

One of the main characterizations of IA6 cells is that they express markers *pou3* and *vasa*, but *vasa* has not been temporally characterized during WBR, as IA6 and *pou3* have. It is difficult to evaluate the whole picture of IA6 gene expression dynamics during WBR if the complete expression patterns are not provided. For example, the authors do not show the complete co-expression results of *pou3+vasa* and *pou3+IA6* during all stages of WBR. Additional controls are also necessary. Consider presenting one last timepoint (stage 5) with the expression of these genes at the end of WBR (full differentiation), as well as a general characterization of the expression of these genes in the blood of whole colony (as in Fig. 1A).

One suggestion is to organize your co-expression data into two figures, one that shows the dynamics of co-expression of stem cell markers in the blood cells at different timepoints, and another figure showing the proliferation dynamics of the cells expressing these markers + histone. For example, you could present one figure with only co-expression of stem cell markers (i.e. all different combinations of double FISH of pou3, vasa, and IF6; not histone) during all WBR stages together with the cell counts of these double labeled experiments into one figure; and another figure could include the co-expression of each of your markers with histone (i.e. all double FISH of pou3, vasa, and IF6 with histone) during all WBR stages along with the cell counts of these double labeled experiments. You have presented all the data for the latter (in Figures 2 and S2), but need to provide the complete dataset for the former (as you only show pou3+vasa and pou3+ia6 on stage 0 in Fig. 1 D and 1E). All single labeled photos and single labeled cell counts for both figures can be placed as supplementary material.

I also found inconsistencies in the presentation of the data when comparing the labels of your photos to the labels in the quantifications (i.e. graphs of cell counts), for example in Fig. 1E you use a red label for pou3 and green for ia6 in the photo, but use red for ia6 and green for pou3 in the graph. Figure 2 also shows inconsistencies in the selection of color labels in the photos and in the cell count graphs. Please be consistent with the color labels.

Another major concern I have is regarding the optimization of the depletion of proliferating cells using Mitomycin C (MMC). Can you show the survival curves and cell depletion experiments to show that the dose you are using (60mM) has been adjusted to botryllid ascidians? These experiments should show the survival curves of the animals (or at least the fragments) with the proper evidence to show that cells have stopped dividing. A simple way to evaluate proliferating cell depletion under different doses of MMC would be to show the M2/G vs G0 FACS profiles under the different doses. In your figure S3A, you only show depletion of pou3/h3 cells 5 days after MMC treatment? A dose response curve showing the depletion would be preferred. As you know MMC at high doses may not only disrupt the proliferating cells, but MMC also disrupts transcription so high levels may affect the overall homeostasis of tissues and become toxic.

I consider that the limiting dilution analyses of the IA6 cell injections in MMC treated fragments represent the most important and exciting results of this study, but have been poorly documented. You only evaluate the proportion of successful events of WBR after 20 days in the Table of Figure 3C, but do not show a more careful analysis of what happened under the different rescue experiments. Did you find that zooids fully differentiated at a later timepoint when lower numbers of IF6+ cells were injected? Did lower numbers of cells injected translate into less numbers of regeneration foci? Did you observe arrested buds that regressed under any of the circumstances? Can you show or describe more carefully what you expected and what you observed under each cell injection scenario? Do you have the evidence that shows that the IF6 cells you injected are also the EdU cells that were tracked? What is the temporal and spatial dynamics of the injected IF6 cells when in low or high numbers in the rescue? Do they few foci and expand? Do they expand first and form foci later? There are many questions to be answered, and important information that can be documented by doing a more careful analyses of the rescue experiments.

How many cells were injected when you used EdU? Could you recover this EdU signal if one single or few IF6A cell were injected? And when many cells are injected? Were the injected EdU cells equally distributed among all foci, or did some foci show bmore EdU cells than others? How far in time (stage?) is the EdU signal traceable during WBR? During organogenesis? I think these experiments could potentially also be explored in more detail.

To my knowledge this is the first time that DAPT and Endo-IWR are used in colonial ascidians as

inhibitors, and therefore the procedures and doses should be well-documented. Can you include some tests where you tested different doses and windows of sensitivity of the inhibitors on untreated whole colonies? What were the dose responses observed on blastogenesis? Documenting effects on blastogenesis could presumably allow you to observe the specific dose that is required to disrupt distinct processes of development.

The results of notch and delta inhibitors show that they affect the final outcome of WBR, as well as formation of regeneration foci. It would be interesting to show either in the main text or as supplementary information more detailed descriptions of the effects at different timepoints if not by histology at least externally: how many foci were observed? What were the final numbers of buds and zooids that differentiated in each replicate and for each treatment? How did these inhibitors affect the timing of each stage of WBR? When the inhibition was reversed, did the different times at which the inhibitors were removed affect the time of recovery of WBR? If these data cannot be included in the main the text due to word number constraints, it should be included in organized Tables as Supplementary Material.

The disruption of cell aggregates and regeneration foci with the inhibition of wnt is interesting, but somewhat surprising for notch signaling as this pathway requires cell-cell contact for activation. To test the hypothesis that notch signaling disrupts regeneration foci, you would expect in control that notch is activated either before or at the very early contact of cells when forming aggregates, therefore it would be interesting to show a control stage 3 untreated fragment presumably with notch expression in those early aggregates, but unfortunately your data only shows WBR fragments until stage 2.

A section needs to be included in the discussion that mentions and discusses possibilities on why radioprotection assays failed in previous *Botryllus schlosseri* experiments (see Laird and Weissman, 2004; and Rinkevich and Weissman, 1990). The results on these studies are somewhat different from those observed here in *B. diegenensis*. For example, in *B. schlosseri* Laird injected up to 80000 cells into radiated colonies and could not find any rescue, and also Rinkevich observed that radiated colony parts could not be reconstituted even after non-irradiated isogenic colonies were fused. How do these apparently opposite results in these two closely related species be explained? Do the authors think different blood cells occur in *B. schlosseri* and *B. diegenensis*? I would encourage the authors to briefly mention these discrepancies and discuss some possible implications of their findings to explain these differences across species.

Minor corrections:

Line 29: Delete 'phyla of'

Line 66: Cite Zondag et al. (2016) here. Even though the stages for *B. diegenensis* may have been modified and adjusted from those stages in *B. leachii*, the main events of WBR stages used are very similar to those proposed by Zondag et al. (2016).

Lines 66-80: Please, include the range of time frames in the paragraph when describing each stage. Thereafter, include both TIME and STAGE when describing your data. There are some instance where you mention only the time and other instances where you only mention the stage, for example in Figure 2 you only mention stages but not time. Because some stages overlap in timeframes, I think both should be used every time, as in stage 5 (day 20) or st. 3 (72h).

Line 86: In fact, four species: *B. primigenus*, *B. schlosseri*, *B. violaceus*, and *B. leachii* [Oka and

Watanabe (1957, 1959), Voskoboynik et al. (2007), Rinkevich (2007)]

Line 88: Correct citation, change Oka, 1959 for Brown et al. (2009a).

Lines 99-101: Include the following citations: Brown et al. (2009b), and your own Langenbacher and De Tomaso (2016).

Lines 133-135: You need to include here that pou3 was previously found expressed with relatively high expression in few scattered cells in the inner vesicle of the early bud during blastogenesis in *Botryllus schlosseri* (Ricci et al. 2016)

Line 138: Why did you normalize your gene expression to vasa? Wouldn't you want a gene that is not affected by your treatments? Vasa is selectively expressed in some cell types and it is known to cycle and change expression levels during blastogenesis.

Line 169: Consider revising the use of 'steady state' to simply define similar proportions of cells. To me the use of 'steady state' has broader implications including an overall maintenance of cell proportions and some sort of homeostasis in the whole system.

Lines 171-174: Correct and change sentence. I do not see 64% quiescent Pou3+ cells at stage 3, nor 71% of cell proliferation in pou3- double vesicles at stage 3 in Figure S2. Where are these cell counts shown? I do not see a single yellow arrow in Figure S2? I only see yellow arrow in Figure 2.

Lines 175-177: Consider softening your concluding sentence here, as you have not shown carefully the spatial and temporal dynamics of double labeled ia6/pou3 cells throughout WBR.

Lines 183-184: In figure S3A, why do you evaluate depletion of pou3 cells, and not IA6 cells directly? To also show that all proliferating cells have been depleted, but the MMC treatment did not show toxicity stronger evidence is required. The ideal experiment would be to show by FACS that G2/M cells have been depleted, or perhaps you can show that the 60nM MMC dose you used also disrupts blastogenesis in complete colony, but does not affect overall homeostasis of the tissues. However, it would be better to show the dose dependence curves requested above.

Lines 186-189: So the MMC treated fragment form typical aggregates of stage 2, and then arrest? It worries me that you mention that '...most MMC treated blood vessels...' do not reach stage 4 or 5, because this could mean that you are not using the optimal dose in your experimental assay...

Line 190: here you mention that you injected '...2000 total blood cells...' but in Figure 3B you state that you injected IA6+ cells. Which is it?

Line 199: In Figure 1E, you show that 9-10%??? of the cells that are pou6+ are IA6-, not 20%.

Line 226: In Figure S3B, where are the white arrows? I only see green arrows.

Lines 256-259: Can you show downregulation of these in later stages of WBR? What are the base levels of expression in a colony?

Lines 278-280: Why do you check Notch inhibitor effects on pou3 cells, and not on IGF6?

Line 286: When were the inhibitors removed? Did the timing of inhibitor removal affect the recovery process and WBR?

Line 299: Year is missing in a reference.

Line 307: Vasa and Piwi have also been previously associated to somatic functions as well, see Alié et al. (2011), Gonzalez et al. (2015), and Poon et al. (2016). These studies could be cited here as well as other cases where these markers are expressed in stem cells, aside from the germline or germ cells.

Line 343: The author states "...all somatic tissues that are formed during asexual reproduction (palleal budding) in healthy animals are derived from the peribranchial epithelium...". I am not sure we know this. To my knowledge, no-one has been able to exclude the contribution of blood borne stem cells in palleal budding. I agree that the initial stages up to the double vesicle are derived from the peribranchial epithelia, but after that it remains uncertain.

Figure 1D: Please do not use e charts, you can use a simple bar graph instead. Also the colors of the labels are not consistent between the photos and the graphs.

Figure 2: Colors of labels between photos and graphs are inconsistent. Why was stage 1 not included? Include the times sampled for each stage.

Figure 3C should be a separate Table and not inserted as C in the figure. This Table is the most important result of this study, but it is not well organized or presented (for example, rates and percentages can be placed in the same box, the names of categories in the left column could be improved to facilitate understanding, include subtotals and totals, explain what experiment 1 and 2 mean (are these replicates different in any way? Were they done in different days or the same days in parallel?)

Figure 5: Where are your negative controls? Do you have some markers of differentiated cells that you can use, or genes that you expect not to be up-regulated?

Figure 6B should also be placed as a Table and not embedded in the figure, perhaps with a better documentation of the different timepoints? One would expect a lower recovery with longer inhibitor incubations.

Figure S2 contains expression data of pou3, one of the main markers (if not the main marker) of stem cells that is proposed to be co-expressed with IA6 cells. Shouldn't this data be in the main text and not in supplementary? Also why are these two markers not shown together in a double FISH during the different timepoints of WBR? Consider moving Fig. S2A to Figure 2, as well as the complete cell counts of the double and single FISH. Then, the micrographs that show h3 and IA6 independently in single color channels (middle and lower row of Fig 2) could be moved to the supplementary material.

Figure S3: Do you have a double FISH with IA6 and h3, instead of pou3 and h3? Where are the white arrows? Do you mean green arrows?

Additional references:

Alié A, Leclère L, Jager M, Dayraud C, Chang P, Le Guyader H, Quéinnec E, Manuel M. Somatic stem cells express Piwi and Vasa genes in an adult ctenophore:

ancient association of "germline genes" with stemness. *Dev Biol.* 2011 Feb 1;350(1):183-97. doi: 10.1016/j.ydbio.2010.10.019. Epub 2010 Oct 29. PubMed PMID: 21036163.

Gonzalez, J., Qi, H., Liu, N., & Lin, H. (2015). Piwi Is a Key Regulator of Both Somatic and Germline Stem Cells in the *Drosophila* Testis. *Cell reports*, 12(1), 150–161. doi:10.1016/j.celrep.2015.06.004

Laird DJ, Weissman IL. Continuous development precludes radioprotection in a colonial ascidian. *Dev Comp Immunol.* 2004 Mar;28(3):201-9. PubMed PMID: 14642887.

Oka, Hidemiti, and Hiroshi Watanabe. "Vascular Budding, a New Type of Budding in *Botryllus*." *Biological Bulletin* 112, no. 2 (1957): 225-40. Accessed February 4, 2020. doi:10.2307/1539200.

Poon J, Wessel GM, Yajima M. An unregulated regulator: *Vasa* expression in the development of somatic cells and in tumorigenesis. *Dev Biol.* 2016 Jul 1;415(1):24-32. doi: 10.1016/j.ydbio.2016.05.012. Epub 2016 May 11. Review. PubMed PMID: 27179696; PubMed Central PMCID: PMC4902722.

Ricci, L., Chaurasia, A., Lapébie, P., Dru, P., Helm, R. R., Copley, R. R., & Tiozzo, S. (2016). Identification of differentially expressed genes from multipotent epithelia at the onset of an asexual development. *Scientific reports*, 6, 27357. doi:10.1038/srep27357

Rinkevich B, Weissman IL. *Botryllus schlosseri* (Tunicata) whole colony irradiation: do senescent zooid resorption and immunological resorption involve similar recognition events? *J Exp Zool.* 1990 Feb;253(2):189-201. PubMed PMID: 2313247.

Rinkevich Y, Paz G, Rinkevich B, Reshef R. Systemic bud induction and retinoic acid signaling underlie whole body regeneration in the urochordate *Botrylloides leachi*. *PLoS Biol.* 2007 Apr;5(4):e71. PubMed PMID: 17341137; PubMed Central PMCID: PMC1808485.

Voskoboynik A, Simon-Blecher N, Soen Y, Rinkevich B, De Tomaso AW, Ishizuka KJ, Weissman IL. Striving for normality: whole body regeneration through a series of abnormal generations. *FASEB J.* 2007 May;21(7):1335-44. Epub 2007 Feb 8. PubMed PMID: 17289924.

Reviewed by,
Federico D. Brown

Reviewer #2 (Remarks to the Author):

Some colonial ascidians can undergo whole body regeneration – that is, reconstitute an entire individual from blood vessels and stem cells. This manuscript uses colonies treated with Mitomycin C (MMC) where cell division is inhibited so that the colony cannot regenerate in order to isolate the stem cell population that is necessary for whole body regeneration. Cell sorting of blood cells revealed a specific type of integrin alpha-6 positive cell was the key to allowing whole body regeneration. Edu labeling subsequently showed that the regenerated tissues were generated from the integrin alpha-6

positive cells. These are extraordinary results, isolating a specific type of circulating stem cell that is required for whole body regeneration in colonial ascidians. These are exciting and novel and give insight into the circulating stem cells in colonial ascidians.

In the second paragraph of the Introduction, there are no references for previous papers on Whole Body Regeneration (WBR) in *Botrylloides* species. There have been some important and insightful papers written on this process and those should be referenced within (for ex. Blanchoud et al. 2018 (*Botrylloides leachei*) Brown et al. 2009 (*Botrylloides violaceus*), etc.). Paragraph 3 in the Introduction also needs references, describing the life cycle and bud formation in *Botrylloides diegensis*.

The results reported here also give insight into the normal process of whole body regeneration in *B. diegensis*. It appears that the integrin alpha-6 positive cells aggregate at the site of WBR, and then the blood vessels close around the clump of cells. It is then shown that the integrin alpha-6 positive cells are responsible for dividing to undergo whole body regeneration and these processes are regulated by Notch and Wnt signaling. These are inhibitor studies, in which results are made stronger by the fact that when the inhibitors are washed out, the integrin alpha-6 positive cells begin to proliferate and undergo whole body regeneration. This shows that Notch and Wnt signaling are necessary to upregulate proliferation of the cells during normal regeneration.

Reviewer #3 (Remarks to the Author):

In this manuscript the authors studied whole body regeneration (WBR) in a wild type *Botrylloides diegensis* species. They used antibodies developed against human/mice integrin alpha-6 (ITGA6) and flow cytometry to isolate a population of cells and perform diverse transplantation experiments to measure their ability to proliferate. They designed probes and primers to detect several genes based on their sequence in other botryllid species (*Botryllus schlosseri* and *Botrylloides leachii*), and used them to measure levels of expression (qPCR) and sites of expression (ISH) of these genes in *Botrylloides diegensis*. By combining EDU tracing experiments and ISH they found that the sorted cell populations (cells stained by the mammalian antibodies of integrin alpha-6) include a highly proliferate population of small cells that are also positive to the *B. schlosseri* and *Botrylloides leachii* probes aimed to detect integrin alpha-6 (ITGA6), *pou3* and *vasa* (DDX4). By treating vasculature fragments taken from *Botrylloides diegensis* (undefined size, undefined stage) with mitomycin C they prevented whole body regeneration and then succeeded in inducing whole body regeneration by transplanting thousands of cells from non-treated individuals, 50 ITGA6+ sorted cells, and even 1 ITGA6+ cells. FACS plots that present the expression of this antibody suggest that at least 90% of the *Botrylloides diegensis* blood cells express this protein in high levels (above 10² Figure S1 C).

Developing new model organisms to study regeneration is important and can add to our basic knowledge about cellular and molecular mechanisms of regeneration. The authors performed an impressive set of experiments and tests aiming to study mechanisms of regeneration in a colonial chordate species for which very little information is available about its life cycle and biology. However, several major issues need to be addressed before one can assess the real impact of this study and publish this paper.

1. Stem cells are self-renewing and multipotent cells. This study does not include experiments that prove self-renewal or multipotency and therefore the use of the term stem cells throughout the paper is inappropriate. When discussing previous work on candidate stem cells in diverse botryllid species, the authors should distinguish between papers that indeed found and verified stem cells (e.g. Laird et al. 2005) versus papers that only point to candidate stem cell populations (e.g Brown et al. 2009b). To prevent confusion, studies that focus on expression of genes associated with stemness without experiments that clearly demonstrate self-renewal and multipotency should not be discussed and reviewed as studies that succeeded in isolating stem cells.

2. The authors present a novel model system but do not include basic information regarding its biology, life cycle, regeneration capacities, and natural fluorescent background under normal and stress conditions, information that is crucial to assess the outcomes of the experiments (mainly experiments that induce stress and outcomes that are detected by fluorescent product).

The authors use *Botrylloides diegensis* colonies collected in the wild; thus age, individuality (naïve/chimeras) and history are unknown and cannot be determined. This is adding unmeasurable variables to the study and needs to be included in consideration when designing the experiments (high numbers of repeats) and when drawing conclusions.

3. The authors use an array of antibodies, small inhibitors, probes and primers developed against mammalians or other botryllid organism. This is a major caveat and without verifying that each of these reagents indeed detects or blocks the gene / protein it was originally designed against, conclusions should not be drawn.

4. The results and methods sections are written in a sketchy and vague way limiting the ability of other researchers to reproduce this work in the future. Important information regarding transplantation experiments and FACS protocols are lacking.

No information is provided regarding natural fluorescent background under normal and stress conditions. No information is provided regarding the size of fragments used for WBR and the stage of development tested. This information is crucial since it's been shown that WBR is dependent on fragment size and stage in botryllid ascidians (e.g. Rinkevich et al. 1995 PNAS; Voskoboynik et al. 2007 FASEB). Images that follow control and transplanted fragments from day 1 until the end of the experiments need to be presented. No information is provided regarding the transplantation assays, making it very difficult to assess the value of this experiment. Transplantation of a small number of cells or even a single cell is the most difficult experiment in the stem cell field, therefore a detailed description that includes images and videos demonstrating the ability to transplant low number of cells into small fragments of vasculature is imperative. The authors need to include Images that follow data regarding the FACS sorting and the dyes used to label cells before transplantation to verify successful transplantation. The authors need to include methods and measures used to assess viability in the fragments treated and normality of developing buds like histological sections that follow development.

Major points:

1. Stem cells are self-renewing and multipotent cells. The paper does not include experiments that prove self-renewal or multipotency, and therefore the use of the term stem cells throughout the paper is inappropriate.

When discussing works done in the past on candidate stem cells in diverse botryllid species the authors should distinguish between papers that indeed found and validated stem cells and between papers that only point to candidate stem cell populations. Studies that focus on expression of genes associated with stemness without experiments that clearly demonstrate self-renewal and multipotency (transplantation and genotyping) should not be discussed and reviewed as studies that isolated stem cells.

Throughout the paper (introduction, discussion) the authors should include comprehensive work done on *Botryllus schlosseri* chimeras which already prove that stem cell originating in cells circulating in the vasculature are contributing to germline and somatic tissues in the developing buds. Just to name a few: Sabbadin and Zaniolo 1979 J. Exp. Zool; Stoner and Weissman, 1996 PNAS; Stoner et al. 1999 PNAS; Voskoboynik et al. 2008 Cell Stem Cell; Rinkevich et al. 2013 Dev. Cell; Rosental et al. 2018 Nature.

The authors also failed to cite original work describing vascular budding / WBR in botryllid species including: Sabbadin et al. 1975 Dev BioI; Rinkevich et al. 1995 PNAS; Voskoboynik et al. 2007 FASEB. Also, the phrase "whole body regeneration"-WBR- was coined by Rinkevich et al in 1995.

2. The authors present a novel model system but do not include basic information regarding its biology, life cycle, regeneration capacities, and natural fluorescent background under normal and

stress conditions.

The authors should describe in detail the life cycle of *Botrylloides diegensis* under normal and stress conditions: mainly during its blastogenic cycle (budding), How long does it take for buds to complete their development? Is the development of buds synchronized? Do their life cycles include hibernation similar to *Botrylloides leachi*? Do they have natural fluorescent background in the channels used for detection of the diverse assays (EDU; FISH; FACS)?

The authors should describe in detail whole body regeneration (WBR) or vascular budding under normal conditions, clearly define the characteristic of stages 1-5 (The information given in Figure 1 and introduction is very limited), measure the effect of size and developmental stages on the ability of a fragmented vasculature embedded within a tunic to regenerate normal whole bodies. For example- can it be that the bigger fragment size in Fig3 B vs. 3A effected the WBR outcomes? Were all samples taken from the same developmental stage? Which stage? Did the zooids developed were normal?. This information needs to be included and images of control and injected fragments on Day 0 and along the regeneration process should be presented.

Does stress (e.g inhibitors of proliferation or specific genes) enhance fluorescent background? Arrest development? What are the % of fragments that resume regeneration by themselves when the stressor is removed?.

Marine ascidians are known to live in highly contaminated areas (harbors) and present robust regeneration and survival capacities. Understanding the baseline regeneration abilities under random stressors is necessary before one can understand the effect of the diverse treatments and experiments done in this study.

In this work the authors use wild type *Botrylloides diegensis* colonies that did not originate from defined crosses. Using colonies where age or individuality (e.g chimera or naïve) are unknown add unmeasurable variables and noise and compromise the ability to draw conclusions from the experiments performed.

Wild type marine ascidians usually contain high levels of auto-fluorescence. Since FACS and other fluorescent based assays are used in this study, the authors need to describe in detail how they handled the background and include ISH and control EDU images as well as FACS panels of unstained tissues and cells.

3. The authors use an array of antibodies, small inhibitors, probes and primers developed against mammals or other botryllid organism. This is a major caveat and without verifying that each of these reagents indeed detects or blocks the gene / protein it was originally designed against: conclusions should not be drawn in the way presented in this paper.

One of the key gene/proteins discussed in this work is Integrin alpha 6+ (IA6) (found in the *Botryllus schlosseri* genome under the name (ITGA6).

The authors used IA6 antibodies developed against human/ mouse to sort *Botrylloides diegensis* cells without demonstrating that IA6 is expressed on these sorted cells, claiming in both the title and throughout the paper that this population expresses this protein.

Stating in the title "Integrin-alpha-6+ Stem Cells (ISCs) are responsible for whole body regeneration in an invertebrate chordate", is misleading.

The authors need to prove that this human antibody can also recognize *Botrylloides diegensis* IA6. Mass spec or RNA sequencing might be helpful to prove that this protein / gene is expressed on the IA6+ sorted cell population but NOT on the IA6- cell population.

The use of antibodies developed for mice and human is very problematic when working with marine organisms. Firstly, the ancestor they shared diverged millions of years ago, and evolved rapidly. Secondly, KLH (derive from marine organisms) is often used to enhance the production of commercially prepared antibodies for mice and human, which can lead to non-specific staining. The FACS plots that present the expression of this antibody (Figure S1 C) show that 90% of

Botrylloides diegensis blood cells express this protein in high levels (above 102), suggesting a non-specific reaction.

The primers that were used in this study to estimate level of expression of several genes by qPCR (IA6, cyclin B, Vasa, Pou3, Notch 1, Notch 2, Hes 1, Frizzled 5/8, Dishevelled, Beta catenin, piwi1 and piwi 2), and the probes that were used in this study to identify cells that express several genes (IA6, Vasa, Histone 3 Notch 1, Notch 2 by FISH) were designed against gene sequences derived from other ascidian species. Since genes among ascidian species are highly polymorphic and evolved rapidly (compare *B. schlosseri* gene model sequences to *B. leachi* gene model sequences or *Ciona* gene model sequences) one cannot assume that they will identify the same gene.

Furthermore, short sequences (about 20bp) designed for primers and probes can also identify other genes and when genome assembly is not available - it is very difficult to design a specific primer or probe that will be specific.

For example: I aligned the primers designed in this paper for the IA6 against the *Botryllus schlosseri* genome (using the *B. schlosseri* genome browser at Stanford university) and found that it matched 6 different gene models including: g39337- cd109, g34568- sele, g25556- PAFAH1b1. This probe did not align to the *B. schlosseri* IA6 gene model g37640 (called ITGA6).

To prevent confusion the authors should clone the genes that are key to this project from *Botrylloides diegensis* (e.g. ITA6, Vasa, Notch and Wnt), and Sanger sequence the PCR products to verify that the primers used amplify the genes of interest.

4. The results and methods sections are written in a sketchy and vague way. Number of colonies / genotypes that been used as controls and for treatments are mentioned briefly in the table but not included in the methods or results sections.

Details regarding blastogenesis stages used in the control and experimental groups are missing. Size of fragments used for WBR experiments is not included, methods used to identify remains of secondary buds within the vasculature fragments following zooid and bud removal (that in many cases can be overlooked) are missing. FACS protocols do not include: how natural fluorescent background was handled, which stains were used to distinguish between live and dead cells? Was live/dead stain used to measure viability of the cells following FACS sorting? FACS plots presenting natural background in the channel used for IA6 and other FACS data need to be presented. The percentage of cells stained by IA6 antibody needs to be clear, plots of FACS IA6 stains need to be presented in main figures (not in the supplement) and images of IA6+ and IA6- sorted populations need to be included. Controls that present natural background for ISH and EDU assays need to be added. Information regarding statistical analysis performed and how the sample size per each experiment was defined should be included.

No information is provided regarding the transplantation essays. Transplantation of a small number of cells or even a single cell is the most difficult experiment in stem cell studies. It is very difficult to assess the value of this experiment without providing any description of how transplantation was accomplished, including a detailed description that include images and videos that demonstrate the ability to transplant low number of cells into small fragments of vasculature. Data regarding the instruments used to inject cells, how cells were counted, how limiting dilutions were calculated to achieve a single cell transplantation, staining used to check viability of cells following FACS sorting and the stains used to label cells before transplantation to verify successful transplantation all need to be included.

Tunicates are known to live in highly contaminated areas (harbors) and present robust regeneration and survival capacities. Therefore, several important controls are needed to test their ability to resume regeneration following stress in order to understand their regeneration abilities under random stressors and understand the effect of the diverse treatments and experiments done in this study.

Other points

lines 15-19: "In the majority of chordates, the ability to regenerate following a major injury is severely limited, usually resulting in scar formation. In contrast, a group of invertebrate chordate species; colonial ascidians of the genus *Botrylloides*, have been shown to regenerate whole bodies, including all tissues and organs, from small fragments of the vasculature. This process is called Whole Body Regeneration (WBR)".

Add references to support claims.

Lines 55-57: "The source of the new bodies is a specialized region of the body wall of the parental zooid called the peribranchial epithelium, and this process of asexual reproduction is called palleal budding (Kassmer et al., 2018)."

Authors need to read and cite studies done by Manni lab (Padova university) that describe budding in *Botryllus schlosseri*. Buds do not originate in the body wall but in the peribranchial epithelium which is connected to the branchial sac.

"In addition to palleal budding, which occurs normally"

Authors need to describe in detail or refer to previous studies (if any) that describe palleal budding in *Botrylloides diegensis*. What are the normal budding stages of development in *Botrylloides diegensis*? Is it synchronized along the colony, how long does it take under specific temperatures? Do they hibernate like other botryllid species?

Lines 62-80: "*B. diegensis* can also use the alternative WBR pathway to produce a zooid following injury. WBR occurs in the vasculature, and is initiated by surgical removal of all zooids, or separation of small vascular fragments from the rest of the colony (Figure 1B). WBR progresses through distinct visual stages. For the first 24 hrs following surgical ablation, the blood vessels undergo regression and remodeling (stage 1, Figure 1B). During this time, blood flow, usually powered by the hearts of each zooid, continues and is driven by contractions of the remaining ampullae (Blanchoud et al., 2017). During the next 48 hrs, the blood vessels continue to remodel and form a dense, contracted network (stage 2, Figure 1B). Within this dense, highly pigmented vascular tissue, an opaque mass of non-pigmented cells becomes apparent; creating a clear area that is the presumptive site of bud development (stage 3, arrow in Figure 1B). The mass of cells next forms into a hollow, blastula-like epithelial sphere. The vascular epithelium then wraps itself around this sphere, leading to the formation of a distinct visible double vesicle. Over the next 48h, the inner vesicle increases in size (Stage 4, Figure 1B), while undergoing a series of invaginations and evaginations that lead to organogenesis and the eventual regeneration of a zooid. WBR is defined as complete when the new zooid is actively filter-feeding, and occurs within a range of 7-10 days (stage 5; Figure 1B). The zooid immediately commences normal palleal budding, and the colony regrow".

This section which discussed WBR in *B. diegensis* includes new information and should be part of the results. It should include the number of fragments tested and size and developmental stage used for WBR. Are the statements of "blastula like epithelial sphere" and "vascular epithelium wraps itself around this sphere" based on *in vivo* images? Or histological sections? If this is based on histological sections they should be included. If not, these terms cannot be used.

Lines 83-86 "WBR has been studied in three different species of botryllid ascidians, and in all cases a population of cells with an undifferentiated appearance, termed hemoblasts, have been suggested to initiate this regenerative process (Brown et al., 2009a)".

Please include and cite other works on vascular budding or WBR in botryllid ascidian including: Sabbadin et al. 1975; Miyamoto and Freeman, 1970; Rinkevich et al. 1995; Voskoboynik et al. 2007.

Lines 88-92: "In *Botrylloides violaceus*, 15-20 small hemoblasts that express Piwi protein have been shown to aggregate under the epidermis of a blood vessel during early WBR (Oka, 1959). These cells can be detected during the early vesicle stage and occasionally within the epithelium of a vesicle, suggesting that they play a role in regeneration (Brown et al., 2009a). In a related species, *Botrylloides leachii*, Blanchoud et al. showed an increase in the population of hemoblasts very early after injury during WBR (Blanchoud et al., 2017)".

Please include the work of Rinkevich et al. 2007 Plos Biology on *Botrylloides leachii* WBR

Lines 97-101: "In several species of colonial ascidians, the blood contains self-renewing, lineage restricted germline stem cells that migrate to newly developing buds during repeated cycles of asexual reproduction, where they give rise to eggs and testes (Brown et al., 2009b; Carpenter et al., 2011; Kawamura and Sunanaga; Laird et al., 2005; Rodriguez et al., 2016; Sunanaga et al., 2010; Sunanaga et al., 2006)".

Up to date self-renewal of stem cells was proven only in *B. schlosseri* (Laird et al. 2005) and not in any other botryllid species. Please do not mislead the readers and suggest that it was proven in other botryllid species.

Authors should also add the original works done on *Botryllus schlosseri* chimeras which were the first to demonstrate the presence stem cells in the of blood of this organism : Sabbadin and Zaniolo, Stoner and Weissman 1966; Stonner et al. 1998; Voskoboynik et al. 2008; Rinkevich et al. 2013

Lines 101-105: In *Botryllus schlosseri*, these germline stem cells can be enriched by flow cytometry using Integrin-alpha-6 (IA6) as a marker (Kassmer SH, 2015) and express piwi as well as other genes associated with germ cells, such as vasa, and pumilio (Brown et al., 2009b; Langenbacher and De Tomaso, 2016; Sunanaga et al., 2006).

Please highlight the limitation of the use of antibodies designed against mammalian cells in these studies and add the work done by Rinkevich et al. 2013 (Dev Cell) which used classic transplantation experiments and genotyping to demonstrate that the cell islands in *Botryllus schlosseri* are enriched with germline stem cells.

Nature Comms revisions

Reviewers' comments:

Reviewer #1 (Remarks to the Author):

This manuscript identifies a population of blood-borne cells responsible for whole body regeneration (WBR) in the colonial ascidian *Botrylloides diegensis*. These cells were found to co-express: (i) integrin- α -6 (IA6, also known as CD49f), a transmembrane receptor expressed in many types of cancer stem cells, (ii) *pou3*, a transcription factor of the Pou 5 gene family closely related to mammalian reprogramming factor Oct4, and (iii) *vasa*, an ATP dependent RNA helicase expressed in the germline of many animals. The authors show that one single IA6 cell can rescue WBR in vascular fragments treated with the anti-tumor Mitomycin C (MMC), in which stem cells and other cells in proliferation have been presumably depleted. Injected IA6 cells proliferate, migrate, and home into regeneration foci contributing to early precursor cells of WBR. These results demonstrate the stem cell potential (i.e. the capacity to self renew and to differentiate into multiple cell types) of IA6 cells.

Last, the authors report the up-regulation of expression of two important signaling pathways known to be involved in cell proliferation and regeneration during early stages of WBR. Finally, the authors conclude that the IA6 blood borne cells represent the pluripotent hemoblast population of colonial botryllid ascidians, functionally in close resemblance to neoblasts of planarians, i-cells of hydra, or archeocytes of sponges. Although not entirely unexpected, the degree of conservation in gene expression of colonial ascidian stem cells and mammalian stem cells contributes significantly to our understanding of the molecular and cellular mechanisms of blood-borne stem cells required to activate WBR, and highlights important implications about the evolution of stem cells in humans.

This study provides answers to long-standing questions about the nature of hemoblasts that have remained unanswered and extensively debated in the tunicate community for decades. Thanks to the progress in cancer and stem cell research, the authors have selectively isolated stem cells in the blood of colonial ascidians using CD49f, a well-known marker for tumor cells, and have used MMC, a chemotherapy agent, to deplete the stem cells in vascular fragments of the colonies. Because these approaches are novel and interesting, the experiments need to be rigorously tested, and experimental optimizations provided and well-presented. I found several instances in the manuscript, where additional evidence is, in my opinion, required. I was also disappointed with the last section of the manuscript, which explores the involvement of Wnt and Notch during WBR, because the qPCR expression patterns shown only leave speculative conclusions and raise questions about where they are expressed, and

the extent of their involvement in WBR. Besides, the expression of these two pathways during early WBR were already published for another *Botrylloides*

species, as noted by the authors themselves. However, this data only represents secondary results, and at this point are irrelevant to the main findings of the study i.e. the molecular characterization of IF6 cells with the functional rescue of these cells to the MMC treated colony fragments.

Major concerns:

One of the main characterizations of IA6 cells is that they express markers pou3 and vasa, but vasa has not been temporally characterized during WBR, as IA6 and pou3 have. It is difficult to evaluate the whole picture of IF6 gene expression dynamics during WBR if the complete expression patterns are not provided. For example, the authors do not show the complete co-expression results of pou3+vasa and pou3+IA6 during all stages of WBR.

We have performed FISH for pou3/ia6 and pou3/vasa at other stages, but had not included them because we did not feel they added relevant information to the paper. We have now added two additional images to the supplement: pou3/ia6 and pou3/vasa at stage 2. They show that pou3, vasa and ia6 are co-expressed at stage 2.

Additional controls are also necessary. Consider presenting one last timepoint (stage 5) with the expression of these genes at the end of WBR (full differentiation), as well as a general characterization of the expression of these genes in the blood of whole colony (as in Fig. 1A).

We focused on the early stages of WBR to determine when and how ia6+pou3+ stem cells proliferate to give rise to regeneration foci. Past the double vesicle stage, these stem cells are no longer involved and regenerating body undergoes normal development. We have added images of pou3/h3 FISH of stage 4 WBR samples to the supplement (Figure S2A). Once the new body begins budding (stage 5), the colony goes back to normal budding and growth and the stem cells go back to steady state (stage 0).

One suggestion is to organize your co-expression data into two figures, one that shows the dynamics of co-expression of stem cell markers in the blood cells at different timepoints, and another figure showing the proliferation dynamics of the cells expressing these markers + histone. For example, you could present one figure with only co-expression of stem cell markers (i.e. all different combinations of double FISH of pou3, vasa, and IF6; not histone) during all WBR stages together with the cell counts of these double labeled experiments into one figure; We appreciate the comment, but do not think that this would add relevant information to this study. We show both by FISH and qPCR that cells expressing IA6 co-express vasa and pou3 at steady state (Figure 1). To analyze the proliferation of these cells during WBR, we picked one marker (ia6) and quantified proliferation. We repeated this experiment with pou3, and found the

same trend. To show co-expression of *ia6*, *vasa* and *pou3*, we have now added two additional images to the supplement: *pou3/ia6* and *pou3/vasa* at stage 2. They show that *pou3*, *vasa* and *ia6* are co-expressed at stage 2. In recent experiments we found that upon asymmetric cell division, *pou3*, *ia6* and *vasa* are downregulated in the differentiating daughter cell at the same time, but these results will be published in a follow up study and are beyond the scope of this paper.

and another figure could include the co-expression of each of your markers with histone (i.e. all double FISH of *pou3*, *vasa*, and *IF6* with histone) during all WBR stages along with the cell counts of these double labeled experiments.

We have changed figure 2 to include all double FISH of *ia6* and *pou3* with histone 3 of all stages plus cell counts.

You have presented all the data for the latter (in Figures 2 and S2), but need to provide the complete dataset for the former (as you only show *pou3+vasa* and *pou3+ia6* on stage 0 in Fig. 1 D and 1E). All single labeled photos and single labeled cell counts for both figures can be placed as supplementary material.

I also found inconsistencies in the presentation of the data when comparing the labels of your photos to the labels in the quantifications (i.e. graphs of cell counts), for example in Fig. 1E you use a red label for *pou3* and green for *ia6* in the photo, but use red for *ia6* and green for *pou3* in the graph. Figure 2 also shows inconsistencies in the selection of color labels in the photos and in the cell count graphs. Please be consistent with the color labels.

Thank you for noticing! We have fixed this.

Another major concern I have is regarding the optimization of the depletion of proliferating cells using Mitomycin C (MMC). Can you show the survival curves and cell depletion experiments to show that the dose you are using (60mM) has been adjusted to botryllid ascidians?

These experiments should show the survival curves of the animals (or at least the fragments) with the proper evidence to show that cells have stopped dividing. A simple way to evaluate proliferating cell depletion under different doses of MMC would be to show the M2/G vs G0 FACS profiles under the different doses. In your figure S3A, you only show depletion of *pou3/h3* cells 5 days after MMC treatment?

A dose response curve showing the depletion would be preferred. As you know MMC at high doses may not only disrupt the proliferating cells, but MMC also disrupts transcription so high levels may affect the overall homeostasis of tissues and become toxic.

The same dose of MMC has been used in a previous study on Botrylloides

(Rinkevich et al., 2010). We tested this same dose (60uM) on our vessel fragments. The vessels continued to have blood flow and were arrested in stage 2, and remained that way for many weeks after removal of the drug. The 2% regeneration rate that we do see in MMC treated fragments suggests that the fragments themselves remain viable. The ability to rescue WBR by injection of IA6+ cells further shows that the fragments are healthy – otherwise WBR would not occur. We have analyzed cell proliferation and ia6 expression as early as 24h after MMC treatment, and both are completely eliminated. We have added these images to the supplement. We have repeated the same experiments with twice the concentration of MMC (120uM), and get the same results, even with rescue, so does not appear to be any general toxicity. At 30uM, WBR is impaired but not eliminated. Treatment of an intact colony with 60uM of MMC does not have any toxic effects and colonies remain normal, though bud development is slowed. We decided to show the 5 day time point to demonstrate that pou3+ cells remain absent even 5 days after removal of MMC.

I consider that the limiting dilution analyses of the IA6 cell injections in MMC treated fragments represent the most important and exciting results of this study, but have been poorly documented. You only evaluate the proportion of successful events of WBR after 20 days in the Table of Figure 3C, but do not show a more careful analysis of what happened under the different rescue experiments. Did you find that zooids fully differentiated at a later timepoint when lower numbers of IF6+ cells were injected?

Yes – as we say in the results, they begin budding and give rise to a new colony.

Did lower numbers of cells injected translate into less numbers of regeneration foci? Did you observe arrested buds that regressed under any of the circumstances?

Yes –it is normal for double vesicles to stop developing and get resorbed. During normal WBR in healthy samples, many foci and double vesicles are usually formed in each sample, and most of them stop developing while only a few continue all the way to stage 4 and stage 5. Therefore, counting foci or double vesicles is a poor measure of WBR capability. In later stages (e.g., stage 5) , one zooid tends to become dominant and begin budding and give rise to the whole new colony, while the others resorb. While definitely interesting, we did not quantify all these events for all the conditions tested, because the most important outcome of the rescue experiments for the present study is the ability to form a complete body with all tissue layers that is able to begin budding. We will analyze the processes underlying foci formation in greater detail in future experiments.

Can you show or describe more carefully what you expected and what you observed under each cell injection scenario? Do you have the evidence that shows that the IF6 cells you injected are also the EdU cells that were tracked?

Since the cells that were injected were Edu-labeled IA6+ cells, there could not have been any other Edu-labeled cells present.

What is the temporal and spatial dynamics of the injected IF6 cells when in low or high numbers in the rescue? Do they form few foci and expand? Do they expand first and form foci later? There are many questions to be answered, and important information that can be documented by doing a more careful analyses of the rescue experiments. These questions are very interesting, and we will address them in future studies, but we feel this goes way beyond the scope of this paper. The point of this paper is that we have identified the stem cell population responsible for WBR. Detailed characterization of the nature of these stem cells and the processes that occur during WBR are future directions. There are more than enough questions to answer for several follow up studies. We have done pou3-FISH on some rescued, stage 4 samples injected with Edu-labeled IA6+ cells and show that some bright Edu-positive cells in the blood express pou3, even after WBR has progressed past the double vesicle stage (image in Figure S3C). This suggests that pou3+ ia6+ cells self renew in the recipient. We have changed Figure 4 to show that Edu-labeled IA6+ cells give rise to small aggregates of Edu+ cells which later grow into bigger aggregates before forming the double vesicle. Assessing when cells lose expression of pou3 and ia6 during foci formation is the subject of follow-up studies, but we have early evidence that pou3 expression is lost as soon as a stem cell divides asymmetrically to give rise to differentiating progeny.

How many cells were injected when you used EdU? Could you recover this EdU signal if one single or few IF6A cell were injected? And when many cells are injected? We injected 1000 Edu-labeled ia6+ cells into MMC treated recipients. We have now added this information to the Results and Methods and apologize for the oversight. If only one Edu+ cell was injected, we believe that the chances of recovering the signal would be low, but we are going to attempt this in future experiments to assess whether one ia6+ cell can give rise to all germ layers within the regenerating body.

Were the injected EdU cells equally distributed among all foci, or did some foci show more EdU cells than others? How far in time (stage?) is the EdU signal traceable during WBR? During organogenesis? I think these experiments could potentially also be explored in more detail. Edu-positive cells derived from the donor can still be detected at stage 4, when organogenesis is occurring. We have now changed Figure 4 to include images showing Edu+ cells derived from IA6+ at stage 4.

To my knowledge this is the first time that DAPT and Endo-IWR are used in colonial ascidians as inhibitors, and therefore the procedures and doses should be well-documented. Can you include some tests where you tested different doses and windows of sensitivity of the inhibitors on untreated whole colonies?

What were the dose responses observed on blastogenesis? Documenting effects on blastogenesis could presumably allow you to observe the specific dose that is required to disrupt distinct processes of development.

We tested different doses of the inhibitors as shown in Figure 6B. The effects on WBR are dose-dependent and there are no signs of toxicity, as treated vessel fragments can recover from treatment when the drug is removed. The inhibitors strongly affect stem cell proliferation. In future experiments we will characterize the effects of the inhibitors on the stem cells in greater detail. For the present study, we have not assessed the effect of the inhibitors on healthy colonies other than that there is no toxicity over the course of several days.

The results of notch and delta inhibitors show that they affect the final outcome of WBR, as well as formation of regeneration foci. It would be interesting to show either in the main text or as supplementary information more detailed descriptions of the effects at different timepoints if not by histology at least externally: how many foci were observed? In the most effective doses (DAPT2uM and Endo-IWR 1uM), no foci formation is observed. For the lower doses, we only counted how many samples regenerated all the way to stage 4. During normal WBR, many foci and double vesicles are formed in each sample, and most of them stop developing while only a few continue all the way to stage 4 and stage 5. Therefore, counting foci or double vesicles is a poor measure of WBR capability.

What were the final numbers of buds and zooids that differentiated in each replicate and for each treatment? The final number of differentiated buds and zooids is very variable even in untreated samples, therefore we did not use this as a measure of the effects of the drugs.

How did these inhibitors affect the timing of each stage of WBR? In the effective dose, the samples get arrested in stage 2. When the inhibition was reversed, did the different times at which the inhibitors were removed affect the time of recovery of WBR? The drugs were removed at 96h because at that time the effect on pou3+ cell proliferation is very profound and it was important to test if it is still reversible at that time. Upon removal of the drug, samples take longer than the usual 7 days to complete WBR, but no longer than 14 days (Figure 6).

If these data cannot be included in the main text due to word number constraints, it should be included in organized Tables as Supplementary Material.

The disruption of cell aggregates and regeneration foci with the inhibition of wnt is interesting, but somewhat surprising for notch signaling as this pathway requires cell-cell contact for activation. With Notch signaling, a sending and receiving cell need to be in close proximity for signaling to occur. The sending cell could be either another stem cell or present in the surrounding niche. The evidence presented here suggests that Notch ligand expressed by sending cell is required for a pou3+ cell expressing Notch receptor to undergo cell division (symmetric or asymmetric, to be determined). When this division is blocked, foci

do not form, and pou3+ cells remain scattered like in steady state (stage 0, Figure 2). If Notch is regulating asymmetric cell division, then differentiating progeny is not generated.

To test the hypothesis that notch signaling disrupts regeneration foci, you would expect in control that notch is activated either before or at the very early contact of cells when forming aggregates, therefore it would be interesting to show a control stage 3 untreated fragment presumably with notch expression in those early aggregates, but unfortunately your data only shows WBR fragments until stage 2. We are in the process of characterizing the precise timing and detailed roles of Notch receptor and ligand expression in stem cells and their progeny. Our data show that Notch signaling is a very early signal that acts on the stem cells themselves and induces proliferation and foci formation. It is possible that Notch signaling plays a different role at later stages when tissues begin to differentiate, and this role will be similar to its role during normal bud development. However, we are specifically interested in how signaling pathways regulate stem cell behavior during the initiation of WBR.

A section needs to be included in the discussion that mentions and discusses possibilities on why radioprotection assays failed in previous Botryllus schlosseri experiments (see Laird and Weissman, 2004; and Rinkevich and Weissman, 1990). The results on these studies are somewhat different from those observed here in B. diegenensis. For example, in B schlosseri Laird injected up to 80000 cells into radiated colonies and could not find any rescue, and also Rinkevich observed that radiated colony parts could not be reconstituted even after non-irradiated isogenic colonies were fused. How do these apparently opposite results in these two closely related species be explained? Do the authors think different blood cells occur in B schlosseri and B. diegenensis? I would encourage the authors to briefly mention these discrepancies and discuss some possible implications of their findings to explain these differences across species. Irradiation could be doing many more things than killing stem cells. It could be affecting physical or cellular niches, who knows? Irradiation experiments used for HSC transplant are not even equivalent among different mouse strains. This is not relevant to this study.

Minor corrections:

Line 29: Delete 'phyla of'

Line 66: Cite Zondag et al. (2016) here. Even though the stages for B. diegenensis may have been modified and adjusted from those stages in B. leachii, the main events of WBR stages used are very similar to those proposed by Zondag et al. (2016). Done

Lines 66-80: Please, include the range of time frames in the paragraph when describing each stage. Thereafter, include both TIME and STAGE when describing your data. There are some instance where you mention only the time and other instances where you only mention the stage, for example in Figure 2 you only mention stages but not time. Because some stages overlap in timeframes, I think both should be used every time, as in stage 5 (day 20) or st. 3 (72h). We changed the paragraph describing the stages to better explain the timing. Because the timing of the stages is variable even among healthy samples, we use staging by morphology for accuracy when analyzing the FISH results and Edu-transplant results. Within the first 48h, samples tend to follow the same pattern and are very synchronized, so we use time as a measure for these early time points when analyzing gene expression. Because inhibitors might change the timing of progression through the stages, we use time as a measure for these experiments.

Line 86: In fact, four species: *B. primigenus*, *B. schlosseri*, *B. violaceus*, and *B. leachii* [Oka and Watanabe (1957, 1959), Voskoboynik et al. (2007), Rinkevich (2007)] We are strictly distinguishing between vascular budding that occurs as part of normal colony development - as in *B. primigenus* and *B. tuberatus* - and whole body regeneration, which is induced by removal of all zooids and buds (Kassmer et al., 2019). *Botrylloides leachii*, *Botrylloides diegensis* and *Botrylloides violaceus* all undergo WBR, whereas *B. primigenus* and *B. tuberatus* undergo vascular budding. *B. schlosseri* does neither. We have edited this section to talk about *Botrylloides* only.

Line 88: Correct citation, change Oka, 1959 for Brown et al. (2009a). Done.

Lines 99-101: Include the following citations: Brown et al. (2009b), and your own Langenbacher and De Tomaso (2016). Done

Lines 133-135: You need to include here that *pou3* was previously found expressed with relatively high expression in few scattered cells in the inner vesicle of the early bud during blastogenesis in *Botryllus schlosseri* (Ricci et al. 2016) If we mention this result, we would need to mention that the *pou3*-FISH signal in Ricci et al is very faint and spotty (reminiscent of probe trapping), and needs to be repeated. We have done *pou3* FISH on whole colonies from both *B. schlosseri* and *B. diegensis* and never seen the same signal that Ricci et al report. We detect *pou3* in blood borne cells and germ cell progenitors (shown in Figure S1B). We have some results suggesting very faint expression in the early secondary bud at A1, but no later than that. And again, this is not relevant to this study.

Line 138: Why did you normalize your gene expression to *vasa*? Wouldn't you want a gene that is not affected by your treatments? *Vasa* is selectively expressed in some cell types and it is known to cycle and change expression levels during blastogenesis. We have edited this section in the results to make

the approach more clear. This qPCR is done only on the IA6+ cells isolated from healthy colonies, so there is no treatment. We show in a separate qPCR (Figure S1 D) that IA6+ cells are enriched for all the genes associated with stemness and germ cell identity that we tested (*ia6*, *vasa*, *pou3*, *piw1* and *piw2*) when compared to IA6- cells. However, to show the relationship of expression levels between those genes within the IA6+ cells themselves, we normalized to *vasa* because this is the most basic and most highly conserved germ cell associated gene. This way, we can see that *pou3* and *ia6* are expressed at slightly higher levels than *vasa*, while *piw2* expression is comparatively low, and *piw1* is almost absent. This is relevant because previous studies had placed emphasis on the role of *piwi* expression during WBR. We show that *pou3* and *ia6* expression are higher and likely more or equally important.

Line 169: Consider revising the use of 'steady state' to simply define similar proportions of cells. To me the use of 'steady state' has broader implications including an overall maintenance of cell proportions and some sort of homeostasis in the whole system. *We have edited this section for clarity.*

Lines 171-174: Correct and change sentence. I do not see 64% quiescent Pou3+ cells at stage 3, nor 71% of cell proliferation in pou3- double vesicles at stage 3 in Figure S2. Where are these cell counts shown? I do not see a single yellow arrow in Figure S2? I only see yellow arrow in Figure 2. *We have edited Figure 2 to include the FISH results for both ia6 and pou3 with histone 3, and changed the graph to show the overlap between ia6 or pou3 and h3. This graph now shows the percentages mentioned in the text. The graph showing the percentages normalized to the number of nuclei was moved to the supplement.*

Lines 175-177: Consider softening your concluding sentence here, as you have not shown carefully the spatial and temporal dynamics of double labeled ia6/pou3 cells throughout WBR.

Lines 183-184: In figure S3A, why do you evaluate depletion of pou3 cells, and not IA6 cells directly? To also show that all proliferating cells have been depleted, but the MMC treatment did not show toxicity stronger evidence is required. The ideal experiment would be to show by FACS that G2/M cells have been depleted, or perhaps you can show that the 60nM MMC dose you used also disrupts blastogenesis in complete colony, but does not affect overall homeostasis of the tissues. However, it would be better to show the dose dependence curves requested above. *We are showing depletion of histone 3+ proliferating cells and added on pou3 as a marker for the stem cells to show that they are eliminated as well. We have now added another image showing depletion of ia6+ cells at 24h post MMC treatment. The same dose of MMC has been used in a previous study on Botrylloides (Rinkevich et al., 2010). We tested this same dose (60uM) on our vessel fragments. The vessels continued to have blood flow and were arrested in stage 2, and remained that way for many weeks after removal of the drug. The 2% regeneration rate that we do see in MMC treated fragments suggests that the*

fragments themselves remain viable. The ability to rescue WBR by injection of IA6+ cells further shows that the fragments are healthy – otherwise WBR would not occur. We have analyzed cell proliferation and ia6 expression as early as 24h after MMC treatment, and both are completely eliminated. We have added these images to the supplement. We have repeated the same experiments with twice the concentration of MMC (120uM), and get the same results, even with rescue, so there does not appear to be any general toxicity. At 30uM, WBR is impaired but not eliminated. Treatment of an intact colony with 60µM of MMC does not have any toxic effects and colonies remain normal, though bud development is slowed.

Lines 186-189: So the MMC treated fragment form typical aggregates of stage 2, and then arrest? It worries me that you mention that '...most MMC treated blood vessels...' do not reach stage 4 or 5, because this could mean that you are not using the optimal dose in your experimental assay... This comment is not entirely clear. The MMC treated vessel fragments undergo the same remodeling process as untreated fragments until stage 2, and then do not progress further after that. This arrest in stage 2 is defined by morphology, because cell proliferation is absent (Figure S3A), so no regeneration foci are formed.

Line 190: here you mention that you injected '...2000 total blood cells...' but in Figure 3B you state that you injected IA6+ cells. Which is it? Thank you for noticing. This sentence was supposed to refer to the table in Figure 3C – we fixed it.

Line 199: In Figure 1E, you show that 9-10%??? of the cells that are pou6+ are IA6-, not 20%. When normalized to pou3+, the percentage of ia6- is 20%. Since we are not showing a graph normalized to pou3, we have changed this sentence.

Line 226: In Figure S3B, where are the white arrows? I only see green arrows. We have changed Figure S3 almost entirely and corrected all errors.

Lines 256-259: Can you show downregulation of these in later stages of WBR? What are the base levels of expression in a colony?

Lines 278-280: Why do you check Notch inhibitor effects on pou3 cells, and not on IGF6? Pou3 is more highly upregulated in proliferating cells during early WBR, and since proliferating cells are affected by the inhibitors, pou3 is a good marker to use in addition to histone 3.

Line 286: When were the inhibitors removed? Did the timing of inhibitor removal affect the recovery process and WBR? As stated in the table in figure 6B, the inhibitors were removed after 96h of treatment, and these samples progressed to complete WBR by day 14. We did not test other time points of inhibitor removal – since vessel fragments retain the ability to regenerate after 96h of treatment, we

expect that the same would be true at earlier time points.

Line 299: Year is missing in a reference. **Fixed**

Line 307: Vasa and Piwi have also been previously associated to somatic functions as well, see Alié et al. (2011), Gonzalez et al. (2015), and Poon et al. (2016). These studies could be cited here as well as other cases where these markers are expressed in stem cells, aside from the germline or germ cells. **The point is that vasa and piwi can be expressed in stem cells that have germline as well as somatic potential. We have expanded the section of the discussion that explains the primordial stem cell hypothesis and the expression of GMP genes in these cells.**

Line 343: The author states "...all somatic tissues that are formed during asexual reproduction (palleal budding) in healthy animals are derived from the peribranchial epithelium...". I am not sure we know this. To my knowledge, no-one has been able to exclude the contribution of blood borne stem cells in palleal budding. I agree that the initial stages up to the double vesicle are derived from the peribranchial epithelia, but after that it remains uncertain. **At least one careful study in Botryllus shows that blood borne stem cells do not contribute to palleal buds (Carpenter et al., 2011).**

Figure 1D: Please do not use e charts, you can use a simple bar graph instead. Also the colors of the labels are not consistent between the photos and the graphs. **We adjusted the colors but like how the pie graph shows the ratios better than a bar graph.**

Figure 2: Colors of labels between photos and graphs are inconsistent. Why was stage 1 not included? Include the times sampled for each stage. **We changed Figure 2, which now has all stages from 0-3 and both ia6 and pou3. We changed the graph to better show the percentages of proliferating cells. All samples used for histological analysis were staged by morphology after sectioning, as timing is very variable between samples. All stage 1 samples are 24h, as explained in Figure 1B.**

Figure 3C should be a separate Table and not inserted as C in the figure. This Table is the most important result of this study, but it is not well organized or presented (for example, rates and percentages can be placed in the same box, the names of categories in the left column could be improved to facilitate understanding, include subtotals and totals, explain what experiment 1 and 2 mean (are these replicates different in any way? Were they done in different days or the same days in parallel? **Experiment 1 and 2 are two independent experiments, as stated in the figure legend. For each experiment, we are showing the counts (is this what you mean by sub-totals??) and we show the total counts and percentages calculated from these.**

Figure 5: Where are your negative controls? Do you have some markers of differentiated cells that you can use, or genes that you expect not to be up-regulated? In Figure 5B, expression levels are normalized to G0 cells, and it shows that G2/M cells express higher levels of cyclin B, as expected. In Figure 5C, all expression levels are normalized to 0h.

Figure 6B should also be placed as a Table and not embedded in the figure, perhaps with a better documentation of the different timepoints? One would expect a lower recovery with longer inhibitor incubations. We think it is important to show the outcomes next to the FISH results. Lower recovery after longer incubation would be expected only if these inhibitors are affecting self-renewal. This is a possibility that we will explore in follow-up studies.

Figure S2 contains expression data of pou3, one of the main markers (if not the main marker) of stem cells that is proposed to be co-expressed with IA6 cells. Shouldn't this data be in the main text and not in supplementary? Also why are these two markers not shown together in a double FISH during the different timepoints of WBR? Consider moving Fig. S2A to Figure 2, as well as the complete cell counts of the double and single FISH. Then, the micrographs that show h3 and IA6 independently in single color channels (middle and lower row of Fig 2) could be moved to the supplementary material. We have changed figure 2 and combined all the ia6/h3 and pou3/h3 results.

Figure S3: Do you have a double FISH with IA6 and h3, instead of pou3 and h3? Where are the white arrows? Do you mean green arrows? We have made substantial changes to Figure S3 and moved many of the images to Figure 4.

Additional references:

Alié A, Leclère L, Jager M, Dayraud C, Chang P, Le Guyader H, Quéinnec E, Manuel M. Somatic stem cells express Piwi and Vasa genes in an adult ctenophore:

ancient association of "germline genes" with stemness. Dev Biol. 2011 Feb 1;350(1):183-97. doi: 10.1016/j.ydbio.2010.10.019. Epub 2010 Oct 29. PubMed PMID:

21036163. Nice but this is not a functional study, and provides no strong support of the primordial stem cell hypothesis.

Gonzalez, J., Qi, H., Liu, N., & Lin, H. (2015). Piwi Is a Key Regulator of Both Somatic and Germline Stem Cells in the Drosophila Testis. Cell reports, 12(1), 150–161. doi:10.1016/j.celrep.2015.06.004 There are so many studies showing piwi expression in somatic stem cells that this cannot be seen as support for the

primordial stem cell hypothesis. The new idea is that piwi plays a role in self renewal of somatic stem cells, so piwi is actually not a good germ cell marker.

Laird DJ, Weissman IL. Continuous development precludes radioprotection in a colonial ascidian. *Dev Comp Immunol*. 2004 Mar;28(3):201-9. PubMed PMID: 14642887.

Oka, Hidemiti, and Hiroshi Watanabe. "Vascular Budding, a New Type of Budding in Botryllus." *Biological Bulletin* 112, no. 2 (1957): 225-40. Accessed February 4, 2020. doi:10.2307/1539200. This study is on vascular budding, which naturally occurs in *Botryllus primigenus* as part of normal asexual reproduction. Here, we purposely focus only on whole body regeneration (WBR), which is induced by removal of all the zooids and buds. Of course, vascular budding and WBR are related to each other. But for the broad audience of this manuscript, this would be going into too much depth on ascidian biology and be very confusing.

Poon J, Wessel GM, Yajima M. An unregulated regulator: Vasa expression in the development of somatic cells and in tumorigenesis. *Dev Biol*. 2016 Jul 1;415(1):24-32. doi: 10.1016/j.ydbio.2016.05.012. Epub 2016 May 11. Review. PubMed PMID: 27179696; PubMed Central PMCID: PMC4902722. We have added a similar reference from the same group, focusing on the function of vasa as a cell cycle regulator in stem cells with somatic potential.

Ricci, L., Chaurasia, A., Lapébie, P., Dru, P., Helm, R. R., Copley, R. R., & Tiozzo, S. (2016). Identification of differentially expressed genes from multipotent epithelia at the onset of an asexual development. *Scientific reports*, 6, 27357. doi:10.1038/srep27357 not relevant (Palleal budding)

Rinkevich B, Weissman IL. *Botryllus schlosseri* (Tunicata) whole colony irradiation: do senescent zooid resorption and immunological resorption involve similar recognition events? *J Exp Zool*. 1990 Feb;253(2):189-201. PubMed PMID: 2313247.

Rinkevich Y, Paz G, Rinkevich B, Reshef R. Systemic bud induction and retinoic acid signaling underlie whole body regeneration in the urochordate *Botrylloides leachi*. *PLoS Biol*. 2007 Apr;5(4):e71. PubMed PMID: 17341137; PubMed Central PMCID: PMC1808485. Not relevant – does not show proof of blood borne stem cells.

Voskoboynik A, Simon-Blecher N, Soen Y, Rinkevich B, De Tomaso AW, Ishizuka KJ, Weissman IL. Striving for normality: whole body regeneration through a series

of abnormal generations. FASEB J. 2007 May;21(7):1335-44. Epub 2007 Feb 8. PubMed
PMID: 17289924. **Not relevant – no vascular budding or WBR in schlosseri (study by our lab to be published soon).**

Reviewed by,
Federico D. Brown

Reviewer #2 (Remarks to the Author):

Some colonial ascidians can undergo whole body regeneration – that is, reconstitute an entire individual from blood vessels and stem cells. This manuscript uses colonies treated with Mitomycin C (MMC) where cell division is inhibited so that the colony cannot regenerate in order to isolate the stem cell population that is necessary for whole body regeneration. Cell sorting of blood cells revealed a specific type of integrin alpha-6 positive cell was the key to allowing whole body regeneration. Edu labeling subsequently showed that the regenerated tissues were generated from the integrin alpha-6 positive cells. These are extraordinary results, isolating a specific type of circulating stem cell that is required for whole body regeneration in colonial ascidians. These are exciting and novel and give insight into the circulating stem cells in colonial ascidians.

In the second paragraph of the Introduction, there are no references for previous papers on Whole Body Regeneration (WBR) in Botrylloides species. There have been some important and insightful papers written on this process and those should be referenced within (for ex. Blanchoud et al. 2018 (Botrylloides leachei) Brown et al. 2009 (Botrylloides violaceus), etc.). **We agree. All of these studies are cited in later paragraphs, but we have now included them here as well.** Paragraph 3 in the Introduction also needs references, describing the life cycle and bud formation in Botrylloides diegensis.

The results reported here also give insight into the normal process of whole body regeneration in B. diegensis. It appears that the integrin alpha-6 positive cells aggregate at the site of WBR, and then the blood vessels close around the clump of cells. It is then shown that the integrin alpha-6 positive cells are responsible for dividing to undergo whole body regeneration and these processes are regulated by Notch and Wnt signaling. These are inhibitor studies, in which results are made stronger by the fact that when the inhibitors are washed out, the integrin alpha-6 positive cells begin to proliferate and undergo whole body regeneration. This shows that Notch and Wnt signaling are necessary to upregulate proliferation of the cells during normal regeneration.

Reviewer #3 (Remarks to the Author):

In this manuscript the authors studied whole body regeneration (WBR) in a wild type *Botrylloides diegensis* species. They used antibodies developed against human/mice integrin alpha-6 (ITGA6) and flow cytometry to isolate a population of cells and perform diverse transplantation experiments to measure their ability to proliferate. They designed probes and primers to detect several genes based on their sequence in other botryllid species (*Botryllus schlosseri* and *Botrylloides leachii*), and used them to measure levels of expression (qPCR) and sites of expression (ISH) of these genes in *Botrylloides diegensis*. By combining EDU tracing experiments and ISH they found that the sorted cell populations (cells stained by the mammalian antibodies of integrin alpha-6) include a highly proliferate population of small cells that are also positive to the *B. schlosseri* and *Botrylloides leachii* probes aimed to detect integrin alpha-6 (ITGA6), *pou3* and *vasa* (DDX4). By treating vasculature fragments taken from *Botrylloides diegensis* (undefined size, undefined stage) with mitomycin C they prevented whole body regeneration and then succeeded in inducing whole body regeneration by transplanting thousands of cells from non-treated individuals, 50 ITGA6+ sorted cells, and even 1 ITGA6+ cells. FACS plots that present the expression of this antibody suggest that at least 90% of the *Botrylloides diegensis* blood cells express this protein in high levels (above 10² Figure S1 C).

Developing new model organisms to study regeneration is important and can add to our basic knowledge about cellular and molecular mechanisms of regeneration. The authors performed an impressive set of experiments and tests aiming to study mechanisms of regeneration in a colonial chordate species for which very little information is available about its life cycle and biology. However, several major issues need to be addressed before one can assess the real impact of this study and publish this paper.

1. Stem cells are self-renewing and multipotent cells. This study does not include experiments that prove self-renewal or multipotency and therefore the use of the term stem cells throughout the paper is inappropriate. When discussing previous work on candidate stem cells in diverse botryllid species, the authors should distinguish between papers that indeed found and verified stem cells (e.g. Laird et al. 2005) versus papers that only point to candidate stem cell populations (e.g. Brown et al. 2009b).

To prevent confusion, studies that focus on expression of genes associated with stemness without experiments that clearly demonstrate self-renewal and multipotency should not be discussed and reviewed as studies that succeeded in isolating stem cells.

ia6+ cells are constantly cycling in a healthy colony- as shown by expression of cyclin b and histone 3, and the fact that G2/M cells express ia6 and pou3. The fact that ia6+pou3+ cells are constantly cycling on a healthy colony and never exhaust

shows that they self renew. We have added an experiment showing pou3 expression in Edu-positive cells after transplantation of Edu-labeled Ia6+/pou3+ cells (FigureS3C). Pou3+ edu+ cells that are not in foci or double vesicles (pink arrows) are derived from self-renewing cells that were transplanted. We have added new text discussing self-renewal to the results and discussion.

2. The authors present a novel model system but do not include basic information regarding its biology, life cycle, regeneration capacities, We disagree. We present detailed information regarding colony morphology and regeneration stages in figure 1. We give extensive detail on the biology of B. diegensis and the process of whole body regeneration in the introduction (page 2 line 19 to page 4 line 2). This paragraph is 465 words long – we feel this is more than sufficient. We give extensive references for readers who are interested in more detail on Botrylloides and WBR.

and natural fluorescent background under normal and stress conditions, information that is crucial to assess the outcomes of the experiments (mainly experiments that induce stress and outcomes that are detected by fluorescent product). We performed negative controls for all the stainings at all the stages. The fluorescent signals shown are much brighter than background. We have included an image of FISH-negative control generated with sense-probes (Figure S2A) and the isotype control for the FACS antibody (Figure S1C).

The authors use Botrylloides diegensis colonies collected in the wild; thus age, individuality (naïve/ chimeras) and history are unknown and cannot be determined. This is adding unmeasurable variables to the study and needs to be included in consideration when designing the experiments (high numbers of repeats) and when drawing conclusions. Only healthy, well growing colonies were used for the experiments presented here. Controls were subclones from the same colonies as the experimental samples, and therefore the same age and genotype. Time points were taken from subclones of the same genotype and repeated on at least 3 different genotypes per experiment. We have added this information to the methods. We have never observed any significant changes in any sample collected. WBR is incredibly robust in B. diegensis.

3. The authors use an array of antibodies, small inhibitors, probes and primers developed against mammals or other botryllid organism. This is a major caveat and without verifying that each of these reagents indeed detects or blocks the gene / protein it was originally designed against, conclusions should not be drawn.

We validated the antibody used for FACS with the appropriate controls and showed that it enriches for IA6+ cells (qPCR in Figure 1C). The probes and primers were actually designed against B. diegensis sequences, because we mainly used our own transcriptome database to find sequences (information now included in Methods). Most sequences labeled B.leachii in public databases are

actually *B. diegensis* (Viard et al., 2019), and we have included this reference in the manuscript. We had previously misidentified our species based on these mistakes in the public databases. We have now corrected all errors regarding species names. All gene fragments cloned for FISH probes were confirmed by sequencing and the probe sequences are now included in the supplement. We used control drugs to show specificity of the inhibitors, and show a specific effect of the inhibitors on the pou3+ stem cells (Figure 6).

4. The results and methods sections are written in a sketchy and vague way limiting the ability of other researchers to reproduce this work in the future. Important information regarding transplantation experiments and FACS protocols are lacking.

If we knew exactly what information we should add? What specific details are missing? We strongly feel that the methods contain more than enough detail for other researchers to reproduce our findings, but we have now added a lot of additional detail to the Methods section. Anyone is always welcome to contact us with questions regarding the methods and we can provide help and pass along detailed protocols.

No information is provided regarding natural fluorescent background under normal and stress conditions. We performed negative controls for all the stainings at all the stages. The fluorescent signals shown are much brighter than background. We have included an image of FISH-negative control generated with sense-probes (Figure S2A) and the isotype control for the FACS antibody (Figure S1C).

No information is provided regarding the size of fragments used for WBR and the stage of development tested. This information is crucial since it's been shown that WBR is dependent on fragment size and stage in botryllid ascidians (e.g. Rinkevich et al. 1995 PNAS; Voskoboynik et al. 2007 FASEB). In *Botrylloides*, WBR is not stage dependent. We have included this now in the introduction. Furthermore, small fragments (5 ampullae) and large fragments (larger than 1cm) alike produce over 90% of WBR in our hands and always give rise to one dominant zooid that begins budding to regenerate the entire colony. In all the experiments, the fragments used were of comparable size (about 5 square millimeters).

Images that follow control and transplanted fragments from day 1 until the end of the experiments need to be presented. No information is provided regarding the transplantation assays, making it very difficult to assess the value of this experiment. Transplantation of a small number of cells or even a single cell is the most difficult experiment in the stem cell field, therefore a detailed description that includes images and videos demonstrating the ability to transplant low number of cells into small fragments of vasculature is imperative. We have added

the following information to the Methods: Sorted cells were collected into 3ml of filtered seawater, pelleted by centrifugation (700g, 10 minutes) and resuspended in 100ul filtered seawater. Cells were counted using a hemocytometer. Only round, bright, intact cells were counted. Cells were labeled with Hoechst 33342 and diluted in filtered seawater to the appropriate concentration. 0.1ul volume containing either 1000, 50, 10, 5 or 3 cells was injected per sample. For injections of 10 cells or less, 1ul cell suspension stained with Hoechst 33342 was pipetted onto a microscope slide and cells were counted under brightfield and fluorescence to verify the correct cell concentration. For single cell injections, cells were diluted so that 1ul contained 5 cells, and 0.1ul were injected per sample = 0.5 cells per sample. Cells were injected into ampullae of MMC treated vessel fragments with pulled glass capillary needles and an Eppendorf FemtoJet microinjector under a stereomicroscope.

The authors need to include Images that follow data regarding the FACS sorting and the dyes used to label cells before transplantation to verify successful transplantation. We have added the following information to the Methods: Sorted cells were collected into 3ml of filtered seawater, pelleted by centrifugation (700g, 10 minutes) and resuspended in 100ul filtered seawater. Cells were counted using a hemocytometer. Only round, bright, intact cells were counted. Cells were labeled with Hoechst 33342 and diluted in filtered seawater to the appropriate concentration. 0.1ul volume containing either 1000, 50, 10, 5 or 3 cells was injected per sample. For injections of 10 cells or less, 1ul cell suspension stained with Hoechst 33342 was pipetted onto a microscope slide and cells were counted under brightfield and fluorescence to verify the correct cell concentration. For single cell injections, cells were diluted so that 1ul contained 5 cells, and 0.1ul were injected per sample = 0.5 cells per sample. Cells were injected into ampullae of MMC treated vessel fragments with pulled glass capillary needles and an Eppendorf FemtoJet microinjector under a stereomicroscope.

The authors need to include methods and measures used to assess viability in the fragments treated and normality of developing buds like histological sections that follow development. If the MMC treated fragments were not viable, rescue by cell transplantation would not work. In order to accurately quantify the success of the rescue, only fragments that reach stage 5 (complete filter-feeding body that begins palleal budding) were counted. All fragments that reach stage 5 eventually give rise to entire new colonies by asexual reproduction.

Major points:

1. Stem cells are self-renewing and multipotent cells. The paper does not include experiments that prove self-renewal or multipotency, and therefore the use of the term stem cells throughout the paper is inappropriate. The assay used in this study is showing complete somatic reconstitution (i.e., WBR), and this is clearly shown using limit dilution rescue and lineage tracing. For self-renewal, this is also clearly demonstrated in nearly every

figure. Ia6+ cells are constantly cycling in a healthy colony– as shown by expression of cyclin b and histone 3, and the fact that G2/M cells express ia6 and pou3 (Figure 2, Figure 5). The fact that ia6+pou3+ cells are constantly cycling in a healthy colony and never exhaust for the life time of the animal shows that they self-renew. We have added an experiment showing pou3 expression in Edu-positive cells after transplantation of Edu-labeled Ia6+/pou3+ cells (FigureS3C). Pou3+ edu+ cells that are not in foci or double vesicles (pink arrows) are derived from self-renewing cells that were transplanted. We have added new text discussing self-renewal to the results and discussion.

When discussing works done in the past on candidate stem cells in diverse botryllid species the authors should distinguish between papers that indeed found and validated stem cells and between papers that only point to candidate stem cell populations. Studies that focus on expression of genes associated with stemness without experiments that clearly demonstrate self-renewal and multipotency (transplantation and genotyping) should not be discussed and reviewed as studies that isolated stem cells. **We have edited the references and text referring to them to make sure everything is accurate.**

Throughout the paper (introduction, discussion) the authors should include comprehensive work done on Botryllus schlosseri chimeras which already prove that stem cell originating in cells circulating in the vasculature are contributing to germline and somatic tissues in the developing buds. Just to name a few: Sabbadin and Zaniolo 1979 J. Exp. Zool; Stoner and Weissman, 1996 PNAS; Stoner et al. 1999 PNAS; Voskoboynik et al. 2008 Cell Stem Cell; Rinkevich et al. 2013 Dev. Cell; Rosental et al. 2018 Nature. First, we are not studying Botryllus schlosseri. Second, the two mobile progenitors in B. schlosseri are for germline and hematopoietic cells, which was already shown in Laird, et al. (2005) and Carpenter et al., (2011). There is no direct evidence at all that mobile somatic progenitors give rise to zooid tissues. This is irrelevant to this study.

The authors also failed to cite original work describing vascular budding / WBR in botryllid species including: Sabbadin et al. 1975 Dev Biol; Rinkevich et al. 1995 PNAS; Voskoboynik et al. 2007 FASEB. Also, the phrase “whole body regeneration”-WBR- was coined by Rinkevich et al in 1995. **It is important to distinguish between vascular budding, which naturally occurs in Botryllus Primigenus (and some other species of colonial ascidians) as part of normal asexual reproduction. Here, we are studying whole body regeneration which occurs in species of the genus Botrylloides and is induced by removal of all the zooids and buds. Of course, vascular budding and WBR are related to each**

other. But for the broad audience of this manuscript, we believe this would be going into too much depth on ascidian biology and be very confusing.

2. The authors present a novel model system but do not include basic information regarding its biology, life cycle, regeneration capacities, and natural fluorescent background under normal and stress conditions.

The authors should describe in detail the life cycle of *Botrylloides diegensis* under normal and stress conditions: mainly during its blastogenic cycle (budding), How long does it take for buds to complete their development? Is the development of buds synchronized? Do their life cycles include hibernation similar to *Botrylloides leachi*? We present detailed information regarding colony morphology and regeneration stages in figure 1. We give extensive detail on the biology of *B. diegensis* and the process of whole body regeneration in the introduction (page 2 line 19 to page 4 line 2). This paragraph is 465 words long – we feel this is more than sufficient. We give extensive references for readers who are interested in more detail on *Botrylloides* and WBR.

Do they have natural fluorescent background in the channels used for detection of the diverse assays (EDU; FISH; FACS)? We performed negative controls for all the stainings at all the stages. The fluorescent signals shown are much brighter than background. A negative control for Edu-staining is shown (now moved to Figure S3B) We have now included an image of FISH-negative control generated with sense-probes (Figure S2A) and the isotype control for the FACS antibody (Figure S1C). Furthermore, the FISH on MMC treated samples shows no signal for *pou3*, *h3* or *ia6* (Figure S1A), confirming that the signal obtained by antisense-FISH probes is specific and above background.

The authors should describe in detail whole body regeneration (WBR) or vascular budding under normal conditions, clearly define the characteristic of stages 1-5 (The information given in Figure 1 and introduction is very limited), We present detailed information regarding colony morphology and regeneration stages in figure 1. We give extensive detail on the biology of *B. diegensis* and the process of whole body regeneration in the introduction (page 2 line 19 to page 4 line 2). This paragraph is 465 words long – we feel this is more than sufficient. We give extensive references for readers who are interested in more detail on *Botrylloides* and WBR.

measure the effect of size and developmental stages on the ability of a fragmented vasculature embedded within a tunic to regenerate normal whole bodies. For example- can it be that the bigger fragment size in Fig3 B vs. 3A effected the WBR outcomes? Were all samples taken from the same developmental stage? Which stage? Did the zooids developed were normal?.

In *Botrylloides*, WBR is not stage dependent. Furthermore, small fragments (5 ampullae) and large fragments (larger than 1cm) alike produce over 90% of WBR

in our hands and always give rise to one dominant zooid that begins budding to regenerate the entire colony. In all the experiments, the fragments used were of comparable size (about 5 square millimeters).

This information needs to be included and images of control and injected fragments on Day 0 and along the regeneration process should be presented. Images of uninjected MMC treated fragments are presented in Figure 3A. Images of MMC treated fragments injected with IA6+ cells are presented in Figure 3B. Since only completed WBR at stage 5 was counted as a rescued outcome, we present images of samples that reach this stage. Samples that do not complete WBR are not counted and therefore not shown.

Does stress (e.g inhibitors of proliferation or specific genes) enhance fluorescent background? Arrest development? Negative controls for FISH were performed on inhibitor-treated samples the same way as for all the other experiments.

What are the % of fragments that resume regeneration by themselves when the stressor is removed?. As stated in Figure 6B, removal of the inhibitors results in 89% of WBR. For MMC treatment, as stated in the results section, 24h of treatment result in permanent loss of WBR.

Marine ascidians are known to live in highly contaminated areas (harbors) and present robust regeneration and survival capacities. Understanding the baseline regeneration abilities under random stressors is necessary before one can understand the effect of the diverse treatments and experiments done in this study.

In this work the authors use wild type *Botrylloides diegensis* colonies that did not originate from defined crosses. Using colonies where age or individuality (e.g chimera or naïve) are unknown add unmeasurable variables and noise and compromise the ability to draw conclusions from the experiments performed. We have added the following information to the methods: Growing colonies were subcloned onto independent slides, until a large number of genetically identical individual colonies was established. These genotypes are propagated in mariculture for many months. Only healthy, well growing colonies with healthy vasculature and transparent tunic were used for WBR experiments. For time point experiments, all time points were collected from subclones of the same genotype, and at least three different genotypes were used to generate averages.

Wild type marine ascidians usually contain high levels of auto-fluorescence. Since FACS and other fluorescent based assays are used in this study, the authors need to describe in detail how they handled the background and include ISH and control EDU images as well as FACS panels of unstained tissues and cells. We performed negative controls for all the stainings at all the stages. The fluorescent signals shown are much brighter than background. We have included

an image of FISH-negative control generated with sense-probes (Figure S2A) and the isotype control for the FACS antibody (Figure S1C).

3. The authors use an array of antibodies, small inhibitors, probes and primers developed against mammalians or other botryllid organism. This is a major caveat and without verifying that each of these reagents indeed detects or blocks the gene / protein it was originally designed against: conclusions should not be drawn in the way presented in this paper. **The probes and primers were actually designed against *B. diegensis* sequences, because we mainly used our own transcriptome database to find sequences (information now included in Methods). Most sequences labeled *B. leachii* in public databases are actually *B. diegensis* (Viard et al., 2019), and we have included this reference in the manuscript. We had previously misidentified our species based on these mistakes in the public databases, because genes we cloned aligned to published *B. leachii* sequences on NCBI. We have now corrected all errors regarding species names. All gene fragments cloned for FISH probes were confirmed by sequencing and analyzed by BLAST.**

One of the key gene/proteins discussed in this work is Integrin alpha 6+ (IA6) (found in the *Botryllus schlosseri* genome under the name (ITGA6).

The authors used IA6 antibodies developed against human/ mouse to sort *Botrylloides diegensis* cells without demonstrating that IA6 is expressed on these sorted cells. claiming in both the title and throughout the paper that this population expresses this protein.

Stating in the title “Integrin-alpha-6+ Stem Cells (ISCs) are responsible for whole body regeneration in an invertebrate chordate”, is misleading.

The authors need to prove that this human antibody can also recognize *Botrylloides diegensis* IA6.

Mass spec or RNA sequencing might be helpful to prove that this protein / gene is expressed on the IA6+ sorted cell population but NOT on the IA6- cell population.

The use of antibodies developed for mice and human is very problematic when working with marine organisms. Firstly, the ancestor they shared diverged millions of years ago, and evolved rapidly. Secondly, KLH (derive from marine organisms) is often used to enhance the production of commercially prepared antibodies for mice and human, which can lead to non-specific staining.

The FACS plots that present the expression of this antibody (Figure S1 C) show that 90% of *Botrylloides diegensis* blood cells express this protein in high levels (above 102), suggesting a non-specific reaction. **In fact, Figure S1C shows that it is 9.7% of total. We show by qPCR that *ia6* expression is highly enriched in IA6+ cells sorted by flow cytometry with the anti-IA6 antibody. The mRNA sequences of integrins are highly conserved between species. In a previous study, we used this same antibody to block binding of IA6 to laminin (Kassmer SH, 2015), proving that this antibody recognizes the correct epitope (IA6B1 is a receptor for laminin). We have included an image showing antibody staining of**

phosphorylated histone 3 and anti-IA6, as well as anti-POU3. Even though these antibodies do detect the proteins by fluorescence, the signal obtained by FISH is much brighter and more consistent, so we used FISH for all quantification of *ia6* and *pou3* expression.

The primers that were used in this study to estimate level of expression of several genes by qPCR (*IA6*, cyclin B, *Vasa*, *Pou3*, Notch 1, Notch 2, *Hes 1*, *Frizzled 5/8*, *Dishevelled*, *Beta catenin*, *piwi1* and *piwi 2*), and the probes that were used in this study to identify cells that express several genes (*IA6*, *Vasa*, *Histone 3* Notch 1, Notch 2 by FISH) were designed against gene sequences derived from other ascidian species. Since genes among ascidian species are highly polymorphic and evolved rapidly (compare *B. schlosseri* gene model sequences to *B. leachi* gene model sequences or *Ciona* gene model sequences) one cannot assume that they will identify the same gene.

The probes and primers were actually designed against *B. diegensis* sequences, because we mainly used our own transcriptome database to find sequences (information now included in Methods). Most sequences labeled *B. leachii* in public databases are actually *B. diegensis* (Viard et al., 2019), and we have included this reference in the manuscript. We had previously misidentified our species based on these mistakes in the public databases. We have now corrected all errors regarding species names. All gene fragments cloned for FISH probes were confirmed by sequencing and analyzed by BLAST. Probe sequences are now included in the supplement.

Furthermore, short sequences (about 20bp) designed for primers and probes can also identify other genes and when genome assembly is not available - it is very difficult to design a specific primer or probe that will be specific.

For example: I aligned the primers designed in this paper for the *IA6* against the *Botryllus schlosseri* genome (using the *B. schlosseri* genome browser at Stanford university) and found that it matched 6 different gene models including: g39337-cd109, g34568- sele, g25556- PAFAH1b1. This probe did not align to the *B. schlosseri* *IA6* gene model g37640 (called *ITGA6*). Again, this primer is designed against *B. diegensis*, not *schlosseri*. The probes and primers were actually designed against *B. diegensis* sequences, because we mainly used our own transcriptome database to find sequences (information now included in Methods). Most sequences labeled *B. leachii* in public databases are actually *B. diegensis* (Viard et al., 2019), and we have included this reference in the manuscript. We had previously misidentified our species based on these mistakes in the public databases. We have now corrected all errors regarding species names. All gene fragments cloned for FISH probes were confirmed by sequencing and analyzed by BLAST. Probe sequences are now included in the supplement.

To prevent confusion the authors should clone the genes that are key to this project from *Botrylloides diegensis* (e.g. *ITA6*, *Vasa*, Notch and Wnt), and Sanger

sequence the PCR products to verify that the primers used amplify the genes of interest. The probes and primers were actually designed against *B. diegensis* sequences, because we mainly used our own transcriptome database to find sequences (information now included in Methods). Most sequences labeled *B. leachii* in public databases are actually *B. diegensis* (Viard et al., 2019), and we have included this reference in the manuscript. We had previously misidentified our species based on these mistakes in the public databases. We have now corrected all errors regarding species names. All gene fragments cloned for FISH probes were confirmed by sequencing and analyzed by BLAST. We tested the primers used for qPCR by melting curve analysis and BLAST. Probe sequences are now included in the supplement.

4. The results and methods sections are written in a sketchy and vague way. Number of colonies / genotypes that been used as controls and for treatments are mentioned briefly in the table but not included in the methods or results sections.

Details regarding blastogenesis stages used in the control and experimental groups are missing. Size of fragments used for WBR experiments is not included, methods used to identify remains of secondary buds within the vasculature fragments following zooid and bud removal (that in many cases can be overlooked) are missing. FACS protocols do not include: how natural fluorescent background was handled, which stains were used to distinguish between live and dead cells? Was live/dead stain was used to measure viability of the cells following FACS sorting? As stated in the Methods: "A live cell gate was selected based on forward/side scatter properties, and samples were gated IA6 (CD49f)-positive or -negative based on isotype control staining (RatlgG2A-isotype-control eFluor450, Ebioscience, San Diego, CA, USA)"

FACS plots presenting natural background in the channel used for IA6 and other FACS data need to be presented. The percentage of cells stained by IA6 antibody needs to be clear See figure S1C. The percentage of IA6+ cells is 9.7%. As stated in the Methods: "A live cell gate was selected based on forward/side scatter properties, and samples were gated IA6 (CD49f)-positive or -negative based on isotype control staining (RatlgG2A-isotype-control eFluor450, Ebioscience, San Diego, CA, USA)". We have added the FACS plot for the isotype control (See Figure S1C), plots of FACS IA6 stains need to be presented in main figures (not in the supplement) and images of IA6+ and IA6- sorted populations need to be included. The FACS plots are in the supplement due to lack of space in the main figures, since they are not crucial for understanding the main figure. We have added brightfield images of sorted cell populations (See new Figure S1E).

Controls that present natural background for ISH and EDU assays need to be added. We performed negative controls for all the stainings at all the stages. The fluorescent signals shown are much brighter than background. A negative control for Edu-staining was shown (now moved to Figure S3B). We have now included an image of FISH-negative control generated with sense-probes (Figure S2A) and the isotype control for the FACS antibody (Figure S1C). Furthermore, the FISH on MMC treated samples shows no signal for *pou3*, *h3* or *ia6* (Figure S1A),

confirming that the signal obtained by antisense-FISH probes is specific and above background.

Information regarding

statistical analysis performed and how the sample size per each experiment was defined should be included. This information is included in all figure legends.

No information is provided regarding the transplantation essays. Transplantation of a small number of cells or even a single cell is the most difficult experiment in stem cell studies. It is very difficult to assess the value of this experiment without providing any description of how transplantation was accomplished, including a detailed description that include images and videos that demonstrate the ability to transplant low number of cells into small fragments of vasculature. Data regarding the instruments used to inject cells, how cells were counted, how limiting dilutions were calculated to achieve a single cell transplantation, staining used to check viability of cells following FACS sorting and the stains used to label cells before transplantation to verify successful transplantation all need to be included.

We have added the following information to the Methods: Sorted cells were collected into 3ml of filtered seawater, pelleted by centrifugation (700g, 10 minutes) and resuspended in 100ul filtered seawater. Cells were counted using a hemocytometer. Only round, bright, intact cells were counted. Cells were labeled with Hoechst 33342 and diluted in filtered seawater to the appropriate concentration. 0.1ul volume containing either 1000, 50, 10, 5 or 3 cells was injected per sample. For injections of 10 cells or less, 1ul cell suspension stained with Hoechst 33342 was pipetted onto a microscope slide and cells were counted under brightfield and fluorescence to verify the correct cell concentration. For single cell injections, cells were diluted so that 1ul contained 5 cells, and 0.1ul were injected per sample = 0.5 cells per sample. Cells were injected into ampullae of MMC treated vessel fragments with pulled glass capillary needles and an Eppendorf FemtoJet microinjector under a stereomicroscope.

Tunicates are known to live in highly contaminated areas (harbors) and present robust regeneration and survival capacities. Therefore, several important controls are needed to test their ability to resume regeneration following stress in order to understand their regeneration abilities under random stressors and understand the effect of the diverse treatments and experiments done in this study. We have added the following information to the methods: Growing colonies were subcloned onto independent slides, until a large number of genetically identical individual colonies was established. These genotypes are propagated in mariculture for many months. Only healthy, well growing colonies with healthy vasculature and transparent tunic were used for WBR experiments. For time point experiments, all time points were collected from subclones of the same genotype, and at least three different genotypes were used to generate averages.

Other points

lines 15-19: “In the majority of chordates, the ability to regenerate following a major injury is severely limited, usually resulting in scar formation. In contrast, a group of invertebrate chordate species; colonial ascidians of the genus *Botrylloides*, have been shown to regenerate whole bodies, including all tissues and organs, from small fragments of the vasculature. This process is called Whole Body Regeneration (WBR)”. **This is repeating a sentence from the manuscript. We don't understand if this is a question or a comment?**

Add references to support claims. **Relevant references have been cited where appropriate.**

Lines 55-57: “The source of the new bodies is a specialized region of the body wall of the parental zooid called the peribranchial epithelium, and this process of asexual reproduction is called palleal budding (Kassmer et al., 2018).” **This is repeating a sentence from the manuscript. We don't understand if this is a question or a comment?**

Authors need to read and cite studies done by Manni lab (Padova university) that describe budding in *Botryllus schlosseri*. Buds do not originate in the body wall but in the peribranchial epithelium which is connected to the branchial sac. **We have changed this sentence to: “The source of the new bodies is a specialized region of epithelium surrounding the branchial sac of the parental zooid called the peribranchial epithelium, and this process of asexual reproduction is called palleal budding (Brown and Swalla, 2012; Kassmer et al., 2018; Kawamura et al., 2008; Manni et al., 2014; Oka, 1959).”**

“In addition to palleal budding, which occurs normally”

Authors need to describe in detail or refer to previous studies (if any) that describe palleal budding in *Botrylloides diegensis*. What are the normal budding stages of development in *Botrylloides diegensis*? Is it synchronized along the colony, how long does it takes under specific temperatures? Do they hibernate like other botryllid species? **The process of palleal budding is identical in within the Botryllinae (Alie et al., 2018; Brown and Swalla, 2012; Oka, 1959). We have added additional references. We have observed hibernation of *Botrylloides diegensis* during sudden changes in temperature when cultured in a facility supplied with raw seawater. This is not surprising given that many species reported to be *B. leachii* in the literature are acutally *B. diegensis*. Since hibernation is not the subject of this study, we did not include these observations but are in the process of studying the relationship between WBR and recovery from hibernation.**

Lines 62-80: “*B. diegensis* can also use the alternative WBR pathway to produce a zooid following injury. WBR occurs in the vasculature, and is initiated by

surgical removal of all zooids, or separation of small vascular fragments from the rest of the colony (Figure 1B). WBR progresses through distinct visual stages. For the first 24 hrs following surgical ablation, the blood vessels undergo regression and remodeling (stage 1, Figure 1B). During this time, blood flow, usually powered by the hearts of each zooid, continues and is driven by contractions of the remaining ampullae (Blanchoud et al., 2017). During the next 48 hrs, the blood vessels continue to remodel and form a dense, contracted network (stage 2, Figure 1B). Within this dense, highly pigmented vascular tissue, an opaque mass of non-pigmented cells becomes apparent; creating a clear area that is the presumptive site of bud development (stage 3, arrow in Figure 1B). The mass of cells next forms into a hollow, blastula-like epithelial sphere. The vascular epithelium then wraps itself around this sphere, leading to the formation of a distinct visible double vesicle. Over the next 48h, the inner vesicle increases in size (Stage 4, Figure 1B), while undergoing a series of invaginations and evaginations that lead to organogenesis and the eventual regeneration of a zooid. WBR is defined as complete when the new zooid is actively filter-feeding, and occurs within a range of 7-10 days (stage 5; Figure 1B). The zooid immediately commences normal palleal budding, and the colony regrow”.

This section which discussed WBR in *B. diegensis* includes new information and should be part of the results. It should include the number of fragments tested and size and developmental stage used for WBR. Are the statements of “blastula like epithelial sphere” and “vascular epithelium wraps itself around this sphere” based on in vivo images? Or histological sections? If this is based on histological sections they should be included. If not, these terms cannot be used. **The morphological stages of whole body regeneration have been described in detail in previous studies, and we give those references. Therefore, these are not new results. We are merely confirming that we can reproduce the findings of others and that our animals undergo WBR in the same manner as described previously. We have added the following statement: “WBR can be induced in fragments as small as 5 ampullae, and is not dependent on the stage of asexual reproduction of the colony.” We show the morphology of the double vesicle in Figure 1B, and in the cryosections in Figure S3B, the morphology of the two epithelial layers is clearly evident. Figure 2A shows early stages of double vesicle formation, and the two layers are evident there as well. Therefore, we feel that using this terminology is very correct.**

Lines 83-86 “WBR has been studied in three different species of botryllid ascidians, and in all cases a population of cells with an undifferentiated appearance, termed hemoblasts, have been suggested to initiate this regenerative process (Brown et al., 2009a)”.

Please include and cite other works on vascular budding or WBR in botryllid ascidian including: Sabbadin et al. 1975; Miyamoto and Freeman, 1970; Rinkevich et al. 1995; Voskoboynik et al. 2007. **We are strictly distinguishing**

between vascular budding that occurs as part of normal colony development - as in *B. primigenus* and *B. tuberatus* – and whole body regeneration, which is induced by removal of all zooids and buds (Kassmer et al., 2019). *Botrylloides leachii*, *Botrylloides diegensis* and *Botrylloides violaceus* all undergo WBR, whereas *B. primigenus* and *B. tuberatus* undergo vascular budding. *B. schlosseri* does neither. We have edited this section to talk about *Botrylloides* only.

Lines 88-92: “In *Botrylloides violaceus*, 15-20 small hemoblasts that express Piwi protein have been shown to aggregate under the epidermis of a blood vessel during early WBR (Oka, 1959). These cells can be detected during the early vesicle stage and occasionally within the epithelium of a vesicle, suggesting that they play a role in regeneration (Brown et al., 2009a). In a related species, *Botrylloides leachii*, Blanchoud et al. showed an increase in the population of hemoblasts very early after injury during WBR (Blanchoud et al., 2017)”.

Please include the work of Rinkevich et al. 2007 Plos Biology on *Botrylloides leachii* WBR We have included this reference

Lines 97-101: “In several species of colonial ascidians, the blood contains self-renewing, lineage restricted germline stem cells that migrate to newly developing buds during repeated cycles of asexual reproduction, where they give rise to eggs and testes (Brown et al., 2009b; Carpenter et al., 2011; Kawamura and Sunanaga; Laird et al., 2005; Rodriguez et al., 2016; Sunanaga et al., 2010; Sunanaga et al., 2006)”.

Up to date self-renewal of stem cells was proven only in *B. schlosseri* (Laird et al. 2005) and not in any other botryllid species. Please do not mislead the readers and suggest that it was proven in other botryllid species. We have edited this section to clearly state that germline stem cells were identified in *B. schlosseri*, and edited the references. It is true that germline parasitism has been proven only in *B. schlosseri*, and this parasitism has been used to formally demonstrate that long-lived germline stem cells are present in the blood of this species. While these experiments have not yet been repeated in *Botrylloides*, the fact that each asexual generation and colonies generated from WBR have a complete germline is strong evidence that long-lived germline stem cells that settle in gonadal niches of asexual generations exist in all Botryllinae.

Authors should also add the original works done on *Botryllus schlosseri* chimeras which were the first to demonstrate the presence stem cells in the of blood of this organism : Sabbadin and Zaniolo, Stoner and Weissman 1966; Stonner et al. 1998; Voskoboynik et al. 2008; Rinkevich et al. 2013 Since this study is focused on WBR in *Botrylloides diegensis*, we cannot include so much detail on germline development and germ cell migration in *Botryllus schlosseri* in the introduction. For interested readers, gonad development germ cell migration and germline parasitism in *B. schlosseri* has been reviewed in (Rodriguez et al., 2017). We have now included Rinkevich et al. 2013 because of the piwi knockdown.

Thank you for reminding us!

Lines 101-105: In *Botryllus schlosseri*, these germline stem cells can be enriched by flow cytometry using Integrin- α -6 (IA6) as a marker (Kassmer SH, 2015) and express *piwi* as well as other genes associated with germ cells, such as *vasa*, and *pumilio* (Brown et al., 2009b; Langenbacher and De Tomaso, 2016; Sunanaga et al., 2006).

Please highlight the limitation of the use of antibodies designed against mammalian cells in these studies **We show by qPCR that *ia6* expression is highly enriched in IA6+ cells sorted by flow cytometry with the anti-IA6 antibody. The mRNA sequences of integrins are highly conserved between species. In a previous study, we used this same antibody to block binding of IA6 to laminin (Kassmer SH, 2015), proving that this antibody recognizes the correct epitope (IA6B1 is a receptor for laminin).**

and add the work done by Rinkevich et al. 2013 (Dev Cell) which used classic transplantation experiments and genotyping to demonstrate that the cell islands in *Botryllus schlosseri* are enriched with germline stem cells. **Since this study is focused on WBR in *Botrylloides diegensis*, we cannot include so much detail on germline development and germ cell migration in *Botryllus schlosseri* in the introduction. For interested readers, gonad development germ cell migration and germline parasitism in *B. schlosseri* has been reviewed in (Rodriguez et al., 2017).**

References relevant to the responses:

- Alie, A., Hiebert, L.S., Simion, P., Scelzo, M., Prunster, M.M., Lotito, S., Delsuc, F., Douzery, E.J.P., Dantec, C., Lemaire, P., *et al.* (2018). Convergent Acquisition of Nonembryonic Development in Styelid Ascidiaceans. *Mol Biol Evol* 35, 1728-1743.
- Brown, F.D., and Swalla, B.J. (2012). Evolution and development of budding by stem cells: ascidian coloniality as a case study. *Dev Biol* 369, 151-162.
- Carpenter, M.A., Powell, J.H., Ishizuka, K.J., Palmeri, K.J., Rendulic, S., and De Tomaso, A.W. (2011). Growth and long-term somatic and germline chimerism following fusion of juvenile *Botryllus schlosseri*. *Biol Bull* 220, 57-70.
- Kassmer, S.H., Nourizadeh, S., and De Tomaso, A.W. (2018). Cellular and molecular mechanisms of regeneration in colonial and solitary Ascidiaceans. *Dev Biol*.
- Kassmer, S.H., Nourizadeh, S., and De Tomaso, A.W. (2019). Cellular and molecular mechanisms of regeneration in colonial and solitary Ascidiaceans. *Dev Biol* 448, 271-278.
- Kassmer SH, R.D., Langenbacher AD, Bui C, De Tomaso AW (2015). Migration of germline progenitor cells is directed by sphingosine-1-phosphate signalling in a basal chordate. *Nature Communications* 6.
- Kawamura, K., Sugino, Y., Sunanaga, T., and Fujiwara, S. (2008). Multipotent epithelial cells in the process of regeneration and asexual reproduction in colonial tunicates. *Dev Growth Differ* 50, 1-11.

Manni, L., Gasparini, F., Hotta, K., Ishizuka, K.J., Ricci, L., Tiozzo, S., Voskoboynik, A., and Dauga, D. (2014). Ontology for the asexual development and anatomy of the colonial chordate *Botryllus schlosseri*. *PLoS One* 9, e96434.

Oka, H.W.H. (1959). VASCULAR BUDDING IN BOTRYLLOIDES. *The Biological Bulletin* 117, 340-346.

Rinkevich, Y., Rosner, A., Rabinowitz, C., Lapidot, Z., Moiseeva, E., and Rinkevich, B. (2010). Piwi positive cells that line the vasculature epithelium, underlie whole body regeneration in a basal chordate. *Dev Biol* 345, 94-104.

Rodriguez, D., Kassmer, S.H., and De Tomaso, A.W. (2017). Gonad development and hermaphroditism in the ascidian *Botryllus schlosseri*. *Mol Reprod Dev* 84, 158-170.

Viard, F., Roby, C., Turon, X., Bouchemousse, S., and Bishop, J. (2019). Cryptic Diversity and Database Errors Challenge Non-indigenous Species Surveys: An Illustration With *Botrylloides* spp. in the English Channel and Mediterranean Sea. *Front Mar Sci*.

REVIEWERS' COMMENTS:

Reviewer #1 (Remarks to the Author):

I find the current version of the manuscript much improved. The authors have provided additional supplemental figures with relevant controls and primary data to support their results. The authors have also carefully responded to all my initial concerns in their rebuttal.

It came to my surprise to read that most of the *B. leachi* sequences in public databases (presumably GenBank) correspond to *B. diegensis*. Thus, I find it extremely important to clearly state this issue in the manuscript (either in the results or in the methods). I suggest that the authors clearly state all cases of genes in their study that were verified to be identical to the *B. leachi* genes of public databases.

Some minor comments to the newly revised version include:

- (1) All supplemental material needs to be properly cited in the main text of the manuscript.
- (2) Fig. S1E, cells do not appear to be in the same magnification. Please check. Also, think of putting a little higher magnification to get a better detail of the individual cells.
- (3) Include proper numbers or labels to the Supplemental material that includes all the sequences (shown after the supplemental figures). Perhaps give each a title and a label such as 'Supplemental file 1' (to distinguish from 'Supplemental figure').
- (4) In the supplemental file "Pou3-phylogenetic analysis – protein sequences", in the title mention that this data was used to generate the phylogeny and is complementary to Supplement S1A.
- (5) In the supplemental file "POU3 ORF 1473 (assembled from RNA-Sequencing) aa478", in the title specify whether this sequence corresponds to *B. diegensis* POU3 (and include genbank number).
- (6) In the supplemental file "*B. leachii* Piwi mRNA sequences:". Why do you write the *B. leachi* Piwi sequences here, and do not include the *B. diegensis* Piwis???

Typos and other minor corrections:

Page 4, line 23: End sentence with '....has been reviewed [22]'. Remove 'in [22]'

Page 6, line 3: Replace 'piw1 and piw2' with 'piwi1 and piwi2'

Page 9, line 20: 'Stage' should not be capitalized.

Page 13, line 2: Change 'it's' for 'its'

Page 15, line 15: Revise sentence. Change '...are propagates...' for '...continue to propagate...'

Page 15, line 16: Replace 'healthy vasculature and' with 'expanded vasculature and ampullae, and a'

Page 17, line 2: 'Supplemental file "primer sequences" should be better organized and given a Supplemental file number so it is easy to find.

Page 17, line 18: Replace 'generate averages' with 'calculate the averages'.

Page 17, line 22: Replace 'has been' with 'was'.

Page 19, line 1: *Ciona* should be in italics.

Page 19, line 3: 'Supplemental file "primer sequences" should be better organized and given a Supplemental file number so it is easy to find (see comment above for Page 17, line 2).

Reviewed by,
Federico D. Brown

Reviewer #3 (Remarks to the Author):

This paper is much improved from its previous version and most issues are covered in response to

review. However, a few changes are very important to make to match the title and summary with the data. All are detailed below. But, just to emphasize, the evidence that the Ia6+ cells are stem cells is incomplete. While it is hard to do the studies that would satisfy stem cell reviewers, the data are good enough to call the cells 'candidate' stem cells. Full longterm tracing of transplanted cells, along with re-isolation and transplantation of Ia6+ cells from primary recipients to secondary hosts with the same WBR and regeneration of Ia6+ cells are not included here. That would be required to change candidate stem cells to stem cells.

REVIEWERS' COMMENTS:

Reviewer #1 (Remarks to the Author):

I find the current version of the manuscript much improved. The authors have provided additional supplemental figures with relevant controls and primary data to support their results. The authors have also carefully responded to all my initial concerns in their rebuttal.

It came to my surprise to read that most of the *B. leachi* sequences in public databases (presumably GenBank) correspond to *B. diegensis*. Thus, I find it extremely important to clearly state this issue in the manuscript (either in the results or in the methods). I suggest that the authors clearly state all cases of genes in their study that were verified to be identical to the *B. leachi* genes of public databases.

Some minor comments to the newly revised version include:

- (1) All supplemental material needs to be properly cited in the main text of the manuscript. **done**
- (2) Fig. S1E, cells do not appear to be in the same magnification. Please check. Also, think of putting a little higher magnification to get a better detail of the individual cells.

We edited these images to show higher magnification.

(3) Include proper numbers or labels to the Supplemental material that includes all the sequences (shown after the supplemental figures). Perhaps give each a title and a label such as 'Supplemental file 1' (to distinguish from 'Supplemental figure'). **All supplementary material has to be in one PDF file. We have edited the headers to help organize the material.**

(4) In the supplemental file "Pou3-phylogenetic analysis – protein sequences", in the title mention that this data was used to generate the phylogeny and is complementary to Supplement S1A. **done**

(5) In the supplemental file "POU3 ORF 1473 (assembled from RNA-Sequencing) aa478", in the title specify whether this sequence corresponds to *B. diegensis* POU3 (and include genbank number). **We do not have genbank numbers. All sequences are *B. diegensis* and labeled accordingly.**

(6) In the supplemental file "B. leachii Piwi mRNA sequences:". Why do you write the *B. leachi* Piwi sequences here, and do not include the *B. diegensis* Piwis???? **thank you for catching this mistake. Everything is *B. diegensis*.**

Typos and other minor corrections:

Page 4, line 23: End sentence with '...has been reviewed [22]'. Remove 'in [22]' **changed "in" to "previously" to generate a grammatically correct sentence.**

Page 6, line 3: Replace 'piw1 and piw2' with 'piwi1 and piwi2' **done**

Page 9, line 20: 'Stage' should not be capitalized. **check**

Page 13, line 2: Change 'it's' for 'its' **done**

Page 15, line 15: Revise sentence. Change '...are propagates...' for '...continue to propagate...' **done**

Page 15, line 16: Replace 'healthy vasculature and' with 'expanded vasculature and ampullae, and a' **done**

Page 17, line 2: 'Supplemental file "primer sequences" should be better organized and given a Supplemental file number so it is easy to find.

Page 17, line 18: Replace 'generate averages' with 'calculate the averages'. **done**

Page 17, line 22: Replace 'has been' with 'was'. **done**

Page 19, line 1: *Ciona* should be in italics. **done**

Page 19, line 3: 'Supplemental file "primer sequences" should be better organized and given a Supplemental file number so it is easy to find (see comment above for Page 17, line 2). **All Supplementary Information have to be together in one PDF file. But we have tried to better organize these items and label them clearly.**

Reviewed by,
Federico D. Brown

Reviewer #3 (Remarks to the Author):

This paper is much improved from its previous version and most issues are covered in response to review. However, a few changes are very important to make to match the title and summary with the data. All are detailed below. But, just to emphasize, the evidence that the Ia6+ cells are stem cells is incomplete. While it is hard to do the studies that would satisfy stem cell reviewers, the data are good enough to call the cells 'candidate' stem cells. Full longterm tracing of transplanted cells, along with re-isolation and transplantation of Ia6+ cells from primary recipients to secondary hosts with the same WBR and regeneration of Ia6+ cells are not included here. That would be required to change candidate stem cells to stem cells.

In this manuscript the authors studied whole body regeneration (WBR) in a wild type *Botrylloides diegensis* species. They used antibodies developed against human/mice integrin alpha-6 (ITGA6) and flow cytometry to isolate a population of cells and perform diverse transplantation experiments to measure their ability to proliferate. They designed probes and primers to detect several genes based on their sequence in other botryllid species (*Botryllus schlosseri* and *Botrylloides leachii*), and used them to measure levels of expression (qPCR) and sites of expression (ISH) of these genes in *Botrylloides diegensis*. By combining EDU tracing experiments and ISH they found that the sorted cell populations (cells stained by the mammalian antibodies directed against integrin alpha-6) include a highly proliferate population of small cells that are also positive to the *B. schlosseri* and *Botrylloides leachii* probes aimed to detect integrin alpha-6 (ITGA6), *pou3* and *vasa* (DDX4). By treating vasculature fragments taken from *Botrylloides diegensis* (undefined size, undefined stage) with mitomycin C they prevented whole body regeneration and then succeeded in inducing whole body regeneration by transplanting thousands of cells from non-treated individuals, 50 ITGA6+ sorted cells, and even 1 ITGA6+ cells. FACS plots that present the expression of this antibody suggest that at least 90% of the *Botrylloides diegensis* blood cells express this protein in high levels (above 10² Figure S1 C).

Developing new model organisms to study regeneration is important and can add to our basic knowledge about cellular and molecular mechanisms of regeneration. The authors performed an impressive set of experiments and tests aiming to study mechanisms of regeneration in a colonial chordate species for which very little information is available about its life cycle and biology. However, several major issues need to be addressed before one can assess the real impact of this study and publish this paper.

1. Stem cells are self-renewing and multipotent cells. This study does not include experiments that prove self-renewal or multipotency and therefore the use of the term stem cells throughout the paper is inappropriate. When discussing previous work on candidate stem cells in diverse botryllid species, the authors should distinguish between papers that indeed found and verified stem cells (e.g. Laird et al. 2005) versus papers

that only point to candidate stem cell populations (e.g Brown et al. 2009b).

To prevent confusion, studies that focus on expression of genes associated with stemness without experiments that clearly demonstrate self-renewal and multipotency should not be discussed and reviewed as studies that succeeded in isolating stem cells.

Ia6+ cells are constantly cycling in a healthy colony– as shown by expression of cyclin b and histone 3, and the fact that G2/M cells express ia6 and pou3. The fact that ia6+pou3+ cells are constantly cycling on a healthy colony and never exhaust shows that they self renew. We have added an experiment showing pou3 expression in Edu-positive cells after transplantation of Edu-labeled Ia6+/pou3+ cells (FigureS3C). Pou3+ edu+ cells that are not in foci or double vesicles (pink arrows) are derived from self-renewing cells that were transplanted. We have added new text discussing self-renewal to the results and discussion.

It would be more appropriate to identify the Ia6+ cells as candidate stem cells. Secondary transplantation studies with purified Ia6+ cells derived from primary transplant recipients is the current standard for claiming stem cells. Proliferation of phenotypically marked cells without lineage tracing or small number of cells in the primary transplant with secondary transplants is indirect and inferred rather than direct and demonstrated. It will not hurt this paper to leave it at candidate stem cells until more definitive studies are done.

Response: We have changed it to “candidate stem cells”.

2. The authors present a novel model system but do not include basic information regarding its biology, life cycle, regeneration capacities, **We disagree. We present detailed information regarding colony morphology and regeneration stages in figure 1. We give extensive detail on the biology of B. diegensis and the process of whole body regeneration in the introduction (page 2 line 19 to page 4 line 2). This paragraph is 465 words long – we feel this is more than sufficient. We give extensive references for readers who are interested in more detail on Botrylloides and WBR.**

Has no other lab described their life cycle and WBR? This could be handled by a reference.

and stress conditions, information that is crucial to assess the outcomes of the experiments (mainly experiments that induce stress and outcomes that are detected by fluorescent product). **We performed negative controls for all the stainings at all the stages. The fluorescent signals shown are much brighter than background. We have included an image of FISH-negative control generated with sense-probes (Figure S2A) and the isotype control for the FACS antibody (Figure S1C).**

AUTHORS SHOULD ALSO PRESENT FACS PLOTS THAT PRESENT NATURAL FLUORSCENT BACKGROUND ON THE 450 CHANNEL, FOR SAMPLES IDENTICAL TO THE ONES USED FOR IT6 STAINING WITHOUT ANY STAIN. BOTRYLLOIDES BLOOD CELLS PRESENT HIGH NATURAL FLUORESCENT BACKGROUND ON THE 450 (FITC; GFP) CHANNEL.

Response: For this exact reason we used the violet channel (Ex 405, Em436). We have included FACS plots showing unstained, isotype control and stained samples.

The authors use *Botrylloides diegensis* colonies collected in the wild; thus age, individuality (naïve/ chimeras) and history are unknown and cannot be determined. This is adding unmeasurable variables to the study and needs to be included in consideration when designing the experiments (high numbers of repeats) and when drawing conclusions. **Only healthy, well growing colonies were used for the experiments presented here. Controls were subclones from the same colonies as the experimental samples, and therefore the same age and genotype. Time points were taken from subclones of the same genotype and repeated on at least 3 different genotypes per experiment. We have added this information to the methods. We have never observed any significant changes in any sample collected. WBR is incredibly robust in *B. diegensis*.**

3. The authors use an array of antibodies, small inhibitors, probes and primers developed against mammals or other botryllid organism. This is a major caveat and without verifying that each of these reagents indeed detects or blocks the gene / protein it was originally designed against, conclusions should not be drawn.

We validated the antibody used for FACS with the appropriate controls and showed that it enriches for IA6+ cells (qPCR in Figure 1C). The probes and primers were actually designed against *B. diegensis* sequences, because we mainly used our own transcriptome database to find sequences (information now included in Methods). Most sequences labeled *B. leachii* in public databases are actually *B. diegensis* (Viard et al., 2019), and we have included this reference in the manuscript. We had previously misidentified our species based on these mistakes in the public databases. We have now corrected all errors regarding species names. All gene fragments cloned for FISH probes were confirmed by sequencing and the probe sequences are now included in the supplement. We used control drugs to show specificity of the inhibitors, and show a specific effect of the inhibitors on the pou3+ stem cells (Figure 6).

Unfortunately this doesn't answer the question. Many monoclonal antibodies have sequence unrelated cross reactivities. If the key antibody to intA6 actually binds in *diegensis* the homologue of human intA6, show the likelihood by comparing the aa sequence of both in a supplementary figure to make your case. Or immunoprecipitate the protein to show some sequence similarity. Without that, you should call these human antiA6 antigen reactive markers on a subset of *diegensis* cells.

Response: In a previous study, we had shown that this antibody blocks binding of the ITGA6:ITGB4 laminin receptor to laminin. In addition, we show that cells in the Integrin-alpha-6-negative gate are not enriched for expression of IA6, whereas cells in the Integrin-alpha-6-positive gate are highly enriched. We have also included an alignment of human Integrin-alpha-6 with the Integrin-alpha-6 sequence from *Botrylloides* in the supplement. Further confirmation that IA6 is expressed in our candidate stem cells comes from FISH showing expression of integrin alpha 6 mRNA within pou3+ cycling cells (Figures 1 and 2) and by showing expression of IA6 and POU3 proteins in the same cell by immunofluorescence (Supplementary Figure 4).

4. The results and methods sections are written in a sketchy and vague way limiting the ability of other researchers to reproduce this work in the future. Important information regarding transplantation experiments and FACS protocols are lacking.

If we knew exactly what information we should add? What specific details are missing? We strongly feel that the methods contain more than enough detail for other researchers to reproduce our findings, but we have now added a lot of additional detail to the Methods section. Anyone is always welcome to contact us with questions regarding the methods and we can provide help and pass along detailed protocols.

1 Integrin-alpha 6 and cell cycle flow cytometry

2 Genetically identical, **stage matched samples were pooled-WHICH STAGE?**

Response: Stage matched means stage 1 samples were pooled with other stage 1 samples, etc. ,

and blood was isolated by

3 cutting blood vessels with a razor blade and gentle squeezing with the smooth side of a
4 syringe plunger. Blood was diluted with filtered seawater and passed through 70 µm and
5 40 µm cell strainers. **Anti-Human/Mouse-CD49f-eFluor450** (Ebioscience, San Diego, CA,
6 USA, cloneGoH3) **was added at a dilution of 1/50 and incubated on ice for 30 min and**
7 **washed with filtered seawater.**

For cell cycle sorting, Vybrant Dye Cycle Ruby Stain

8 (Thermo Fisher Scientific, Waltham, MA, USA) was added at a dilution of 1/100 and
9 incubated for 30 minutes at room temperature. Fluorescence activated cell sorting
10 (FACS) was performed using a FACSAria (BD Biosciences, San Jose, CA, USA) cell
11 sorter. A live cell gate was selected based on forward/side scatter properties, and
12 samples were gated IA6 (CD49f)-positive or -negative based on isotype control staining
13 (RatIgG2A-isotype-control eFluor450, Ebioscience, San Diego, CA, USA). Analysis was
14 performed using FACSDiva software (BD Biosciences, San Jose, CA, USA). Cells were
15 sorted using high-pressure settings and a 70 µm nozzle and collected into filtered
16 seawater.

No information is provided regarding natural fluorescent background under normal and stress conditions. We performed negative controls for all the stainings at all the stages. The fluorescent signals shown are much brighter than background. We have included an image of FISH-negative control generated with sense-probes (Figure S2A) and the isotype control for the FACS antibody (Figure S1C).

No information is provided regarding the size of fragments used for WBR and the stage of development tested. This information is crucial since it's been shown that WBR is dependent on fragment size and stage in botryllid ascidians (e.g. Rinkevich et al. 1995 PNAS; Voskoboynik et al. 2007 FASEB). In Botrylloides, WBR is not stage dependent. We have included this now in the introduction. Furthermore, small fragments (5 ampullae) and large fragments (larger than 1cm) alike produce over 90% of WBR in our hands and always give rise to one dominant zooid that begins budding to regenerate the

entire colony. In all the experiments, the fragments used were of comparable size (about 5 square millimeters).

Images that follow control and transplanted fragments from day 1 until the end of the experiments need to be presented. No information is provided regarding the transplantation assays, making it very difficult to assess the value of this experiment. Transplantation of a small number of cells or even a single cell is the most difficult experiment in the stem cell field, therefore a detailed description that includes images and videos demonstrating the ability to transplant low number of cells into small fragments of vasculature is imperative. We have added the following information to the Methods: Sorted cells were collected into 3ml of filtered seawater, pelleted by centrifugation (700g, 10 minutes) and resuspended in 100ul filtered seawater. Cells were counted using a hemocytometer. Only round, bright, intact cells were counted. Cells were labeled with Hoechst 33342 and diluted in filtered seawater to the appropriate concentration. 0.1ul volume containing either 1000, 50, 10, 5 or 3 cells was injected per sample. For injections of 10 cells or less, 1ul cell suspension stained with Hoechst 33342 was pipetted onto a microscope slide and cells were counted under brightfield and fluorescence to verify the correct cell concentration. For single cell injections, cells were diluted so that 1ul contained 5 cells, and 0.1ul were injected per sample = 0.5 cells per sample. Cells were injected into ampullae of MMC treated vessel fragments with pulled glass capillary needles and an Eppendorf FemtoJet microinjector under a stereomicroscope.

The authors need to include Images that follow data regarding the FACS sorting and the dyes used to label cells before transplantation to verify successful transplantation. We have added the following information to the Methods: Sorted cells were collected into 3ml of filtered seawater, pelleted by centrifugation (700g, 10 minutes) and resuspended in 100ul filtered seawater. Cells were counted using a hemocytometer. Only round, bright, intact cells were counted. Cells were labeled with Hoechst 33342 and diluted in filtered seawater to the appropriate concentration. 0.1ul volume containing either 1000, 50, 10, 5 or 3 cells was injected per sample. For injections of 10 cells or less, 1ul cell suspension stained with Hoechst 33342 was pipetted onto a microscope slide and cells were counted under brightfield and fluorescence to verify the correct cell concentration. For single cell injections, cells were diluted so that 1ul contained 5 cells, and 0.1ul were injected per sample = 0.5 cells per sample. Cells were injected into ampullae of MMC treated vessel fragments with pulled glass capillary needles and an Eppendorf FemtoJet microinjector under a stereomicroscope.

The authors need to include methods and measures used to assess viability in the fragments treated and normality of developing buds like histological sections that follow development. If the MMC treated fragments were not viable, rescue by cell transplantation would not work. In order to accurately quantify the success of the rescue, only fragments that reach stage 5 (complete filter-feeding body that begins pallear budding) were counted. All fragments that reach stage 5 eventually give rise to entire new colonies by asexual reproduction.

Major points:

1. Stem cells are self-renewing and multipotent cells. The paper does not include experiments that prove self-renewal or multipotency, and therefore the use of the term stem cells throughout the paper is inappropriate.

The assay used in this study is showing complete somatic reconstitution (i.e., WBR), and this is clearly shown using limit dilution rescue and lineage tracing. For self-renewal, this is also clearly demonstrated in nearly every figure. Ia6+ cells are constantly cycling in a healthy colony– as shown by expression of cyclin b and histone 3, and the fact that G2/M cells express ia6 and pou3 (Figure 2, Figure 5). The fact that ia6+pou3+ cells are constantly cycling in a healthy colony and never exhaust for the life time of the animal shows that they self-renew. We have added an experiment showing pou3 expression in Edu-positive cells after transplantation of Edu-labeled Ia6+/pou3+ cells (FigureS3C). Pou3+ edu+ cells that are not in foci or double vesicles (pink arrows) are derived from self-renewing cells that were transplanted. We have added new text discussing self-renewal to the results and discussion.

As noted above, cell division of a cell subset does not prove that the cells dividing a day or a week or a month later were derived from dividing cells of that subset. Therefore it is wise to call them candidate stem cells rather than stem cells until the evidence is stronger

Response: ok

. When discussing works done in the past on candidate stem cells in diverse botryllid species the authors should distinguish between papers that indeed found and validated stem cells and between papers that only point to candidate stem cell populations. Studies that focus on expression of genes associated with stemness without experiments that clearly demonstrate self-renewal and multipotency (transplantation and genotyping) should not be discussed and reviewed as studies that isolated stem cells. We have edited the references and text referring to them to make sure everything is accurate.

Throughout the paper (introduction, discussion) the authors should include comprehensive work done on Botryllus schlosseri chimeras which already prove that stem cell originating in cells circulating in the vasculature are contributing to germline and somatic tissues in the developing buds. Just to name a few: Sabbadin and Zaniolo 1979 J. Exp. Zool; Stoner and Weissman, 1996 PNAS; Stoner et al. 1999 PNAS; Voskoboynik et al. 2008 Cell Stem Cell; Rinkevich et al. 2013 Dev. Cell; Rosental et al. 2018 Nature. First, we are not studying Botryllus schlosseri. Second, the two mobile progenitors in B. schlosseri are for germline and hematopoietic cells, which was already shown in Laird, et al. (2005) and Carpenter et al., (2011). There is no direct evidence at all that mobile somatic progenitors give rise to zooid tissues. This is irrelevant to this study.

The authors also failed to cite original work describing vascular budding / WBR in botryllid species including: Sabbadin et al. 1975 Dev Biol; Rinkevich et al. 1995 PNAS; Voskoboinik et al. 2007 FASEB. Also, the phrase “whole body regeneration”-WBR- was coined by Rinkevich et al in 1995. It is important to distinguish between vascular budding, which naturally occurs in Botryllus Primigenus (and some other species of colonial ascidians) as part of normal asexual reproduction. Here, we are studying whole body regeneration which occurs in species of the genus Botrylloides and is induced by removal of all the zooids and buds. Of course, vascular budding and WBR are related to each other. But for the broad audience of this manuscript, we believe this would be going into too much depth on ascidian biology and be very confusing.

2. The authors present a novel model system but do not include basic information regarding its biology, life cycle, regeneration capacities, and natural fluorescent background under normal and stress conditions.

The authors should describe in detail the life cycle of Botrylloides diegensis under normal and stress conditions: mainly during its blastogenic cycle (budding), How long does it take for buds to complete their development? Is the development of buds synchronized? Do their life cycles include hibernation similar to Botrylloides leachi? We present detailed information regarding colony morphology and regeneration stages in figure 1. We give extensive detail on the biology of B. diegensis and the process of whole body regeneration in the introduction (page 2 line 19 to page 4 line 2). This paragraph is 465 words long – we feel this is more than sufficient. We give extensive references for readers who are interested in more detail on Botrylloides and WBR.

Do they have natural fluorescent background in the channels used for detection of the diverse assays (EDU; FISH; FACS)? We performed negative controls for all the stainings at all the stages. The fluorescent signals shown are much brighter than background. A negative control for Edu-staining is shown (now moved to Figure S3B) We have now included an image of FISH-negative control generated with sense-probes (Figure S2A) and the isotype control for the FACS antibody (Figure S1C). Furthermore, the FISH on MMC treated samples shows no signal for pou3, h3 or ia6 (Figure S1A), confirming that the signal obtained by antisense-FISH probes is specific and above background.

The authors should describe in detail whole body regeneration (WBR) or vascular budding under normal conditions, clearly define the characteristic of stages 1-5 (The information given in Figure 1 and introduction is very limited), We present detailed information regarding colony morphology and regeneration stages in figure 1. We give extensive detail on the biology of B. diegensis and the process of whole body regeneration in the introduction (page 2 line 19 to page 4 line 2). This paragraph is 465 words long – we feel this is more than sufficient. We give extensive references for readers who are interested in more detail on Botrylloides and WBR.

measure the effect of size and developmental stages on the ability of a fragmented vasculature embedded within a tunic to regenerate normal whole bodies. For example- can it be that the bigger fragment size in Fig3 B vs. 3A effected the WBR outcomes? Were all samples taken from the same developmental stage? Which stage? Did the zooids developed were normal?.

In *Botrylloides*, WBR is not stage dependent. Furthermore, small fragments (5 ampullae) and large fragments (larger than 1cm) alike produce over 90% of WBR in our hands and always give rise to one dominant zooid that begins budding to regenerate the entire colony. In all the experiments, the fragments used were of comparable size (about 5 square millimeters). -“small fragments (5 ampullae) and large fragments (larger than 1cm) alike produce over 90% of WBR in our hands”,

This information needs to be included and images of control and injected fragments on Day 0 and along the regeneration process should be presented. Images of uninjected MMC treated fragments are presented in Figure 3A. Images of MMC treated fragments injected with IA6+ cells are presented in Figure 3B. Since only completed WBR at stage 5 was counted as a rescued outcome, we present images of samples that reach this stage. Samples that do not complete WBR are not counted and therefore not shown.

Does stress (e.g inhibitors of proliferation or specific genes) enhance fluorescent background? Arrest development? Negative controls for FISH were performed on inhibitor-treated samples the same way as for all the other experiments.

What are the % of fragments that resume regeneration by themselves when the stressor is removed?. As stated in Figure 6B, removal of the inhibitors results in 89% of WBR. For MMC treatment, as stated in the results section, 24h of treatment result in permanent loss of WBR.

Marine ascidians are known to live in highly contaminated areas (harbors) and present robust regeneration and survival capacities. Understanding the baseline regeneration abilities under random stressors is necessary before one can understand the effect of the diverse treatments and experiments done in this study.

In this work the authors use wild type *Botrylloides diegensis* colonies that did not originate from defined crosses. Using colonies where age or individuality (e.g chimera or naïve) are unknown add unmeasurable variables and noise and compromise the ability to draw conclusions from the experiments performed. We have added the following information to the methods: Growing colonies were subcloned onto independent slides, until a large number of genetically identical individual colonies was established. These genotypes are propagates in mariculture for many months. Only healthy, well growing colonies with healthy

vasculature and transparent tunic were used for WBR experiments. For time point experiments, all time points were collected from subclones of the same genotype, and at least three different genotypes were used to generate averages.

Wild type marine ascidians usually contain high levels of auto-fluorescence.

Since FACS and other fluorescent based assays are used in this study, the authors need to describe in detail how they handled the background and include ISH and control EDU images as well as FACS panels of unstained tissues and cells. We performed negative controls for all the stainings at all the stages. The fluorescent signals shown are much brighter than background. We have included an image of FISH-negative control generated with sense-probes (Figure S2A) and the isotype control for the FACS antibody (Figure S1C).

2. The authors use an array of antibodies, small inhibitors, probes and primers developed against mammals or other botryllid organism. This is a major caveat and without verifying that each of these reagents indeed detects or blocks the gene / protein it was originally designed against: conclusions should not be drawn in the way presented in this paper. The probes and primers were actually designed against *B. diegensis* sequences, because we mainly used our own transcriptome database to find sequences (information now included in Methods). Most sequences labeled *B. leachii* in public databases are actually *B. diegensis* (Viard et al., 2019), and we have included this reference in the manuscript. We had previously misidentified our species based on these mistakes in the public databases, because genes we cloned aligned to published *B. leachii* sequences on NCBI. We have now corrected all errors regarding species names. All gene fragments cloned for FISH probes were confirmed by sequencing and analyzed by BLAST.

Showing the comparative sequences of Ia6 as requested above is one step in verifying the antibody target. Showing the sequences of the RNAseq or other transcriptome verification is another. You have the sequences; I didn't see them, but of course I could have missed them. So please show them.

Response: We have included sequences and alignments in the Supplement.

One of the key gene/proteins discussed in this work is Integrin alpha 6+ (IA6) (found in the *Botryllus schlosseri* genome under the name (ITGA6)).

The authors used IA6 antibodies developed against human/ mouse to sort *Botrylloides diegensis* cells without demonstrating that IA6 is expressed on these sorted cells, claiming in both the title and throughout the paper that this population expresses this protein.

Stating in the title "Integrin-alpha-6+ Stem Cells (ISCs) are responsible for whole body regeneration in an invertebrate chordate", is misleading.

The authors need to prove that this human antibody can also recognize *Botrylloides diegensis* IA6.

Mass spec or RNA sequencing might be helpful to prove that this protein / gene is expressed on the IA6+ sorted cell population but NOT on the IA6- cell population.

The use of antibodies developed for mice and human is very problematic when working with marine organisms. Firstly, the ancestor they shared diverged millions of years ago, and evolved rapidly. Secondly, KLH (derive from marine organisms) is often used to enhance the production of commercially prepared antibodies for mice and human, which can lead to non-specific staining.

The FACS plots that present the expression of this antibody (Figure S1 C) show that 90% of *Botrylloides diegensis* blood cells express this protein in high levels (above 102), suggesting a non-specific reaction. In fact, Figure S1C shows that it is 9.7% of total. We show by qPCR that *ia6* expression is highly enriched in IA6+ cells sorted by flow cytometry with the anti-IA6 antibody. The mRNA sequences of integrins are highly conserved between species. In a previous study, we used this same antibody to block binding of IA6 to laminin (Kassmer SH, 2015), proving that this antibody recognizes the correct epitope (IA6B1 is a receptor for laminin). We have included an image showing antibody staining of phosphorylated histone 3 and anti-IA6, as well as anti-POU3. Even though these antibodies do detect the proteins by fluorescence, the signal obtained by FISH is much brighter and more consistent, so we used FISH for all quantification of *ia6* and *pou3* expression.

The primers that were used in this study to estimate level of expression of several genes by qPCR (IA6, cyclin B, Vasa, Pou3, Notch 1, Notch 2, Hes 1, Frizzle 5/8, Dishevelled, Beta catenin, piwi1 and piwi 2) , and the probes that were used in this study to identify cells that express several genes (IA6, Vasa, Histone 3 Notch 1, Notch 2 by FISH) were designed against gene sequences derived from other ascidian species. Since genes among ascidian species are highly polymorphic and evolved rapidly (compare *B. schlosseri* gene model sequences to *B. leachi* gene model sequences or *Ciona* gene model sequences) one cannot assume that they will identify the same gene. The probes and primers were actually designed against *B. diegensis* sequences, because we mainly used our own transcriptome database to find sequences (information now included in Methods). Most sequences labeled *B. leachii* in public databases are actually *B. diegensis* (Viard et al., 2019), and we have included this reference in the manuscript. We had previously misidentified our species based on these mistakes in the public databases. We have now corrected all errors regarding species names. All gene fragments cloned for FISH probes were confirmed by sequencing and analyzed by BLAST. Probe sequences are now included in the supplement.

Again, if you have the full sequences of the gene products [transcriptomes] it is relatively simple to show them, and in each case show the published sequences of other species to compare. .

Response: In the supplementary information, we have included alignments of our probe sequences with publicly available sequences from a *B. leachii* transcriptome on the ANISEED database. We show alignment with human Integrin-alpha 6 and with *Botrylloides leachii* integrin-alpha 6.

Furthermore, short sequences (about 20bp) designed for primers and probes can also identify other genes and when genome assembly is not available - it is very difficult to design a specific primer or probe that will be specific.

For example: I aligned the primers designed in this paper for the IA6 against the *Botryllus schlosseri* genome (using the *B. schlosseri* genome browser at Stanford university) and found that it matched 6 different gene models including: g39337- cd109, g34568- sele, g25556- PAFAH1b1. This probe did not align to the *B. schlosseri* IA6 gene model g37640 (called ITGA6). Again, **this primer is designed against *B. diegensis*, not *schlosseri*. The probes and primers were actually designed against *B. diegensis* sequences, because we mainly used our own transcriptome database to find sequences (information now included in Methods). Most sequences labeled *B. leachii* in public databases are actually *B. diegensis* (Viard et al., 2019), and we have included this reference in the manuscript. We had previously misidentified our species based on these mistakes in the public databases. We have now corrected all errors regarding species names. All gene fragments cloned for FISH probes were confirmed by sequencing and analyzed by BLAST. Probe sequences are now included in the supplement. **Again, if you have the full sequences of the gene products [transcriptomes] it is relatively simple to show them, and in each case show the published sequences of other species to compare.****

Response: In the supplementary information, we have included alignments of our probe sequences with publicly available sequences from a *B. leachii* transcriptome on the ANISEED database. We show alignment with human Integrin-alpha 6 and with *Botrylloides leachii* integrin-alpha 6.

To prevent confusion the authors should clone the genes that are key to this project from *Botrylloides diegensis* (e.g. ITA6, Vasa, Notch and Wnt), and Sanger sequence the PCR products to verify that the primers used amplify the genes of interest. **The probes and primers were actually designed against *B. diegensis* sequences, because we mainly used our own transcriptome database to find sequences (information now included in Methods). Most sequences labeled *B. leachii* in public databases are actually *B. diegensis* (Viard et al., 2019), and we have included this reference in the manuscript. We had previously misidentified our species based on these mistakes in the public databases. We have now corrected all errors regarding species names. All gene fragments cloned for FISH probes were confirmed by sequencing and analyzed by BLAST. We tested the primers used for qPCR by melting curve analysis and BLAST. Probe sequences are now included in the supplement. **Again, if you have the full sequences of the gene products [transcriptomes] it is relatively simple to show them, and in each case show the published sequences of other species to compare.** **Response: In the supplementary information, we have included alignments of our probe sequences with publicly available sequences from a *B. leachii* transcriptome on the ANISEED database. We show alignment with human Integrin-alpha 6 and with *Botrylloides leachii* integrin-alpha 6.****

4. The results and methods sections are written in a sketchy and vague way. Number of

colonies / genotypes that been used as controls and for treatments are mentioned briefly in the table but not included in the methods or results sections.

Details regarding blastogenesis stages used in the control and experimental groups are missing. Size of fragments used for WBR experiments is not included, methods used to identify remains of secondary buds within the vasculature fragments following zooid and bud removal (that in many cases can be overlooked) are missing. FACS protocols do not include: how natural fluorescent background was handled, which stains were used to distinguish between live and dead cells? Was live/dead stain was used to measure viability of the cells following FACS sorting? As stated in the Methods: “A live cell gate was selected based on forward/side scatter properties, and samples were gated IA6 (CD49f)-positive or –negative based on isotype control staining (RatIgG2A-isotype-control eFluor450, Ebioscience, San Diego,CA, USA)”

FACS plots presenting natural background in the channel used for IA6 and other FACS data need to be presented. The percentage of cells stained by IA6 antibody needs to be clear See figure S1C. The percentage of IA6+ cells is 9.7%. As stated in the Methods: “A live cell gate was selected based on forward/side scatter properties, and samples were gated IA6 (CD49f)-positive or –negative based on isotype control staining (RatIgG2A-isotype-control eFluor450, Ebioscience, San Diego,CA, USA)”. We have added the FACS plot for the isotype control (See Figure S1C), plots of FACS IA6 stains need to be presented in main figures (not in the supplement) and images of IA6+ and IA6- sorted populations need to be included. The FACS plots are in the supplement due to lack of space in the main figures, since they are not crucial for understanding the main figure. We have added brightfield images of sorted cell populations (See new Figure S1E).

Controls that present natural background for ISH and EDU assays need to be added. We performed negative controls for all the stainings at all the stages. The fluorescent signals shown are much brighter than background. A negative control for Edu-staining was shown (now moved to Figure S3B). We have now included an image of FISH-negative control generated with sense-probes (Figure S2A) and the isotype control for the FACS antibody (Figure S1C). Furthermore, the FISH on MMC treated samples shows no signal for pou3, h3 or ia6 (Figure S1A), confirming that the signal obtained by antisense-FISH probes is specific and above background.

Information regarding

statistical analysis performed and how the sample size per each experiment was defined should be included. This information is included in all figure legends.

No information is provided regarding the transplantation essays. Transplantation of a small number of cells or even a single cell is the most difficult experiment in stem cell studies. It is very difficult to assess the value of this experiment without providing any description of how transplantation was accomplished, including a detailed description that include images and videos that demonstrate the ability to transplant low number of cells into small fragments of vasculature. Data regarding the instruments used to inject

cells, how cells were counted, how limiting dilutions were calculated to achieve a single cell transplantation, staining used to check viability of cells following FACS sorting and the stains used to label cells before transplantation to verify successful transplantation all need to be included.

We have added the following information to the Methods: Sorted cells were collected into 3ml of filtered seawater, pelleted by centrifugation (700g, 10 minutes) and resuspended in 100ul filtered seawater. Cells were counted using a hemocytometer. Only round, bright, intact cells were counted. Cells were labeled with Hoechst 33342 and diluted in filtered seawater to the appropriate concentration. 0.1ul volume containing either 1000, 50, 10, 5 or 3 cells was injected per sample. For injections of 10 cells or less, 1ul cell suspension stained with Hoechst 33342 was pipetted onto a microscope slide and cells were counted under brightfield and fluorescence to verify the correct cell concentration. For single cell injections, cells were diluted so that 1ul contained 5 cells, and 0.1ul were injected per sample = 0.5 cells per sample. Cells were injected into ampullae of MMC treated vessel fragments with pulled glass capillary needles and an Eppendorf FemtoJet microinjector under a stereomicroscope.

Tunicates are known to live in highly contaminated areas (harbors) and present robust regeneration and survival capacities. Therefore, several important controls are needed to test their ability to resume regeneration following stress in order to understand their regeneration abilities under random stressors and understand the effect of the diverse treatments and experiments done in this study. We have added the following information to the methods: Growing colonies were subcloned onto independent slides, until a large number of genetically identical individual colonies was established. These genotypes are propagated in mariculture for many months. Only healthy, well growing colonies with healthy vasculature and transparent tunic were used for WBR experiments. For time point experiments, all time points were collected from subclones of the same genotype, and at least three different genotypes were used to generate averages.

Other points

lines 15-19: “In the majority of chordates, the ability to regenerate following a major injury is severely limited, usually resulting in scar formation. In contrast, a group of invertebrate chordate species; colonial ascidians of the genus Botrylloides, have been shown to regenerate whole bodies, including all tissues and organs, from small fragments of the vasculature. This process is called Whole Body Regeneration (WBR)”. This is repeating a sentence from the manuscript. We don’t understand if this is a question or a comment?

Add references to support claims. Relevant references have been cited where appropriate.

Were all relevant references cited? **Response:** yes

Lines 55-57: “The source of the new bodies is a specialized region of the body wall of the parental zooid called the peribranchial epithelium, and this process of asexual reproduction is called palleal budding (Kassmer et al., 2018).” This is repeating a

sentence from the manuscript. We don't understand if this is a question or a comment?
Were all relevant references cited? **Response:** yes

Authors need to read and cite studies done by Manni lab (Padova university) that describe budding in *Botryllus schlosseri*. Buds do not originate in the body wall but in the peribranchial epithelium which is connected to the branchial sac. **We have changed this sentence to: "The source of the new bodies is a specialized region of epithelium surrounding the branchial sac of the parental zooid called the peribranchial epithelium, and this process of asexual reproduction is called paleal budding (Brown and Swalla, 2012; Kassmer et al., 2018; Kawamura et al., 2008; Manni et al., 2014; Oka, 1959)."**

"In addition to paleal budding, which occurs normally"

Authors need to describe in detail or refer to previous studies (if any) that describe paleal budding in *Botrylloides diegensis*. What are the normal budding stages of development in *Botrylloides diegensis*? Is it synchronized along the colony, how long does it take under specific temperatures? Do they hibernate like other botryllid species? **The process of paleal budding is identical in within the Botryllinae (Alie et al., 2018; Brown and Swalla, 2012; Oka, 1959). We have added additional references. We have observed hibernation of *Botrylloides diegensis* during sudden changes in temperature when cultured in a facility supplied with raw seawater. This is not surprising given that many species reported to be *B. leachii* in the literature are actually *B. diegensis*. Since hibernation is not the subject of this study, we did not include these observations but are in the process of studying the relationship between WBR and recovery from hibernation.**

Lines 62-80: "*B. diegensis* can also use the alternative WBR pathway to produce a zooid following injury. WBR occurs in the vasculature, and is initiated by surgical removal of all zooids, or separation of small vascular fragments from the rest of the colony (Figure 1B). WBR progresses through distinct visual stages. For the first 24 hrs following surgical ablation, the blood vessels undergo regression and remodeling (stage 1, Figure 1B). During this time, blood flow, usually powered by the hearts of each zooid, continues and is driven by contractions of the remaining ampullae (Blanchoud et al., 2017). During the next 48 hrs, the blood vessels continue to remodel and form a dense, contracted network (stage 2, Figure 1B). Within this dense, highly pigmented vascular tissue, an opaque mass of non-pigmented cells becomes apparent; creating a clear area that is the presumptive site of bud development (stage 3, arrow in Figure 1B). The mass of cells next forms into a hollow, blastula-like epithelial sphere. The vascular epithelium then wraps itself around this sphere, leading to the formation of a distinct visible double vesicle. Over the next 48h, the inner vesicle increases in size (Stage 4, Figure 1B), while undergoing a series of invaginations and evaginations that lead to organogenesis and the eventual regeneration of a zooid. WBR is defined as complete when the new zooid is actively filter-feeding, and occurs within a range of 7-10 days (stage 5; Figure 1B). The zooid

immediately commences normal palleal budding, and the colony regrow”.

This section which discussed WBR in *B. diegensis* includes new information and should be part of the results. It should include the number of fragments tested and size and developmental stage used for WBR. Are the statements of “blastula like epithelial sphere” and “vascular epithelium wraps itself around this sphere” based on in vivo images? Or histological sections? If this is based on histological sections they should be included. If not, these terms cannot be used. **The morphological stages of whole body regeneration have been described in detail in previous studies, and we give those references. Therefore, these are not new results. We are merely confirming that we can reproduce the findings of others and that our animals undergo WBR in the same manner as described previously. We have added the following statement: “WBR can be induced in fragments as small as 5 ampullae, and is not dependent on the stage of asexual reproduction of the colony.” We show the morphology of the double vesicle in Figure 1B, and in the cryosections in Figure S3B, the morphology of the two epithelial layers is clearly evident. Figure 2A shows early stages of double vesicle formation, and the two layers are evident there as well. Therefore, we feel that using this terminology is very correct.**

Lines 83-86 “WBR has been studied in three different species of botryllid ascidians, and in all cases a population of cells with an undifferentiated appearance, termed hemoblasts, have been suggested to initiate this regenerative process (Brown et al., 2009a)”.

Please include and cite other works on vascular budding or WBR in botryllid ascidian including: Sabbadin et al. 1975; Miyamoto and Freeman, 1970; Rinkevich et al. 1995; Voskoboynik et al. 2007. **We are strictly distinguishing between vascular budding that occurs as part of normal colony development - as in *B. primigenus* and *B. tuberatus* – and whole body regeneration, which is induced by removal of all zooids and buds (Kassmer et al., 2019). *Botrylloides leachii*, *Botrylloides diegensis* and *Botrylloides violaceus* all undergo WBR, whereas *B. primigenus* and *B. tuberatus* undergo vascular budding. *B. schlosseri* does neither. We have edited this section to talk about *Botrylloides* only.**

What do you mean by *B. schlosseri* does neither? When all buds and zooids are removed during the takeover stage vascular budding is induced in *B. schlosseri* (e.g. Sabbadin et al. 1975; Voskoboynik et al. 2007). **Response:** This is not relevant here because due to space constraints we can only explain and reference WBR in *Botrylloides*. Studies in other species are not as relevant here. In addition, WBR in *Botrylloides* is much better studied and is not stage dependent. We have published a review on regeneration in tunicates that includes studies on *B. schlosseri* and are referencing it (Ref 10).

Our lab has unpublished data suggesting that true WBR does not occur in *B. schlosseri* (at least not in genotypes we have collected in Southern California), but we are not showing this data here.

Lines 88-92: “In *Botrylloides violaceus*, 15-20 small hemoblasts that express Piwi

protein have been shown to aggregate under the epidermis of a blood vessel during early WBR (Oka, 1959). These cells can be detected during the early vesicle stage and occasionally within the epithelium of a vesicle, suggesting that they play a role in regeneration (Brown et al., 2009a). In a related species, *Botrylloides leachii*, Blanchoud et al. showed an increase in the population of hemoblasts very early after injury during WBR (Blanchoud et al., 2017)".

Please include the work of Rinkevich et al. 2007 Plos Biology on *Botrylloides leachii* WBR **We have included this reference**

Lines 97-101: "In several species of colonial ascidians, the blood contains self-renewing, lineage restricted germline stem cells that migrate to newly developing buds during repeated cycles of asexual reproduction, where they give rise to eggs and testes (Brown et al., 2009b; Carpenter et al., 2011; Kawamura and Sunanaga; Laird et al., 2005; Rodriguez et al., 2016; Sunanaga et al., 2010; Sunanaga et al., 2006)".

Up to date self-renewal of stem cells was proven only in *B. schlosseri* (Laird et al. 2005) and not in any other botryllid species. Please do not mislead the readers and suggest that it was proven in other botryllid species. **We have edited this section to clearly state that germline stem cells were identified in *B. schlosseri*, and edited the references. It is true that germline parasitism has been proven only in *B. schlosseri*, and this parasitism has been used to formally demonstrate that long-lived germline stem cells are present in the blood of this species. While these experiments have not yet been repeated in *Botrylloides*, the fact that each asexual generation and colonies generated from WBR have a complete germline is strong evidence that long-lived germline stem cells that settle in gonadal niches of asexual generations exist in all Botryllinae.**

Authors should also add the original works done on *Botryllus schlosseri* chimeras which were the first to demonstrate the presence stem cells in the of blood of this organism : Sabbadin and Zaniolo, Stoner and Weissman 1966; Stonner et al. 1998; Voskoboynik et al. 2008; Rinkevich et al. 2013 **Since this study is focused on WBR in *Botrylloides diegensis*, we cannot include so much detail on germline development and germ cell migration in *Botryllus schlosseri* in the introduction. For interested readers, gonad development germ cell migration and germline parasitism in *B. schlosseri* has been reviewed in (Rodriguez et al., 2017).**

We have now included Rinkevich et al. 2013 because of the piwi knockdown. Thank you for reminding us!

Lines 101-105: In *Botryllus schlosseri*, these germline stem cells can be enriched by flow cytometry using Integrin-alpha-6 (IA6) as a marker (Kassmer SH, 2015) and express piwi as well as other genes associated with germ cells, such as vasa, and pumilio (Brown et al., 2009b; Langenbacher and De Tomaso, 2016; Sunanaga et al., 2006).

Please highlight the limitation of the use of antibodies designed against mammalian cells in these studies **We show by qPCR that *ia6* expression is highly enriched in IA6+ cells sorted by flow cytometry with the anti-IA6 antibody. The mRNA sequences of integrins are highly conserved between species. In a previous study, we used this same antibody to block binding of IA6 to laminin (Kassmer SH, 2015), proving that this antibody recognizes the correct epitope (IA6B1 is a receptor for laminin).**

As mentioned above, please provide the sequences from your own studies. Response: done

and add the work done by Rinkevich et al. 2013 (Dev Cell) which used classic transplantation experiments and genotyping to demonstrate that the cell islands in *Botryllus schlosseri* are enriched with germline stem cells. **Since this study is focused on WBR in *Botrylloides diegensis*, we cannot include so much detail on germline development and germ cell migration in *Botryllus schlosseri* in the introduction. For interested readers, gonad development germ cell migration and germline parasitism in *B. schlosseri* has been reviewed in (Rodriguez et al., 2017).**

References relevant to the responses:

- Alie, A., Hiebert, L.S., Simion, P., Scelzo, M., Prunster, M.M., Lotito, S., Delsuc, F., Douzery, E.J.P., Dantec, C., Lemaire, P., *et al.* (2018). Convergent Acquisition of Nonembryonic Development in Styelid Ascidians. *Mol Biol Evol* 35, 1728-1743.
- Brown, F.D., and Swalla, B.J. (2012). Evolution and development of budding by stem cells: ascidian coloniality as a case study. *Dev Biol* 369, 151-162.
- Carpenter, M.A., Powell, J.H., Ishizuka, K.J., Palmeri, K.J., Rendulic, S., and De Tomaso, A.W. (2011). Growth and long-term somatic and germline chimerism following fusion of juvenile *Botryllus schlosseri*. *Biol Bull* 220, 57-70.
- Kassmer, S.H., Nourizadeh, S., and De Tomaso, A.W. (2018). Cellular and molecular mechanisms of regeneration in colonial and solitary Ascidians. *Dev Biol*.
- Kassmer, S.H., Nourizadeh, S., and De Tomaso, A.W. (2019). Cellular and molecular mechanisms of regeneration in colonial and solitary Ascidians. *Dev Biol* 448, 271-278.
- Kassmer SH, R.D., Langenbacher AD, Bui C, De Tomaso AW (2015). Migration of germline progenitor cells is directed by sphingosine-1-phosphate signalling in a basal chordate. *Nature Communications* 6.
- Kawamura, K., Sugino, Y., Sunanaga, T., and Fujiwara, S. (2008). Multipotent epithelial cells in the process of regeneration and asexual reproduction in colonial tunicates. *Dev Growth Differ* 50, 1-11.
- Manni, L., Gasparini, F., Hotta, K., Ishizuka, K.J., Ricci, L., Tiozzo, S., Voskoboynik, A., and Dauga, D. (2014). Ontology for the asexual development and anatomy of the colonial chordate *Botryllus schlosseri*. *PLoS One* 9, e96434.
- Oka, H.W.H. (1959). VASCULAR BUDDING IN BOTRYLLOIDES. *The Biological Bulletin* 117, 340-346.
- Rinkevich, Y., Rosner, A., Rabinowitz, C., Lapidot, Z., Moiseeva, E., and Rinkevich, B. (2010). Piwi positive cells that line the vasculature epithelium, underlie whole body regeneration in a basal chordate. *Dev Biol* 345, 94-104.

Rodriguez, D., Kassmer, S.H., and De Tomaso, A.W. (2017). Gonad development and hermaphroditism in the ascidian *Botryllus schlosseri*. *Mol Reprod Dev* 84, 158-170.

Viard, F., Roby, C., Turon, X., Bouchemousse, S., and Bishop, J. (2019). Cryptic Diversity and Database Errors Challenge Non-indigenous Species Surveys: An Illustration With *Botrylloides* spp. in the English Channel and Mediterranean Sea. *Front Mar Sci*.